# Minimax-Optimal Univariate Function Selection in Sparse Additive Models: Rates, Adaptation, and the Estimation-Selection Gap

**Shixiang Liu**
School of Statistics
Renmin University of China
`liushixiang_stat@ruc.edu.cn`

## Abstract

The sparse additive model (SpAM) offers a trade-off between interpretability and flexibility, and hence is a powerful model for high-dimensional research. This paper focuses on the variable selection, i.e., the univariate function selection problem in SpAM. We establish the minimax separation rates from both the perspectives of sparse multiple testing (FDR + FNR control) and support recovery (wrong recovery probability control). We further study how adaptation to unknown smoothness affects the minimax separation rate, and propose an adaptive selection procedure. Finally, we discuss the difference between estimation and selection in SpAM: Procedures achieving optimal function estimation may fail to achieve optimal univariate function selection.

## 1 Introduction

The Sparse Additive Model (SpAM) is a pivotal topic of recent statistical research [Ravikumar et al., 2009, Meier et al., 2009, Koltchinskii and Yuan, 2010, Raskutti et al., 2012, Dalalyan et al., 2014, Yuan and Zhou, 2016, Tyagi et al., 2016, Tan and Zhang, 2019, Haris et al., 2022]. It extends the generalized additive model [Hastie and Tibshirani, 1987], balancing interpretability and flexibility while avoiding the curse of dimensionality and adapting to high-dimensional settings.

In this paper, we focus on the variable selection, i.e., univariate function selection problem of the SpAM, which is a fundamental problem with broad implications in multi-channel detection [Ingster and Lepski, 2003], multi-task learning [Wang et al., 2020], sparse neural network [Xu et al., 2023], and so on. We consider a Gaussian white noise (GWN) model with $p$ covariates $x = (x_1, \cdots, x_p) \in \mathcal{X}^p$, which takes the form as

$$\mathrm{d}Y_x = f(x)\mathrm{d}x + \sigma \mathrm{d}B_x = \sum_{j=1}^p f_j(x_j)\mathrm{d}x_j + \sigma \mathrm{d}B_x,$$

where $\mathcal{X}$ is the domain of each covariate $x_j$, $B_x$ is a standard Wiener process on $\mathcal{X}^p$, and $\sigma > 0$ measures the intensity of the white noise. We assume $f_j$ is the univariate function corresponding to variable $x_j$. Under the setting of sparsity, the response $Y_x$ is influenced by no more than $s$ covariates, and hence $f$ can be expressed as $f(x) = \sum_{j \in S_f} f_j(x_j)$, where $S_f \subseteq \{1, \cdots, p\}$ is the index set of these support covariates. In this continuous-time SpAM framework, our main goal is to recover the index set $S_f$, i.e., to select which $f_j \neq 0$. This paper studies the univariate function selection in SpAMs from two perspectives—namely, as a sparse multiple testing problem and as a support recovery problem. We employ the truncated procedures and establish the non-asymptotic minimax separation rates, delivering, to our knowledge, the first optimal finite-sample guarantees for univariate function selection in SpAMs.

39th Conference on Neural Information Processing Systems (NeurIPS 2025).

## 1.1 Related work

**Background of variable selection**  The variable selection problem has attracted significant interest recently [Butucea et al., 2018, Rabinovich et al., 2020, Belitser and Nurushev, 2022, Song and Cheng, 2023, Butucea et al., 2023a, Abraham et al., 2024]. The general assumption is that the response depends on only a few covariates, and the main aim is to find them. This problem can be framed as either a sparse multiple testing problem (controlling False Discovery Rate (FDR), False Negative Rate (FNR), etc.) [Rabinovich et al., 2020, Song and Cheng, 2023, Abraham et al., 2024], or a support recovery problem (controlling Hamming loss) [Wainwright, 2007, Butucea et al., 2018, Gao and Stoev, 2020, Butucea et al., 2023a], based on different setting of the loss function. Much of the existing studies concentrated on the sparse sequence model $X_i = \beta_i + \epsilon_i$, $i = 1, \cdots, p$ independently, with assuming $\sum_{i=1}^{p} \mathbf{1}(\beta_i \neq 0) \leq s$ and each $\epsilon_i$ drawn from distributions like Gaussian [Butucea et al., 2018, Song and Cheng, 2023] or generalized Gaussian [Gao and Stoev, 2020, Rabinovich et al., 2020, Abraham et al., 2024]. Though these studies demonstrated interesting phase-transition phenomena and established the asymptotically sharp minimax separation rate, they cannot be directly applied to the univariate function selection in SpAM.

**Background of univariate function selection in SpAM**  Univariate function selection in SpAM stands as a pivotal problem in statistical learning [Lin and Zhang, 2006, Ravikumar et al., 2009, Huang et al., 2010, Chouldechova and Hastie, 2015, Xu et al., 2016, Wood et al., 2015, Butucea and Stepanova, 2017, Dai et al., 2023]. Most existing studies firstly provided minimax-optimal estimators for the function $f$ via M-estimation with group-lasso-type penalties on each $f_j$. Then, by utilizing the estimation results, the selection performances were often established as by-products [Ravikumar et al., 2009, Huang et al., 2010, Dai et al., 2023]. Although these methods ensured asymptotic variable selection consistency [Ravikumar et al., 2009, Huang et al., 2010] or FDR control [Dai et al., 2023], they did not guarantee minimax optimality for the univariate function selection problem. This implies that their minimum signal conditions, typically quantified by "$\min_{j \in S_f} \|f_j\|_2^2 \geq$ some rate", are sufficient but not necessary: Their signal strength assumptions may be overly restrictive.

**Existing optimal univariate function selection in SpAM**  From the viewpoint of support recovery with Hamming loss, Ingster and Stepanova [2014] and Butucea and Stepanova [2017] provided the minimax optimal (i.e., necessary and sufficient) signal condition for exact support recovery and almost-full support recovery, respectively. Comminges and Dalalyan [2012] analyzed support recovery in a $p$-dimensional nonparametric regression with an intrinsic $s$-variate underlying function. In an additive model allowing $k$-dimensional interaction effects, Stepanova and Turcicova [2025a,b] provided the optimal signal condition for exact support recovery. These studies offered asymptotically minimax optimal results in some specific function classes, but may not be persuasive in the general function space with a finite sample size. For instance, they rely on certain additional assumptions, like $\log p = o(\sigma^{-2/(2\alpha+1)})$ and $\sigma \to 0$, in the Sobolev space with smoothness parameter $\alpha$.

**Inspiration from cutting-edge work**  Building on the monotone likelihood ratio property, Butucea et al. [2023a] recently established rate-optimal signal conditions for support recovery under group sparsity, improving upon conclusions from Lounici et al. [2011]. Kotekal and Gao [2024] extended the hard-thresholding estimator of Collier et al. [2017] to develop a minimax optimal goodness-of-fit test for SpAM (i.e., testing whether $f = \sum_{j \in S_f} f_j = 0$). These advances motivate the development of a non-asymptotic minimax optimal univariate function selector within a generalized SpAM framework, covering Sobolev-smooth, analytic, and other function classes.

## 1.2 Main contributions and organization

This paper answers the following questions:

> *In a generalized SpAM framework, can we achieve non-asymptotic and minimax optimal univariate function selection? What is the difference between function estimation and univariate function selection?*

The main contributions are threefold:

1. **Minimax separation rates**  From both the viewpoints of sparse multiple testing (FDR+FNR control) and support recovery (wrong recovery probability control), we establish

the non-asymptotic minimax separation rates for univariate function selection in a generalized SpAM framework. This result is, to our knowledge, the first optimal finite-sample guarantees. We also develop truncated-type selectors to achieve the minimax rate-optimality, respectively.

2. **Minimax adaptation**   We provide a rate-optimal selection procedure that adapts to the smoothness parameter of the Sobolev spaces. We show that an additional $\log\left(\log(\sigma^{-2})\right)$ term in the signal condition is required for this adaptation.

3. **Difference between estimation and selection**   Within the class of truncated-type estimators, we demonstrate that the optimal function estimations can not yield optimal univariate function selection in some cases. This gap underscores the necessity to proceed differently in selection versus estimation, a finding with deep statistical implications.

The rest of the paper is organized as follows: Section 1 establishes the notation used throughout the paper. Section 2 introduces the model setup and the background of our problem. Section 3 establishes the minimax separation rates for univariate function selection from two viewpoints. Section 4 provides a rate-optimal selector adaptive to the smoothness parameter in the Sobolev space. Section 5 offers an in-depth discussion about the difference between estimation and selection in SpAMs. The limitations, future directions, numerical experiments, and all technical proofs are provided in the appendices.

### 1.3   Notation

For the given sequences $a_n$ and $b_n$, we write that $a_n = O(b_n)$ and $a_n \lesssim b_n$ (resp. $a_n = \Omega(b_n)$ and $a_n \gtrsim b_n$) if $a_n \leq cb_n$ (resp. $a_n \geq cb_n$) for some absolute positive constant $c$. We write that $a_n \asymp b_n$ if $a_n = O(b_n)$ and $b_n = O(a_n)$. Denote by $[m]$ the set $\{1, 2, \cdots, m\}$, and $\mathbf{1}(\cdot)$ the indicator function. Denote by $x \vee y$ the maximum of $x$ and $y$, and $x \wedge y$ the minimum of $x$ and $y$. Denote by $S_f = \{j \in [p] : f_j \neq 0\} \subseteq [p]$ the support univariate function set of a SpAM function $f$. For a square intergral function $f$ with support $\mathcal{X}$, denote by $\|f\|_2 = \left(\int_{\mathcal{X}} f^2(x)\mathrm{d}x\right)^{1/2}$ its $L_2$ norm. Let $C, C_0, C_1, \cdots$ denote absolute positive constants whose values may change from one occurrence to the next.

## 2   Preliminary and problem setup

Let us recall that we observe $Y_x$ and $x \in \chi^p$ such that

$$\mathrm{d}Y_x = \sum_{j \in S_f} f_j(x_j)\mathrm{d}x + \sigma\mathrm{d}B_x. \tag{1}$$

To ensure the identifiability of univariate functions, we assume $\int_{\mathcal{X}} f_j(x_j)\mathrm{d}x_j = 0$ for each $j \in [p]$. In theoretical research, the GWN model and nonparametric regression model are asymptotically equivalent, as shown by Brown and Low [1996], Reiß [2008][1]. Moreover, the GWN model simplifies the analysis by avoiding unnecessary technical complexities while keeping the focus on the statistical essence [Kotekal and Gao, 2024]. Consequently, many foundational nonparametric statistics theories are developed based on the GWN model [Fan, 1991, Donoho and Johnstone, 1998, Baraud, 2002, Tsybakov, 2009, Comminges and Dalalyan, 2012, Johnstone, 2017, Han et al., 2020]. Therefore, to maintain this theory-driven tradition, we conduct our analysis based on the GWN model (1).

### 2.1   Function settings

We propose a general smoothness assumption based on the series expansion of univariate functions. For each $j \in [p]$, assume that $f_j : \mathcal{X} \to \mathbb{R}$ can be decomposed from an orthonormal basis $\{\psi_i\}_{i \in \mathbb{N}^+}$, as $f_j(x_j) = \sum_{i=1}^{\infty} \theta_{ij}\psi_i(x_j)$, where $\theta_{ij} = \theta_{ij}(f_j) := \int_{\mathcal{X}} \psi_i(x_j)f_j(x_j)\mathrm{d}x_j$ is the coefficient of $\psi_i$ for each $i \in \mathbb{N}^+$. Define $\theta_{\cdot j} = \theta_{\cdot j}(f_j) := \{\theta_{ij}\}_{i \in \mathbb{N}^+}$. We assume that each $f_j$ is sufficiently smooth and belongs to the ellipsoid class

$$\mathcal{E} := \left\{ f_j = \sum_{i=1}^{\infty} \theta_{ij}\psi_i : \sum_{i=1}^{\infty} \frac{\theta_{ij}^2}{\mu_i} \leq 1 \right\}, \tag{2}$$

---

[1]Also see Section 1.10 of Tsybakov [2009] for the connection between the GWN and nonparametric regression.

where $\{\mu_i\}_{i=1}^{\infty}$ is a non-increasing sequence of positive numbers, i.e., $\mu_1 \geq \mu_2 \geq \cdots$, and we assume $\mu_1 \asymp 1$ to ensure $f_j$ has finite $L_2$ norm. This ellipsoid setting is a broad smoothness assumption that renders our theoretical results applicable to Reproducing Kernel Hilbert Space (RKHS) [Raskutti et al., 2012, Yuan and Zhou, 2016, Kotekal and Gao, 2024], Fourier basis [Comminges and Dalalyan, 2012, Ingster and Stepanova, 2014, Butucea and Stepanova, 2017], etc. The function space of SpAM is defined as

$$\mathcal{F}_s := \left\{ f(x) = \sum_{j=1}^{p} f_j(x_j) : \sum_{j=1}^{p} \mathbf{1}(f_j \neq 0) \leq s, \ f_j \in \mathcal{E} \text{ for all } j \in [p] \right\}. \tag{3}$$

Each $f \in \mathcal{F}_s$ corresponds uniquely to a $\Theta = \Theta(f) := (\theta_{\cdot 1}(f_1), \cdots, \theta_{\cdot p}(f_p)) \in \mathbb{R}^{\mathbb{N}^+ \times p}$. Therefore, $f \in \mathcal{F}_s$ and $\Theta \in \mathcal{F}_s$ will be used interchangeably in the subsequent text. For every $f \in \mathcal{F}_s$ and every $i \in \mathbb{N}^+, j \in [p]$, based on the continuous process $Y_x$ in model (1), we have access to the following random variables

$$X_{ij} := \int_{\mathcal{X}^p} \psi_i(x_j) \mathrm{d}Y_x = \theta_{ij} + \int_{\mathcal{X}^p} \psi_i(x_j)\sigma \mathrm{d}B_x \sim N(\theta_{ij}, \sigma^2).$$

By orthogonality, the set $X = \{X_{ij}\}_{i \in \mathbb{N}^+, j \in [p]}$ is a collection of independent random observations.

## 2.2 Problem setup

Within the SpAM space $\mathcal{F}_s$, our primary task is to establish a minimax optimal (i.e., **necessary and sufficient**) signal condition of each support $f_j$, for the univariate function selection. Before delving into our analysis, we revisit the function estimation problem in SpAM, where Raskutti et al. [2012] established the minimax rate as:

$$\inf_{\hat{f}} \sup_{f \in \mathcal{F}_s} \mathbf{E}_f \left( \left\| \hat{f}(X) - f \right\|_2^2 \right) \asymp s \times \underbrace{\sigma^2 \log(ep/s)}_{\text{High-dimensional selection error}} + s \times \underbrace{\max_{k \in \mathbb{N}^+} \left( (\sigma^2 k) \wedge \mu_k \right)}_{\inf_{\widehat{f_j}} \sup_{f_j \in \mathcal{E}} \mathbf{E}_{f_j} \left\| \widehat{f_j}(X_{\cdot j}) - f_j \right\|_2^2}, \tag{4}$$

which is composed of $s$ times the "high-dimensional selection error" and $s$ times the "minimax estimation rate of a single univariate function", with **no interplay** between these two parts. This result shows that the first term $\sigma^2 s \log(ep/s)$ is independent of the univariate function space $\mathcal{E}$, and the estimation term (the second term) is dimension-free ($p$-free) [Kotekal and Gao, 2024]. Therefore, it is natural to **speculate** that the univariate function selection shares a similar property, with its optimal signal condition, quantified by the squared $L_2$ norm, of the rate:

$$\underbrace{\sigma^2 \log(ep/s)}_{\text{High-dimensional selection error}} + \max_{k \in \mathbb{N}^+} \left( (\sigma^2 \sqrt{k}) \wedge \mu_k \right), \tag{5}$$

where the second term is the minimax separation rate for the goodness-of-fit test of a single univariate function in $\mathcal{E}$ [Baraud, 2002].

However, in Section 3 we prove that this is not the case. In univariate function selection, there is an **interplay** between the high-dimensional sparse structure (selection error) and the ellipsoid space $\mathcal{E}$, complicating the form of its minimax rate.

## 3 Main result: optimal univariate function selection

In this section, we demonstrate that the truncated-type selectors lead to minimax optimal results. Define the decoder $\eta_j = \eta_j(f_j) := \mathbf{1}(f_j \neq 0)$, and the corresponding vector $\eta = \eta(f) := (\eta_1(f_1), \cdots, \eta_p(f_p)) \in \{0,1\}^p$. We also define the selector, i.e., the estimation of $\eta$, as $\hat{\eta} = \hat{\eta}(X) = (\hat{\eta}_1(X), \cdots, \hat{\eta}_p(X)) \in \{0,1\}^p$, and $\hat{S} = \{j \in [p] : \hat{\eta}_j = 1\}$ as the estimated support set corresponding to $\hat{\eta}$. Define the SpAM space with the signal strength condition as

$$\mathcal{F}_s(r^2) := \left\{ f = \sum_{j \in [p]} f_j \in \mathcal{F}_s : \|f_j\|_2^2 \geq r^2 \text{ for all } f_j \neq 0 \right\}, \tag{6}$$

indicating that each support $f_j$ has a signal separated from 0. Here $r^2$ is a positive value and we additionally assume $r^2 \leq \mu_1$ to ensure $\mathcal{F}_s(r^2) \neq \emptyset$ (since $\|f_j\|_2^2 \leq \mu_1 \sum_i \frac{\theta_{ij}^2}{\mu_i} \leq \mu_1$ based on $f_j \in \mathcal{E}$). In the next two subsections, we derive the minimax separation rates from two viewpoints, sparse multiple testing and support recovery, respectively.

### 3.1 From sparse multiple testing: FDR + FNR control

**Preliminary setup** From the viewpoint of testing, the selection can be realized as a multiple-testing problem

$$H_{0j} : f_j = 0, \quad H_{1j} : f_j \neq 0, \quad \text{for all } j \in [p],$$

under the exactly $s$-sparse function space

$$f \in \mathcal{F}_{=s}(r^2) := \left\{ f = \sum_{j \in [p]} f_j \in \mathcal{F}_s(r^2) : \sum_{j \in [p]} \mathbf{1}(f_j \neq 0) = s \right\}.$$

We consider the multiple testing risk combined with the false discovery rate (FDR) plus the false negative rate (FNR), which is of the form

$$R(f, \hat{\eta}) = \mathbf{E}_f \left( \frac{\sum_{j \notin S_f} \hat{\eta}_j}{1 \vee \sum_{j \in [p]} \hat{\eta}_j} + \frac{\sum_{j \in S_f} (1 - \hat{\eta}_j)}{s} \right).$$

This combined risk balances the proportion of type I and type II errors, and is frequently used in the sparse testing [Arias-Castro and Chen, 2017, Rabinovich et al., 2020, Abraham et al., 2024].

**Definition 1 (Minimax separation rate of sparse multiple testing)** *We say $\epsilon_{test}^2$ is the non-asymptotic minimax separation rate of the sparse multiple testing problem for* (1) *if:*

*(1) For all $\delta \in (0, 1)$, there exists $c_\delta > 0$ depending only on $\delta$ such that for all $0 < c < c_\delta$,*

$$\inf_{\hat{\eta}} \sup_{f \in \mathcal{F}_{=s}(c\epsilon_{test}^2)} R(f, \hat{\eta}) \geq 1 - \delta.$$

*(2) For all $\delta \in (0, 1)$, there exists $C_\delta > 0$ depending only on $\delta$ such that for all $C > C_\delta$,*

$$\inf_{\hat{\eta}} \sup_{f \in \mathcal{F}_{=s}(C\epsilon_{test}^2)} R(f, \hat{\eta}) \leq \delta,$$

*where $\inf_{\hat{\eta}}$ denotes the infimum over all selector $\hat{\eta}(X) : \mathbb{R}^{\mathbb{N}^+ \times p} \to \{0, 1\}^p$.*

**$K$-truncated selector** For each sequence $X_{\cdot j}$, we truncate by the first $K$ entries and construct the corresponding selector

$$\hat{\eta}_j^{test}(X_{\cdot j}) = \mathbf{1}\left( \sum_{i=1}^K X_{ij}^2 \geq \sigma^2 K + \lambda^2(K) \right), \quad j \in [p], \tag{7}$$

where the truncation $K := \min\left\{ k \in \mathbb{N}^+ : \mu_k \leq \sigma^2 \sqrt{k \log(p/s)} \right\}$, and the parameter $\lambda^2(K)$ will be determined in Theorem 1. Denote by $\hat{\eta}^{test} = (\hat{\eta}_1^{test}, \cdots, \hat{\eta}_p^{test}) \in \{0, 1\}^p$ the corresponding selector vector. The following theorem employs an analysis to control the combined risk at a low level.

**Theorem 1 (Upper bound for sparse multiple testing)** *Let $\delta$ be an arbitrary number in $(0, 1)$, and assume that $\sigma^{-2} > \frac{C_{\delta,1} \log(p/s)}{\mu_1}$, $p/s \geq C_{\delta,2}$, and $s \geq C_{\delta,3}$. Then, assuming*

$$r^2 \geq \left( \frac{6}{\sqrt{\delta}} \left( \sqrt{10} + 2 \right) + \sqrt{2} \right) \max_{k \in \mathbb{N}^+} \left( \sigma^2 \sqrt{k \log(p/s)} \wedge \mu_k \right) + \frac{36}{\delta} \sigma^2 \log(p/s) \tag{8}$$

*and taking*

$$\lambda^2(K) = 2\sigma^2 \left( \sqrt{5K \log\left(\frac{p}{s\delta}\right)} + 5 \log\left(\frac{p}{s\delta}\right) \right),$$

*we have*

$$\sup_{f \in \mathcal{F}_{=s}(r^2)} R(f, \hat{\eta}^{test}) \leq \delta,$$

*where $C_{\delta,1}, C_{\delta,2}, C_{\delta,3}$ are positive constants only determined by $\delta$.*

The next theorem shows that the rate in (8) is also necessary for controlling the testing risk.

**Theorem 2 (Lower bound for sparse multiple testing)** *Let $\delta$ be an arbitrary number in $(0, 1)$, and assume that $\sigma^{-2} > \frac{C_{\delta,1} \log(p/s)}{\mu_1}$, $p/s \geq C_{\delta,2}$, and $s \geq C_{\delta,3}$. Then, for all $r^2$ satisfies*

$$0 < r^2 \leq c_{\delta,4} \left\{ \sigma^2 \log(p/s) + \max_{k \in \mathbb{N}^+} \left( \sigma^2 \sqrt{k \log(p/s)} \wedge \mu_k \right) \right\},$$

*we have*

$$\inf_{\hat{\eta}} \sup_{f \in \mathcal{F}_{=s}(r^2)} R(f, \hat{\eta}) \geq 1 - \delta,$$

*where $C_{\delta,1}, C_{\delta,2}, C_{\delta,3}$ and $c_{\delta,4}$ are four positive constants only determined by $\delta$.*

Therefore, combining Theorem 1 and 2, we establish the minimax separation rate for the sparse multiple testing in the SpAM (1) as

$$\epsilon_{test}^2 \asymp \sigma^2 \log(p/s) + \max_{k \in \mathbb{N}^+} \left( \sigma^2 \sqrt{k \log(p/s)} \wedge \mu_k \right). \tag{9}$$

We also illustrate that a truncated-type selector possesses such minimax optimality.

**Remark 1 (Truncation)** *So far, the equation (9) reveals that our initial speculation (5), in the end of Section 2.2, is inaccurate: The high-dimensional sparsity structure influences both terms in the minimax separation rate. This is because the selection problem is related to the chi-squared distribution, whose heavy tail leads to the selection error of the rate $\sigma^2 \left\{ \log(p/s) + \sqrt{K \log(p/s)} \right\}$. Therefore, we have to choose an appropriate truncation level $K$ to balance the residual signal strength $\mu_K$ with this composite error bound, i.e., $\mu_K \asymp \sigma^2 \left\{ \log(p/s) + \sqrt{K \log(p/s)} \right\}$. Consequently, the high-dimensional structure affects the choice of truncation, revealing an interplay that is not only sufficient but also necessary.*

**Remark 2 (SpAM and GSM)** *The Gaussian sequence model (GSM, mentioned in Section 1.1) can be seen as a simplified SpAM, where $\theta_{1j} = 1$ and $\theta_{ij} = 0$ for each $i \geq 2$ and $j \in S_f$. Therefore, in GSM, we can just choose truncation $K \equiv 1$, and analyze the selection error caused by the Gaussian distribution [Butucea et al., 2018, Song and Cheng, 2023]. In contrast, to get an optimal truncation $K$ in general SpAM space, our selector (7) requires trading off the truncation bias against sub-exponential error. Both the analysis and outcome demonstrate that univariate function selection in SpAM is more challenging than variable selection in GSM.*

Additionally, our theoretical results can be extended to the following specific cases.

**Corollary 1** *Assume that all assumptions in Theorem 2 and Theorem 1 hold. Then we have:*

- **Sobolev**  *Take $\mu_i \asymp i^{-2\alpha}$ with smoothness parameter $\alpha$, the minimax separation rate for multiple testing is*

$$\epsilon_{test}^2 \asymp \sigma^2 \log(p/s) + \left( \sigma^4 \log(p/s) \right)^{\frac{2\alpha}{1+4\alpha}}.$$

- **Finite dimension**  *Take $\mu_1 = \cdots = \mu_m > \mu_{m+1} = \mu_{m+2} = \cdots = 0$ for some positive integer $m$, the minimax separation rate for multiple testing is*

$$\epsilon_{test}^2 \asymp \sigma^2 \log(p/s) + \left( \sigma^2 \sqrt{m \log(p/s)} \wedge \mu_1 \right).$$

- **Exponential decay**  *Take $\mu_i \asymp \exp(-c_1 i^\gamma)$, where $c_1$ is a positive constant and $\gamma > 0$, the minimax separation rate for multiple testing is*

$$\epsilon_{test}^2 \asymp \sigma^2 \log(p/s) + \sigma^2 \sqrt{\log(p/s)} \cdot \log^{\frac{1}{2\gamma}} \left( \frac{\sigma^{-4}}{\log(p/s)} \right).$$

**Remark 3 (Finite dimension case)** *We now give a further discussion of the finite dimension case. Under the assumption $\sigma^{-2} \gtrsim \frac{\log(p/s)}{\mu_1}$, the minimax separation rate exhibits two regimes:*

1. *If $\mu_1 \lesssim \sigma^2 \sqrt{m \log(p/s)}$, then we derive that $\frac{\sqrt{m \log(p/s)}}{\mu_1} \gtrsim \sigma^{-2} \gtrsim \frac{\log(p/s)}{\mu_1}$, leading $m \gtrsim \sigma^{-2} \mu_1$. In this case $\epsilon_{test}^2 \asymp \sigma^2 \log(p/s) + \mu_1 \asymp \mu_1$.*

2. *If $\mu_1 \succ \sigma^2 \sqrt{m \log(p/s)}$, then we get $\epsilon_{test}^2 \asymp \sigma^2 \log(p/s) + \sigma^2 \sqrt{m \log(p/s)}$, which aligns with the minimax separation rate in the group sparsity setting [Butucea et al., 2023a].*

*Combining these cases gives a more intuitive separation rate*

$$\epsilon_{test}^2 \asymp \min \left\{ \sigma^2 \left( \log(p/s) + \sqrt{m \log(p/s)} \right), \ \mu_1 \right\}.$$

*Here the minimum reflects our ellipsoid space constraint: by definition of $\mathcal{E}$ and $\mathcal{F}_s(r^2)$ in Section 2.1, every active univariate function $f_j$ obeys $\|f_j\|_2^2 \leq \mu_1 \sum_{i \in \mathbb{N}^+} \frac{\theta_{ij}^2}{\mu_i} \leq \mu_1$, leading its $L_2$ norm upper bounded by $\mu_1$.*

*In the case $\mu_1 \lesssim \sigma^2 \sqrt{m \log(p/s)}$, we have $\epsilon_{test}^2 \asymp \mu_1$. In other words, to control FDR+FNR, we would need each $f_j$ to satisfy $\|f\_j\|\_2 \geq C\mu_1$ for some large constant $C > 0$. However, each support $f_j$ obeys $\|f\_j\|_2 \leq \mu_1$. Hence, there are basically no $f_j$ that can attain a detectable norm, and the support recovery problem is essentially trivial in this case.*

### 3.2 From support recovery: wrong recovery probability control

**Preliminary setup** The univariate function selection can also be viewed as a support recovery problem. We measure the selection error between the estimated support $\hat{S}$ and the true support $S$ by using the Hamming loss $\mathbf{1}(\hat{\eta}(X) \neq \eta(f))$, where the probability of wrong recovery

$$\mathbf{P}_f \left( \hat{S}(X) \neq S(f) \right) = \mathbf{P}_f \left( \hat{\eta}(X) \neq \eta(f) \right) = \mathbf{E} \left( \mathbf{1} \left( \hat{\eta}(X) \neq \eta(f) \right) \right)$$

serves as the risk function, which characterizes how we can exactly recover the support set [Wainwright, 2007, Butucea et al., 2023a].

**Definition 2 (Minimax separation rate of support recovery)** *We say $\epsilon_{rec}^2$ is the non-asymptotic minimax separation rate of support recovery for (1) if:*

*(1) For all $\delta \in (0, 1)$, there exists $c_\delta > 0$ depending only on $\delta$ such that for all $0 < c < c_\delta$,*

$$\inf_{\hat{\eta}} \sup_{f \in \mathcal{F}_s(c\epsilon_{rec}^2)} \mathbf{P}_f \left( \hat{\eta}(X) \neq \eta(f) \right) \geq 1 - \delta.$$

*(2) For all $\delta \in (0, 1)$, there exists $C_\delta > 0$ depending only on $\delta$ such that for all $C > C_\delta$,*

$$\inf_{\hat{\eta}} \sup_{f \in \mathcal{F}_s(C\epsilon_{rec}^2)} \mathbf{P}_f \left( \hat{\eta}(X) \neq \eta(f) \right) \leq \delta,$$

*where $\inf_{\hat{\eta}}$ denotes the infimum over all selector $\hat{\eta}(X) : \mathbb{R}^{\mathbb{N}^+ \times p} \to \{0, 1\}^p$.*

Similar to Section 3.1, we next establish the minimax separation rate for the support recovery problem in SpAM (1).

**Theorem 3 (Minimax separation rate of support recovery)** *Let $\delta$ be an arbitrary number in $(0, 1)$, and assume that $\sigma^{-2} > \frac{C_{\delta,1} \log p}{\mu_1}$, $p \geq C_{\delta,2}$, and $s \geq C_{\delta,3}$. Then the minimax separation rate for the support recovery problem with respect to the wrong recovery probability $\mathbf{P}_f(\hat{\eta}(X) \neq \eta(f))$ is*

$$\epsilon_{rec}^2 \asymp \sigma^2 \log p + \max_{k \in \mathbb{N}^+} \left( \sigma^2 \sqrt{k \log p} \wedge \mu_k \right). \tag{10}$$

The minimax separation rate for support recovery in (10) is a little greater than that for sparse multiple testing in (9) ($\log p$ versus $\log(p/s)$), showing that controlling the wrong recovery probability is

more demanding than controlling the combined risk (FDR + FNR). Indeed, sparse testing requires $|\hat{S}\Delta S| = o(s)$, while exact recovery requires $|\hat{S}\Delta S| = o(1)$, necessitating a slightly stronger signal condition. In addition, this discrepancy leads to a higher thresholding level for optimal selection in support recovery, as detailed in the following.

**Remark 4 (Rate-optimal selector)** *Under assumptions in Theorem 3 and the signal condition*

$$r^2 \geq C_\delta \left\{ \sigma^2 \log p + \max_{k \in \mathbb{N}^+} \left( \sigma^2 \sqrt{k \log p} \wedge \mu_k \right) \right\},$$

*the selector*

$$\hat{\eta}_j^{rec}(X) = \mathbf{1} \left\{ \sum_{i=1}^{K'} X_{ij}^2 \geq \sigma^2 K' + 2\sigma^2 \left( \sqrt{K' \log(2p/\delta)} + \log(2p/\delta) \right) \right\}, \; j \in [p] \quad (11)$$

*controls the wrong recovery probability effectively:*

$$\sup_{f \in \mathcal{F}_s(r^2)} \mathbf{P}_f \left( \hat{\eta}^{rec}(X) \neq \eta(f) \right) \leq \delta,$$

*where* $K' := \min \left\{ k \in \mathbb{N}^+ : \mu_k \leq \sigma^2 \sqrt{k \log p} \right\}$ *and* $C_\delta > 0$ *is a constant only determined by* $\delta$.

**Remark 5 (Relation to existing work)** *For the Sobolev space with smoothness parameter* $\alpha$, *we rewrite the minimax separation rate* (10) *for exact support recovery as:*

$$\epsilon_{rec}^2 \asymp \begin{cases} \sigma^{\frac{8\alpha}{4\alpha+1}} (\log p)^{\frac{2\alpha}{4\alpha+1}} & \text{if } \log p \lesssim \sigma^{\frac{-2}{2\alpha+1}}, \\ \sigma^2 \log p & \text{if } \sigma^{-2} \gtrsim \log p \gtrsim \sigma^{\frac{-2}{2\alpha+1}}. \end{cases} \quad (12)$$

*Therefore, in the case* $\log p = o\left(\sigma^{\frac{-2}{2\alpha+1}}\right)$, *we match the rate derived from Ingster and Stepanova [2014], Butucea and Stepanova [2017]. Additionally, our findings establish the non-asymptotic minimax separation rate for the case* $\sigma^{-2} \gtrsim \log p > \sigma^{\frac{-2}{2\alpha+1}}$, *which was not provided in previous studies. In this case, the selection error exhibits sub-Gaussian behavior, resulting in the rate aligning with that in the Gaussian sequence model [Butucea et al., 2018, Song and Cheng, 2023].*

## 4 Adaptation to the smoothness

Thus far, our analysis has assumed full knowledge of the smoothness sequence $\{\mu_i\}_{i \in \mathbb{N}^+}$, which is often unrealistic. This section investigates how adaptation to unknown smoothness affects the minimax separation rate. For simplicity, we consider the Sobolev space with $\mu_i = i^{-2\alpha}$, $\alpha > 0$, and rewrite the original space $\mathcal{F}_s(r^2)$ as $\mathcal{F}_s(r^2, \alpha)$. The wrong recovery probability $\mathbf{P}_f(\hat{\eta}(X) \neq \eta(f))$ is used as the risk function.

**A selector adaptive to the unknown** $\alpha$ Define the truncation set

$$\mathcal{K}_{rec} := \left\{ 2, 4, \cdots, 2^{\left\lceil \log_2\left(\frac{\sigma^{-4}}{\log p}\right) \right\rceil} \right\}.$$

For every $\delta \in (0, 1)$ and $k \in \mathcal{K}_{rec}$, we denote $\hat{\eta}^{(k)}(X) := \left( \hat{\eta}_1^{(k)}(X), \cdots, \hat{\eta}_p^{(k)}(X) \right) \in \{0, 1\}^p$ as the selector vector with respect to $k$, where

$$\hat{\eta}_j^{(k)} := \mathbf{1} \left\{ \sigma^{-2} \sum_{i=1}^{k} X_{ij}^2 \geq k + 2\sqrt{k \log\left(\frac{8p \log(\sigma^{-2})}{\delta}\right)} + 2\log\left(\frac{8p \log(\sigma^{-2})}{\delta}\right) \right\}.$$

Now, we define the adaptive selector

$$\hat{\eta}^{ad}(X) := \left( \max_{k \in \mathcal{K}_{rec}} \hat{\eta}_1^{(k)}, \cdots, \max_{k \in \mathcal{K}_{rec}} \hat{\eta}_p^{(k)} \right) \in \{0, 1\}^p. \quad (13)$$

For each $f_j$, our selector (13) firstly constructs individual tests for each $k \in \mathcal{K}_{rec}$, and then aggregates them by taking the maximum over $\mathcal{K}_{rec}$. Equivalently, $f_j$ is declared supported as soon as it is identified as nonzero under any candidate $k \in \mathcal{K}_{rec}$; conversely, $f_j$ is declared non-supported only if it is identified as zero for all $k \in \mathcal{K}_{rec}$. We next establish the sufficient signal condition for the wrong recovery probability control.

**Theorem 4 (Upper bound for adaptation)** *Let $\delta$ be an arbitrary number in $(0,1)$, and assume that $\sigma^{-2} > \frac{C_{\delta,1} \log p}{\mu_1}$. Then, for all $r^2$ satisfies*

$$r^2 \geq \left(12\sqrt{2}+1\right) \sigma^{\frac{8\alpha}{1+4\alpha}} \log^{\frac{2\alpha}{1+4\alpha}} \left(\frac{8p\log(\sigma^{-2})}{\delta}\right) + 18\sigma^2 \log\left(\frac{8p\log(\sigma^{-2})}{\delta}\right),$$

*we have*

$$\sup_{\alpha>0} \sup_{f\in\mathcal{F}_s(r^2,\alpha)} \mathbf{P}_f\left(\hat{\eta}^{ad}(X) \neq \eta\right) \leq \delta.$$

Compared to (12), an additional $\log\left(\log(\sigma^{-2})\right)$ term in the signal strength condition is required. The following theorem shows that $\log\left(\log(\sigma^{-2})\right)$ is also necessary for the adaptation to the smoothness.

**Theorem 5 (Lower bound for adaptation)** *Let $\delta$ be an arbitrary number in $(0,1)$, and assume that $\sigma^{-2} > \frac{C_{\delta,1} \log p}{\mu_1}$ and $p \geq C_{\delta,2}$. Then, for all $r^2$ satisfies*

$$0 < r^2 \leq c_{\delta,3} \left\{ \sigma^{\frac{8\alpha}{1+4\alpha}} \log^{\frac{2\alpha}{1+4\alpha}} \left(p\log(\sigma^{-2})\right) + \sigma^2 \log\left(p\log(\sigma^{-2})\right) \right\},$$

*we have*

$$\inf_{\hat{\eta}} \sup_{\alpha>0} \sup_{f\in\mathcal{F}_s(r^2,\alpha)} \mathbf{P}_f\left(\hat{\eta}(X) \neq \eta\right) \geq 1-\delta.$$

Theorem 4 and 5 establish the adaptive minimax separation rate as

$$\sigma^{\frac{8\alpha}{1+4\alpha}} \log^{\frac{2\alpha}{1+4\alpha}} \left(p\log(\sigma^{-2})\right) + \sigma^2 \log\left(p\log(\sigma^{-2})\right). \tag{14}$$

In the high-dimensional case $\log p \gtrsim \log\left(\log(\sigma^{-2})\right)$, the $\log\left(\log(\sigma^{-2})\right)$ term becomes negligible and (14) achieves the same rate as (12), indicating that adaptation incurs no additional cost on the rate. However, when $p$ is a large constant that is much smaller than $\sigma^{-2}$, (14) indicates that, with the smoothness unknown, achieving support recovery requires a stronger signal strength compared to (12).

## 5 Discussion: difference between optimal estimation and selection

We finally end this paper by discussing the difference between estimation and selection. For simplicity, we assume $s \leq p^{1-\beta}$, where $\beta \in (0,1)$ is a constant, therefore $\log(ep/s) \asymp \log p$. Next, we establish a minimax-optimal estimator for $f \in \mathcal{F}_s$ through a truncated hard-thresholding procedure:

$$\hat{\theta}_{ij} = X_{ij} \cdot \underbrace{\mathbf{1}(i \leq K_e)}_{\text{Truncation}} \cdot \underbrace{\mathbf{1}\left(\sigma^{-2} \sum_{i'=1}^{K_e} X_{ij}^2 \geq K_e + \sqrt{CK_e \log p} + C\log p\right)}_{\text{Hard thresholding}}, \tag{15}$$

where we define $K_e := \min\{k \in \mathbb{N}^+ : \mu_k \leq \sigma^2 k\}$, and $C > 0$ is a fixed constant.

**Theorem 6 (Optimal truncation for function estimation)** *Assume $\sigma^{-2} \geq \frac{C_1 \log p}{\mu_1}$ and the canstant $C$ in (15) satisfies $C \geq 4$. Then the estimator (15) is rate-optimal:*

$$\sup_{f\in\mathcal{F}_s} \mathbf{E}_f \|\hat{f}(\hat{\Theta}) - f\|_2^2 = \sup_{f\in\mathcal{F}_s} \mathbf{E}_f \|\hat{\Theta} - \Theta(f)\|_2^2 \lesssim \sigma^2 s\log p + s \times \max_{k\in\mathbb{N}+} \left((\sigma^2 k) \wedge \mu_k\right).$$

Combined with (4), Theorem 6 implies that by only using the first $K_e$ entries in each observation sequence (i.e., only using $X_{[K_e]\times[p]} := \{X_{ij}\}_{1\leq i\leq K_e, 1\leq j\leq p}$), one can achieve a minimax optimal function estimation. However, it may fail to guarantee optimal univariate function selection by only using these truncated observations, as shown below.

**Theorem 7 (Suboptimal selection)** *Assume $\sigma^{-2} \geq \frac{C_1 \log p}{\mu_1}$ and $p \geq C_2$. Then, for all $r^2$ satisfies*

$$0 < r^2 \leq c_3 \left\{ \sigma^2 \log p + \max_{k\in[K_e]} \left(\sigma^2\sqrt{k\log p} \wedge \mu_k\right) + \mu_{K_e+1} \right\}, \tag{16}$$

*we have a lower bound as*

$$\inf_{\hat{\eta}(X_{[K_e]\times[p]})\in\{0,1\}^p} \sup_{f\in\mathcal{F}_s(r^2)} \mathbf{P}_f\left(\hat{\eta}(X_{[K_e]\times[p]})\neq\eta\right) \geq \frac{1}{2},$$

*where the infimum* $\inf_{\hat{\eta}(X_{[K_e]\times[p]})\in\{0,1\}^p}$ *takes over all restricted selectors that only use observations* $X_{[K_e]\times[p]}$, *and* $C_1, C_2, c_3 > 0$ *are absolute constants.*

This theorem demonstrates that in the family of truncation estimators, optimal estimation sometimes leads to a suboptimal univariate function selection. For example, consider the Sobolev space with $\mu_i = i^{-2\alpha}$ in the case $\log p = o\left(\sigma^{-\frac{2}{1+2\alpha}}\right)$. To exactly recover the support set, the necessary signal strength (16) (by only using $X_{[K_e]\times[p]}$) is of the rate $\sigma^{\frac{4\alpha}{1+2\alpha}}$, which exceeds the minimax separation rate $\sigma^{\frac{8\alpha}{4\alpha+1}}(\log p)^{\frac{2\alpha}{4\alpha+1}}$, as illustrated in (12). This gap directly shows that optimal univariate function selection cannot be treated as a byproduct of optimal SpAM function estimation. See Figure 1 for a clearer difference.

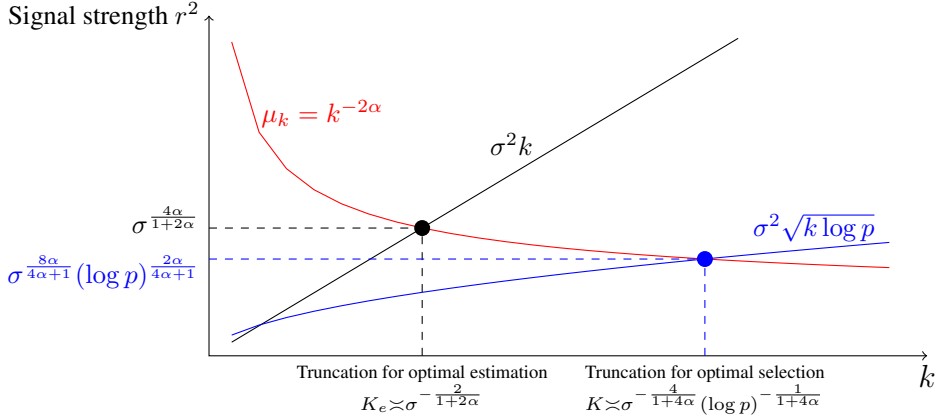

Figure 1: The difference between optimal estimation and selection in the case $\log p = o\left(\sigma^{-\frac{2}{1+2\alpha}}\right)$.

Appendix A discusses some future directions for this paper, and Appendix B provides the numerical experiment to confirm our theoretical findings.

## Acknowledgments and Disclosure of Funding

This work was partially supported by the Academic Scholarship Program of Renmin University of China. The author thanks Cristina Butucea, Jianxin Yin, and Zhifan Li for helpful discussions.

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

# A    Limitations and future directions

**Besov ball or $L_q$ ball**    One key direction for future work is to extend our univariate function selection results from the $L_2$-ellipsoid (2) to richer nonparametric classes such as Besov balls $B_{r,q}^\sigma$ or, more generally, $L_q$-ellipsoids. These spaces naturally align with wavelet bases and are foundational to practical methods in signal processing and denoising. However, under the high-dimensional setting, the techniques in Baraud [2002] may not be useful anymore. Perhaps a more viable approach is to construct selectors based on the nonquadratic estimation procedure in Cai and Low [2005, 2006], which may also lead to a minimax adaptation result simultaneously.

**Univariate function selection under local differential privacy**    Integrating the differential privacy (DP) mechanism into univariate function selection for SpAM represents a direction for future research. DP ensures rigorous protection of individual data while allowing valid statistical inference; therefore is welcomed by the computer science, machine learning, and statistics communities recently. In the local DP setting, Butucea et al. [2023b] established phase transitions for support recovery in the sparse mean model, deriving minimax separation rates for exact recovery and for almost-full recovery. Butucea et al. [2020, 2023c] studied the function estimation and the quadratic functional estimation in the nonparametric univariate function, respectively, where the latter plays an important role in goodness-of-fit testing. All these works demonstrated that DP leads to some markedly different minimax rates compared to non-private benchmarks.

Consequently, when extending univariate function selection in SpAM to local DP constraints, one should expect that the minimax separation rates will differ from the results in this paper: the optimal truncation should be recalibrated to account for the additional privacy-induced noise. Designing and analyzing such privacy-preserving selectors for SpAMs remains an important and challenging problem.

**A general conclusion about estimation and selection**    Another significant extension lies in generalizing the minimax lower bound in Theorem 7, which currently restricts the infimum to selectors relying solely on truncated observations $X_{[K_e]\times[p]}$. To this end, we define the minimax optimal estimation class

$$E_{opt} := \left\{ \hat{f} : \mathbb{R}^{\mathbb{N}^+ \times [p]} \to \mathbb{R}^{\mathbb{N}^+ \times [p]} \;\middle|\; \sup_{f \in \mathcal{F}_s} \mathbf{E}_f \left\| \hat{f}(X) - f \right\|_2^2 \lesssim \sigma^2 s \log p + s \times \max_{k \in \mathbb{N}^+} \left( (\sigma^2 k) \wedge \mu_k \right) \right\}.$$

The general version of Theorem 7 should focus on the necessary signal condition for selectors induced by estimation class $E_{opt}$:

$$\inf_{\hat{f} \in E_{opt}} \inf_{\hat{\eta} = \hat{\eta}(\hat{f})} \sup_{f \in \mathcal{F}_s(r^2)} \mathbf{P}_f \left\{ \hat{\eta}(\hat{f}) \neq \eta(f) \right\} \geq c.$$

Ideally, this lower bound could quantify how the minimax optimal estimations perform in the support recovery problem. It could also lead to a more comprehensive realization of the difference between estimation and selection.

Establishing such a result will likely require some new analytic tools, and we think the techniques in Song and Cheng [2023] may give some help. We leave this interesting problem for future research.

# B    Numerical experiment

We conduct three simulation studies to evaluate the performance of our truncated-type selectors in sparse additive models. For ease of display, we define $n = \sigma^{-2}$.

1. Compare the performance of our proposed method across varying dimension $p$ and signal strength $r^2$.
2. Compare the performance of different selection methods across varying variance $1/n$.

3. Compare the performance of different selection methods across varying smoothness parameters.

In all experiments, we take $\mathcal{X} = [0, 1]$, $s = 5$, and let the support covariates be $j = 1, \dots, 5$, with centered functions

$$
\begin{aligned}
f_1(x) &= x^2\big(2^{x-1} - (x - 0.5)^2\big)e^x - 0.5424, \\
f_2(x) &= 12(x - 0.5)^2 - 12, \\
f_3(x) &= 3x^2\, 2^{x-1} \cos(15x) - 0.1002, \\
f_4(x) &= 2x - 1, \\
f_5(x) &= 8(x - 0.7)^3 + 0.4640,
\end{aligned}
$$

which all belong to the Sobolev space with $\alpha = 1/2$.

Performance is measured by the Hamming loss

$$
\mathbf{1}\big(\hat{\eta}(X) \neq \eta(f)\big),
$$

and the combined FDR plus FNR loss

$$
\frac{\sum_{j \notin S_f} \hat{\eta}_j}{1 \vee \sum_{j \in [p]} \hat{\eta}_j} + \frac{\sum_{j \in S_f} (1 - \hat{\eta}_j)}{s}.
$$

For each simulation, we execute 300 repetitions, with 1-sigma error bars provided in the figures. All simulations are conducted using R and executed on a personal laptop equipped with an AMD Ryzen 7 5800H processor operating at 3.20 GHz and 16.00GB of RAM.

## B.1 Simulation 1: dimension and signal strength

We fix $n = 300$, and vary $p \in \{10, 100, 1000, 10000\}$. We take $a \cdot f_j$ as the support function, for $j = 1, \cdots, 5$, where $a > 0$ quantifies the effect of the signal strength.

Figure 2 shows that, as the signal strength $a$ increases, the selection errors (both Hamming loss and FDR plus FNR loss) for each $p$ decay toward a relatively low level, but larger $p$ demands higher $a$ to reach the same error level. Moreover, controlling FDR + FNR requires weaker signal strengths: at $p = 10000$, $a = 0.5$ suffices to keep FDR + FNR = 0.5, whereas the Hamming loss drops below 0.5 until $a = 0.7$. This behavior reflects the fundamental difference between sparse multiple testing and exact support recovery, as we discussed after Theorem 3.

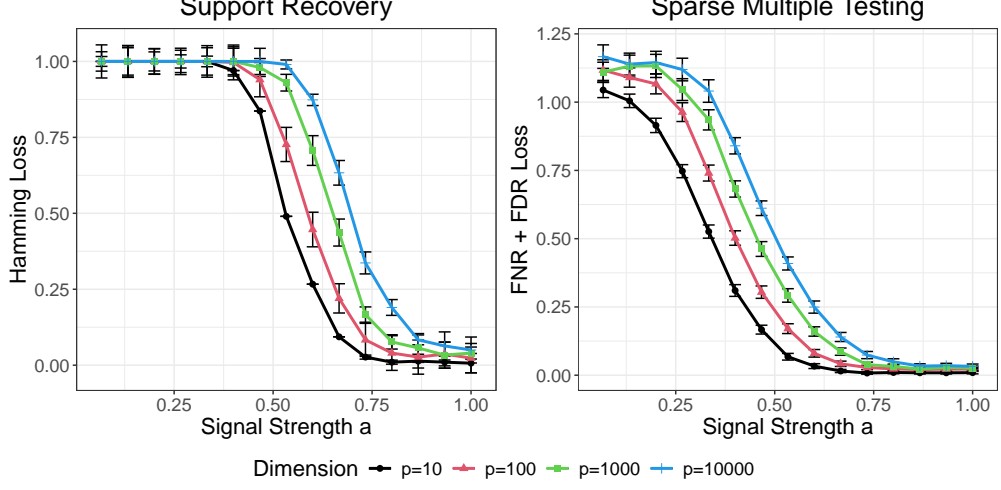

Figure 2: Selection performance with different dimensions and signal strength.

## B.2 Simulation 2: noise variance $1/n$

We fix $p = 500$ and vary $n$ from 20 to 300. Four types of selectors are considered in this simulation:

1. **Optimal**   The rate-optimal selector (11).

2. **Adaptation**   The adaptive selector (13).

3. **Univariate**   The selector that takes truncation at $K_u = \min\left\{k \in \mathbb{N}^+ : \mu_k \leq \frac{\sqrt{k}}{n}\right\}$.

4. **Suboptimal**   The selector that takes truncation at $K_e = \min\left\{k \in \mathbb{N}^+ : \mu_k \leq \frac{k}{n}\right\}$.

Figure 3 illustrates that, as $n$ grows, all methods see error decay, but the Optimal and Adaptation methods maintain the lowest selection errors across most regimes. Additionally, as we discussed in Remark 5, for $n \lesssim (\log p)^{1+2\alpha}$, the minimax separation rate is $\log p/n$, under which the $K_e$-truncation remains rate-optimal, giving the Suboptimal selector a temporary advantage (for $n < 100$). Once $n \gtrsim (\log p)^{1+2\alpha}$, the minimax separation rate becomes $n^{-\frac{4\alpha}{4\alpha+1}}(\log p)^{\frac{2\alpha}{4\alpha+1}}$, and truncation at $K_e$ cannot be optimal anymore.

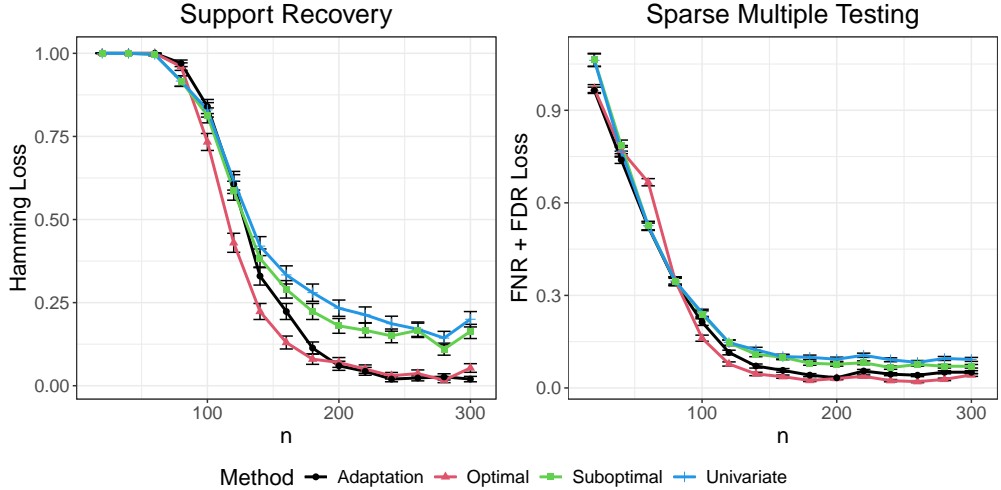

Figure 3: Selection performance of the four methods with different noise level $n$.

## B.3 Simulation 3: smoothness parameter $\beta$

We fix $p = 500$ and $n = 300$, and assess the effect of smoothness on univariate function selection. First, for $j = 1, \ldots, 5$, we compute the original basis coefficients of each $f_j$, denoted by $\{\theta_{ij}\}_{i \in \mathbb{N}^+}$. We next reweight these coefficients and get the new functions

$$f_j^{(\beta)} = \sum_{i \in \mathbb{N}^+} i^{\frac{1}{2}-\beta}\theta_{ij}\psi_i, \quad j = 1, \cdots, 5,$$

so that each $f_j^{(\beta)}$ lies in the Sobolev ball with smoothness parameter $\beta$. We vary $\beta \in [0.2, 1]$ and compare the performance of the four methods.

As shown in Figure 4, only the Adaptation method consistently achieves low error across all $\beta$, demonstrating its optimality and robustness to unknown smoothness and verifying our theoretical guarantees in Section 4.

## C   Proof of Theorem 3

We first introduce the proof of the lower bound and upper bound in Theorem 3. These proofs are instructive and lead to clearer proofs of Theorem 1 and Theorem 2.

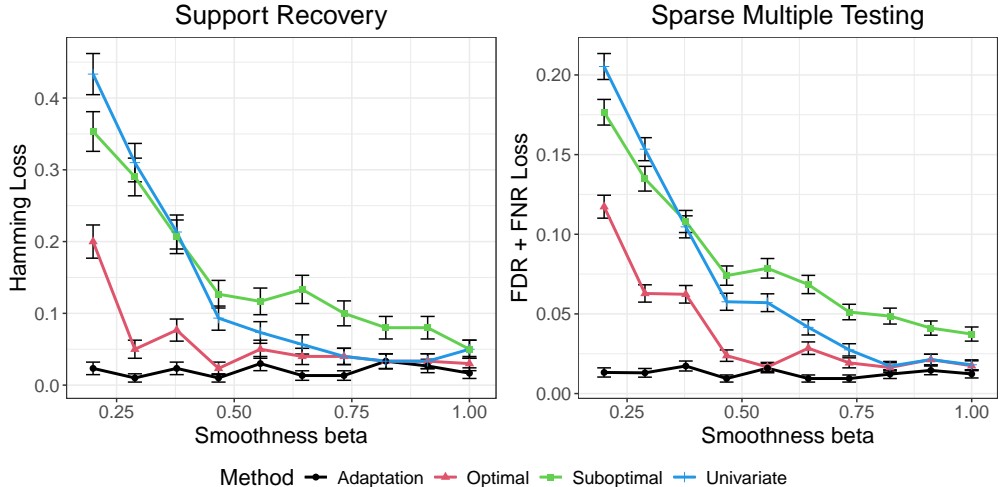

Figure 4: Selection performance of the four methods with different smoothness parameter $\beta$.

We first define the Hamming distance

$$H(\hat{\eta}(X), \eta(f)) := \sum_{j=1}^{p} |\hat{\eta}_j - \eta_j| = \sum_{j=1}^{p} \mathbf{1}(\hat{\eta}_j \neq \eta_j),$$

which can upper-bound the wrong classification probability

$$\mathbf{E}_f\big(H(\hat{\eta}, \eta)\big) = \sum_{w=1}^{p} w \mathbf{P}_f\big(H(\hat{\eta}, \eta) = w\big) \geq \mathbf{P}_f\big(\hat{\eta} \neq \eta\big). \tag{17}$$

For notational convenience, throughout all proofs we set $n = \sigma^{-2}$ to represent the noise intensity.

### C.1 The lower bound

To better clarify the truncation construction, we define

$$\mathcal{E}^{(k)}(r^2) := \left\{ \theta \in \mathbb{R}^{\mathbb{N}^+} : \theta_i = \frac{r}{\sqrt{k}} \text{ for all } 1 \leq i \leq k, \ \theta_i = 0 \text{ for all } i > k \right\} \subset \mathcal{E}.$$

The SPAM function set induced by $\mathcal{E}^{(k)}(r^2)$ is:

$$\mathcal{F}_s^{(k)}(r^2) := \left\{ f = \sum_{j=1}^{p} f_j \in \mathcal{F}_s(r^2) : f_j \in \mathcal{E}^{(k)}(r^2) \text{ for all } f_j \neq 0 \right\}.$$

Now we consider:

$$\inf_{\hat{\eta}:\mathbb{R}^{\mathbb{N}^+ \times p} \to \{0,1\}^p} \sup_{f \in \mathcal{F}_s(r^2)} \mathbf{P}_f\Big(\hat{\eta}(X) \neq \eta(f)\Big) \geq \inf_{\hat{\eta}:\mathbb{R}^{\mathbb{N}^+ \times p} \to \{0,1\}^p} \sup_{f \in \mathcal{F}_s^{(k)}(r^2)} \mathbf{P}_f\Big(\hat{\eta}(X) \neq \eta(f)\Big)$$

$$= \inf_{\hat{\eta}:\mathbb{R}^{k \times p} \to \{0,1\}^p} \sup_{f \in \mathcal{F}_s^{(k)}(r^2)} \mathbf{P}_f\Big(\hat{\eta}(X_{k \times p}) \neq \eta(f)\Big), \tag{18}$$

which means that in $\mathcal{F}_s^{(k)}(r^2)$, we only need to consider those selectors $\hat{\eta}$ based on the first $k$ observations in each univariate function $f_j, j \in [p]$.

Now, for some fixed $k \in \mathbb{N}^+$, we set a least favorable subset of $\mathcal{F}_s^{(k)}(r^2)$, and then derive its lower bound of the minimax separation rate.

### C.1.1 The least favorable subset

For every fixed $\delta \in (0,1)$, $k \in \mathbb{N}^+$ which satisfy $\mu_k \geq \left(\frac{c_1\delta}{25} \wedge 1\right) \max\left(\frac{\log(p-s)}{n}, \frac{\sqrt{k\log(p-s)}}{n}\right)$ (where $c_1 > 0$ is a constant defined in Lemma 2), consider the subset:

$$\tilde{\mathcal{F}}_s^{(k)}(r^2) := \left\{ \Theta \in \mathcal{F}_s^{(k)}(r^2) : \sum_{j=1}^{p} \mathbf{1}(\theta_{\cdot j} \neq \mathbf{0}) = s \right\}. \tag{19}$$

Therefore, for each $f \in \tilde{\mathcal{F}}_s^{(k)}(r^2)$, if its $j$-th univariate function $f_j \equiv 0$, the random variable $n \sum_{i=1}^{k} X_{ij}^2$ follows from a central $\chi^2$-distribution with $k$ degrees of freedom (note that $X_{ij} \sim N(\theta_{ij}, 1/n)$). If $f_j \neq 0$, $n \sum_{i=1}^{k} X_{ij}^2$ follows from a non-central $\chi^2$-distribution with $k$ degrees of freedom and with non-centrality parameter $nr^2$. Let $f_0$ and $f_1$ be the densities of these two distributions with respect to the Lebesgue measure:

$$f_0(z) = \frac{z^{k/2-1} \, e^{-z/2}}{2^{k/2} \, \Gamma(k/2)}, \ z > 0,$$

$$f_1(z) = \left(\frac{1}{2}\right)^{k/2} e^{-nr^2/2} \sum_{i=0}^{\infty} \frac{\left(\frac{nr^2}{4}\right)^i z^{k/2+i-1} \, e^{-z/2}}{i! \, \Gamma(k/2+i)}, \ z > 0. \tag{20}$$

Once the positive integer $k$ is fixed, by Lemma 1 we only need to consider the selector based on the norm $\|X_{1:k,j}\|_2$, which we call them the **norm selectors**. Then we conclude

$$\inf_{\hat{\eta}:\mathbb{R}^{k \times p} \to \{0,1\}^p} \sup_{f \in \mathcal{F}_s^{(k)}(r^2)} \mathbf{P}_f\left(\hat{\eta}(X_{k \times p}) \neq \eta(f)\right)$$

$$\geq \inf_{\hat{\eta}:\mathbb{R}^{k \times p} \to \{0,1\}^p} \sup_{f \in \tilde{\mathcal{F}}_s^{(k)}(r^2)} \mathbf{P}_f\left(\hat{\eta}(X_{k \times p}) \neq \eta(f)\right)$$

$$\overset{(i)}{\geq} \inf_{\hat{\eta}: \text{ norm selector}} \sup_{f \in \tilde{\mathcal{F}}_s^{(k)}(r^2)} \mathbf{P}_f\left(\hat{\eta}\left(\|X_{1:k,1}\|_2, \cdots, \|X_{1:k,p}\|_2\right) \neq \eta(f)\right) \tag{21}$$

$$\overset{(ii)}{\geq} \mathbf{P}_{e(s)}\left(\min_{j=1,\cdots,s} \frac{f_1}{f_0}\left(n\|X_{1:k,j}\|_2^2\right) \leq \max_{j=s+1,\cdots,p} \frac{f_1}{f_0}\left(n\|X_{1:k,j}\|_2^2\right)\right)$$

$$\overset{(iii)}{=} \mathbf{P}_{e(s)}\left(\min_{j=1,\cdots,s} n\|X_{1:k,j}\|_2^2 \leq \max_{j=s+1,\cdots,p} n\|X_{1:k,j}\|_2^2\right),$$

where inequality (i) follows from Lemma 1, inequality (ii) follows from Theorem 6 in Butucea et al. [2023a], where we denote by $\mathbf{P}_{e(s)}$ a probability measure in which only the first $s$ univariate functions are non-zero, i.e., $f_j = 0 \Leftrightarrow j \notin [s]$. Equality (iii) follows from the monotonic increasing property of the likelihood ratio $\frac{f_1}{f_0}(z)$ on $z \in \mathbb{R}^+$.

### C.1.2 The tail probabilities

With the fixed $\delta \in (0,1)$, $k \in \mathbb{N}^+$ which satisfy $\mu_k \geq \left(\frac{c_1\delta}{25} \wedge 1\right) \max\left(\frac{\log(p-s)}{n}, \frac{\sqrt{k\log(p-s)}}{n}\right)$, we aim to prove that the last probability in (21) is greater than $1-\delta$ if $\log(p-s) \geq \left(\frac{16}{c_1^2} + \log\frac{\log(2/\delta)}{c_2}\right) \vee \left(2\log\frac{\log(2/\delta)}{c_2}\right) \vee \frac{8}{c_1}$ and $r^2 \leq \left(\frac{c_1\delta}{25} \wedge 1\right) \max\left(\frac{\log(p-s)}{n}, \frac{\sqrt{K\log(p-s)}}{n}\right)$, where $c_1 > 0$ and $c_2 \in (0,1)$ are two positive constants defined in Lemma 2. Firstly, by taking $x = \log\frac{c_2(p-s)}{\log(2/\delta)} > 0$, we

conclude

$$\mathbf{P}_{e(s)}\left(\max_{j=s+1,\cdots,p} n\|X_{1:k,j}\|_2^2 \geq k + c_1 x + c_1\sqrt{kx}\right) = 1 - \left\{1 - \mathbf{P}\left(\chi_k^2(0) \geq k + c_1 x + c_1\sqrt{kx}\right)\right\}^{p-s}$$

$$\overset{(i)}{\geq} 1 - \left(1 - c_2 e^{-x}\right)^{p-s}$$

$$= 1 - \left(1 - \frac{\log(2/\delta)}{p-s}\right)^{p-s}$$

$$\overset{(ii)}{\geq} 1 - \frac{\delta}{2},$$

$$(22)$$

where inequality (i) follows from (59) in Lemma 2, inequality (ii) follows from the assumption $p - s \geq \left(\frac{\log(2/\delta)}{c_2}\right)^2 > \frac{\log(2/\delta)}{c_2} > 1$.

Besides, by taking $r^2 \leq \frac{c_1\delta}{25}\max\left(\frac{\log(p-s)}{n}, \frac{\sqrt{k\log(p-s)}}{n}\right)$ and $\log(p-s) \geq 2\log\frac{\log(2/\delta)}{c_2}$, we conclude $x \geq \frac{1}{2}\log(p-s) > \frac{\delta}{2}\log(p-s)$ and

$$nr^2 + 2\sqrt{2nr^2} < \frac{2c_1}{25}\left(x + \sqrt{kx}\right) + 2\sqrt{2 \cdot \frac{2c_1}{25}\left(x + \sqrt{kx}\right)} < \frac{c_1}{2}\left(x + \sqrt{kx}\right), \quad (23)$$

where the last inequality follows from $c_1\left(x + \sqrt{kx}\right) \geq 4$ led by $\log(p-s) \geq 8/c_1$. Then, by assuming $s \geq \log(2/\delta)$ and $\log(p-s) \geq \frac{16}{c_1^2} + \log\frac{\log(2/\delta)}{c_2}$, we conclude $x \geq 16/c_1^2$, therefore

$$nr^2 + 2\sqrt{(k + 2nr^2)\frac{\log(2/\delta)}{s}} + \frac{2\log(2/\delta)}{s} \leq nr^2 + 2\sqrt{k + 2nr^2} + 2$$

$$\leq nr^2 + 2\sqrt{2nr^2} + 2\left(\sqrt{k} + 1\right)$$

$$\overset{(i)}{<} \frac{c_1}{2}\left(x + \sqrt{kx}\right) + \frac{c_1}{2}\sqrt{x}\left(\sqrt{k} + 1\right)$$

$$\leq c_1\left(x + \sqrt{kx}\right),$$

where inequality (i) follows from (23). Therefore, we conclude

$$\mathbf{P}_{e(s)}\left(\min_{j=1,\cdots,s} n\|X_{1:k,j}\|_2^2 \leq k + c_1 x + c_1\sqrt{kx}\right)$$

$$= 1 - \left\{\mathbf{P}\left(\chi_k^2(nr^2) \geq k + c_1 x + c_1\sqrt{kx}\right)\right\}^s$$

$$\geq 1 - \left\{\mathbf{P}\left(\chi_k^2(nr^2) \geq k + nr^2 + 2\sqrt{(k + 2nr^2)\frac{\log(2/\delta)}{s}} + \frac{2\log(2/\delta)}{s}\right)\right\}^s \quad (24)$$

$$\overset{(i)}{\geq} 1 - \left(\exp\left(-\frac{\log(2/\delta)}{s}\right)\right)^s = 1 - \frac{\delta}{2},$$

where inequality (i) follows from (61) in Lemma 2 with non-centrality parameter $B = nr^2$. Combining (21), (22) and (24), we conclude that

$$\inf_{\hat{\eta}} \sup_{f\in\mathcal{F}_s(r^2)} \mathbf{P}_f\left(\hat{\eta}(X) \neq \eta(f)\right) \geq \mathbf{P}_{e(s)}\left(\min_{j=1,\cdots,s}\|X_{1:k,j}\|_2^2 \leq \max_{j=s+1,\cdots,p}\|X_{1:k,j}\|_2^2\right)$$

$$\geq \mathbf{P}_{e(s)}\left(\min_{j=1,\cdots,s} n\|X_{1:k,j}\|_2^2 \leq k + c_1 x + c_1\sqrt{kx}\right)$$

$$\times \mathbf{P}_{e(s)}\left(\max_{j=s+1,\cdots,p} n\|X_{1:k,j}\|_2^2 \geq k + c_1 x + c_1\sqrt{kx}\right)$$

$$\geq \left(1 - \frac{\delta}{2}\right)^2 > 1 - \delta.$$

### C.1.3 The optimal truncation K

By the definition of $\tilde{\mathcal{F}}_s^{(k)}(r^2)$, $\mathcal{F}_s^{(k)}(r^2)$ and $\mathcal{U}^{(k)}(r^2)$, the minimax separation rate is lower bounded by the constrained maximum:

$$
\begin{aligned}
\max : &\; c_\delta \max\left(\frac{\log(p-s)}{n}, \frac{\sqrt{k\log(p-s)}}{n}\right), \\
\text{subject to} : &\; c_\delta \max\left(\frac{\log(p-s)}{n}, \frac{\sqrt{k\log(p-s)}}{n}\right) \leq \mu_k, \\
&\; k \in \mathbb{N}^+,
\end{aligned}
\tag{25}
$$

where $c_\delta = \frac{c_1\delta}{25} \wedge 1 \in (0, 1]$. For ease of display, we define

$$
\begin{aligned}
K^{(c_\delta)} &:= \min\left\{k \in \mathbb{N}^+ : \mu_k \leq \frac{c_\delta\sqrt{k\log(p-s)}}{n}\right\} \\
L^{(c_\delta)} &:= \max\left\{k \in \mathbb{N}^+ : \mu_k \geq \frac{c_\delta\sqrt{k\log(p-s)}}{n}\right\},
\end{aligned}
\tag{26}
$$

By assuming $n > \frac{c_\delta \log(p-s)}{\mu_1}$, we derive that $1 \leq L^{(c_\delta)} \leq K^{(c_\delta)} \leq L^{(c_\delta)} + 1$. Then we analyze the maximum into two cases:

- **Case A: When** $\mu_{\lceil\log(p-s)\rceil} \geq \frac{c_\delta\sqrt{\lceil\log(p-s)\rceil\log(p-s)}}{n}$. We derive that $L^{(c_\delta)} \geq \lceil\log(p-s)\rceil$ hence $\frac{c_\delta\sqrt{L^{(c_\delta)}\log(p-s)}}{n} \geq \frac{c_\delta\log(p-s)}{n}$. Then the maximum of (25) is $\frac{c_\delta\sqrt{L^{(c_\delta)}\log(p-s)}}{n}$.

- **Case B: When** $\mu_{\lceil\log(p-s)\rceil} < \frac{c_\delta\sqrt{\lceil\log(p-s)\rceil\log(p-s)}}{n}$. We derive that $1 \leq L^{(c_\delta)} \leq \lfloor\log(p-s)\rfloor$ hence $\frac{c_\delta\sqrt{L^{(c_\delta)}\log(p-s)}}{n} \leq \frac{c_\delta\log(p-s)}{n}$. Then the maximum of (25) is $\frac{c_\delta\log(p-s)}{n}$.

Therefore, we establish the lower bound of the minimax separation rate as:

$$
\begin{aligned}
c_\delta \cdot \max\left\{\frac{\log(p-s)}{n}, \frac{\sqrt{L^{(c_\delta)}\log(p-s)}}{n}\right\} &\stackrel{(i)}{\asymp} \max\left\{\frac{\log(p-s)}{n}, \frac{\sqrt{K^{(c_\delta)}\log(p-s)}}{n}\right\} \\
&\stackrel{(ii)}{\asymp} \max\left\{\frac{\log(p-s)}{n}, \max_{k\in\mathbb{N}^+}\left(\mu_k \wedge \frac{\sqrt{k\log(p-s)}}{n}\right)\right\} \\
&\asymp \frac{\log(p-s)}{n} + \max_{k\in\mathbb{N}^+}\left(\mu_k \wedge \frac{\sqrt{k\log(p-s)}}{n}\right),
\end{aligned}
\tag{27}
$$

where equality (i) follows from $1 \leq L^{(c_\delta)} \leq K^{(c_\delta)} \leq L^{(c_\delta)} + 1$ and equality (ii) follows from Lemma 4. By (27), we derive the lower bound of the minimax separation rate.

### C.2 The upper bound

By (17), we only need to prove that $\sup_{f\in\mathcal{F}_s(r^2)} \mathbf{E}_f\left(H(\hat{\eta}(X), \eta(f))\right) \leq \delta$. For ease of display, we denote $\lambda^2(K') = \frac{2}{n}\left(\sqrt{K'\log(2p/\delta)} + \log(2p/\delta)\right)$.

### C.2.1 Preliminary

For a fixed SPAM $f \in \mathcal{F}_s(r^2)$, we use $S_f \subset [p]$ as the index set of the support univariate functions $f_j \neq 0$. Then we have

$$
\begin{aligned}
\mathbf{E}_f\Big(H(\hat{\eta}(X), \eta(f))\Big) =& \mathbf{E}_f \sum_{j=1}^{p} \mathbf{1}\Big(\hat{\eta}_j(X) \neq \eta_j(f_j)\Big) \\
=& \sum_{j=1}^{p} \mathbf{P}_f\Big(\hat{\eta}_j(X) \neq \eta_j(f_j)\Big) \\
=& \sum_{j \in S_f} \mathbf{P}_{f_j}\left(\sum_{i=1}^{K'} X_{ij}^2 < \frac{K'}{n} + \lambda^2(K')\right) + \sum_{j \notin S_f} \mathbf{P}_{f_j}\left(\sum_{i=1}^{K'} X_{ij}^2 \geq \frac{K'}{n} + \lambda^2(K')\right).
\end{aligned}
\tag{28}
$$

Therefore, we will discuss the Hamming loss on the support and non-support separately.

### C.2.2 Support

With signal condition

$$
\|f_j\|_2^2 \geq \left(24\sqrt{\frac{1}{\delta}} + \sqrt{2}\right) \max_{k \in \mathbb{N}^+}\left(\frac{\sqrt{k \log p}}{n} \wedge \mu_k\right) + \frac{36 \log p}{\delta n}
$$

holds for all $j \in S_f$, we have

$$
\begin{aligned}
\|\theta_{1:K',j}\|_2^2 = \sum_{i=1}^{K'} \theta_{ij}^2 \geq& \|f_j\|_2^2 - \mu_{K'} \sum_{i=K+1}^{\infty} \frac{\theta_{ij}^2}{\mu_i} \\
\overset{(i)}{\geq}& \left(12\sqrt{\frac{2}{\delta}} + 1\right) \frac{\sqrt{K' \log(p)}}{n} + \frac{36 \log p}{\delta n} - \mu_{K'} \\
\overset{(ii)}{\geq}& 12\sqrt{\frac{2}{\delta}} \frac{\sqrt{K' \log(p)}}{n} + \frac{36 \log p}{\delta n} \\
\overset{(iii)}{\geq}& \frac{12\sqrt{K' \log(2p/\delta)}}{n} + \frac{18 \log(2p/\delta)}{n},
\end{aligned}
\tag{29}
$$

where inequality (i) follows from the signal condition and a proof strategy similar to (63) in Lemma 4, inequality (ii) follows from the definition of $K'$ and inequality (iii) follows from $(2/\delta) \log p \geq \log(2p/\delta)$ when $p \geq 2$.

We decompose $X_{ij} = \theta_{ij} + \xi_{ij}$ with each $\xi_{ij} \sim N(0, 1/n)$ independently. Then we get

$$\mathbf{P}_{f_j}\left(\sum_{i=1}^{K'} X_{ij}^2 < \frac{K}{n} + \lambda^2\right)$$

$$= \mathbf{P}_{f_j}\left(\|\xi_{1:K',j}\|_2^2 + \|\theta_{1:K',j}\|_2^2 + 2\langle\xi_{1:K',j}, \theta_{1:K',j}\rangle < \frac{K'}{n} + \lambda^2(K')\right)$$

$$\overset{(i)}{\leq} \mathbf{P}_{f_j}\left(\left\{\|\xi_{1:K,j}\|_2^2 + \|\theta_{1:K,j}\|_2^2 + 2\langle\xi_{1:K,j}, \theta_{1:K,j}\rangle < \frac{K}{n} + \lambda^2(K')\right\} \cap \mathcal{A}_j\right) + \mathbf{P}_{f_j}\left(\mathcal{A}_j^c\right)$$

$$\leq \mathbf{P}_{f_j}\left(\|\xi_{1:K',j}\|_2^2 + \|\theta_{1:K',j}\|_2^2 - \|\theta_{1:K',j}\|_2\sqrt{\frac{8\log(2p/\delta)}{n}} < \frac{K'}{n} + \lambda^2\right) + \exp\left(-\log(2p/\delta)\right)$$

$$\overset{(ii)}{\leq} \mathbf{P}_{f_j}\left(\|\xi_{1:K,j}\|_2^2 + \frac{1}{3}\|\theta_{1:K,j}\|_2^2 < \frac{K}{n} + \lambda^2(K')\right) + \frac{\delta}{2p}$$

$$\overset{(iii)}{\leq} \mathbf{P}_{f_j}\left(n\|\xi_{1:K',j}\|_2^2 < K' - 2\sqrt{K'\log(2p/\delta)}\right) + \frac{\delta}{2p}$$

$$\leq \frac{\delta}{p},$$

(30)

where in inequality (i) we define event $\mathcal{A}_j = \left\{\langle\xi_{1:K',j}, \theta_{1:K',j}\rangle > -\sqrt{\frac{2\|\theta_{1:K',j}\|_2^2\log(2p/\delta)}{n}}\right\}$, where

$\langle\xi_{1:K',j}, \theta_{1:K',j}\rangle \sim N\left(0, \frac{\|\theta_{1:K',j}\|_2^2}{n}\right)$. Inequality (ii) and (iii) follow from (29), and the last inequality follows from (60) in Lemma 2.

### C.2.3 Non-support

We now focus on the Hamming loss on the non-support. For every $j \notin S$, we have $n\sum_{i=1}^K X_{ij}^2 \sim \chi_K^2(0)$, therefore

$$\mathbf{P}_{f_j}\left(\sum_{i=1}^K X_{ij}^2 \geq \frac{K}{n} + \lambda^2\right) \leq \frac{\delta}{2p}, \tag{31}$$

where the last inequality follows from (61) in Lemma 2.

Combining (28), (30), and (31), we conclude

$$\mathbf{E}_f\left(H(\hat{\eta}(X), \eta(f))\right) = \sum_{j \in S_f} \mathbf{P}_{f_j}\left(\sum_{i=1}^{K'} X_{ij}^2 < \frac{K}{n} + \lambda^2(K')\right) + \sum_{j \notin S_f} \mathbf{P}_{f_j}\left(\sum_{i=1}^{K'} X_{ij}^2 \geq \frac{K}{n} + \lambda^2(K')\right)$$

$$\overset{(i)}{\leq} \frac{|S_f|\delta}{p} + \frac{(p - |S_f|)\delta}{2p}$$

$$= \frac{(p + |S_f|)\delta}{2p} \leq \delta,$$

where in inequality (i), we use $|S_f|$ to denote the cardinal number of the support index set $S_f$, hence $1 \leq |S_f| \leq s \leq p$. Therefore, we complete the proof of the upper bound and also Theorem 3.

## D  Proof of Theorem 1

In the proof of the upper bound, we first recall our signal condition

$$\|f_j\|_2^2 \geq \left(\frac{6}{\sqrt{\delta}}\left(\sqrt{10} + 2\right) + \sqrt{2}\right)\max_{k \in \mathbb{N}^+}\left(\frac{\sqrt{k\log(p/s)}}{n} \wedge \mu_k\right) + \frac{36}{\delta n}\log(p/s)$$

and the selector

$$\hat{\eta}_j = \mathbf{1}\left(n\sum_{i=1}^K X_{ij}^2 \geq K + 2\sqrt{5K\log(p/(s\delta))} + 10\log(p/(s\delta))\right),$$

where $K = \min \left\{ k \in \mathbb{N}^+ : \mu_k \leq \frac{\sqrt{k \log(p/s)}}{n} \right\}$ and $n = \sigma^{-2}$. We assume $p/s \geq \sqrt{12} \vee \sqrt{6/\delta}$, $s \geq 16 \vee C_\delta$, and $n \geq \frac{C_\delta \log(p/s)}{\mu_1}$, where $C_\delta$ is a positive constant solely determined by $\delta \in (0, 1)$.

## D.1 FNR control

Similar to (29) in the proof of Theorem 3, we get

$$\|\theta_{1:K,j}\|_2^2 \geq 6 \left( \sqrt{5} + \sqrt{2} \right) \frac{\sqrt{K \log(p/(s\delta))}}{n} + \frac{36}{n} \log(p/(s\delta)).$$

Then for $j \in S_f$, similar to (30) we get

$$
\begin{aligned}
\mathbf{E}_f \left( \frac{\sum_{j \in S_f} (1 - \hat{\eta}_j)}{s} \right) &= \mathbf{E}_f \left( 1 - \hat{\eta}_j \right) \\
&\leq \mathbf{P} \left( \chi_K^2(0) + \frac{n}{3} \|\theta_{1:K,j}\|_2^2 < K + 2\sqrt{5K \log(p/(s\delta))} + 10 \log(p/(s\delta)) \right) \\
&\quad + \mathbf{P} \left( 2n \langle \xi_{1:K,j}, \theta_{1:K,j} \rangle \leq -4\sqrt{n \|\theta_{1:K,j}\|_2^2 \log(p/(s\delta))} \right) \\
&\leq \mathbf{P}_{f_j} \left( \chi_K^2(0) < K - \sqrt{8K \log(p/(s\delta))} \right) + \left( \frac{s}{p} \right)^2 \delta \\
&\leq 2 \left( \frac{s}{p} \right)^2 \delta,
\end{aligned}
\tag{32}
$$

which also leads to

$$\mathbf{E}_f \left( \sum_{j \in S_f} \hat{\eta}_j \right) \geq s - 2s \left( \frac{s}{p} \right)^2 \delta.$$

Besides, by Hoeffding's inequality, we get

$$\mathbf{P} \left( \sum_{j \in S_f} \hat{\eta}_j - \mathbf{E} \sum_{j \in S_f} \hat{\eta}_j \leq -s^{3/4} \right) \leq \exp \left( -2\sqrt{s} \right),$$

yielding

$$\mathbf{P} \left( \sum_{j \in S_f} \hat{\eta}_j \leq s - 2s \left( \frac{s}{p} \right)^2 \delta - s^{3/4} \right) \leq \exp \left( -2\sqrt{s} \right).
\tag{33}$$

## D.2 FDR control

By Markov's inequality, we conclude

$$
\begin{aligned}
\mathbf{P} \left( \sum_{j \notin S_f} \hat{\eta}_j > \frac{s}{(p/s)^2} \right) &\leq \frac{(p/s)^2}{s} \sum_{j \notin S_f} \mathbf{P} \left( n \sum_{i=1}^K X_{ij}^2 \geq K + 2\sqrt{5K \log(p/(s\delta))} + 10 \log(p/(s\delta)) \right) \\
&\leq \left( \frac{s}{p} \right)^2 \delta,
\end{aligned}
\tag{34}
$$

which yields

$$
\begin{aligned}
\mathbf{E} & \left\{ \frac{\sum_{j \notin S_f} \hat{\eta}_j}{1 \vee \sum_{j \in [p]} \hat{\eta}_j} \cdot \mathbf{1} \left( \sum_{j \in S_f} \hat{\eta}_j > s - 2s \left( \frac{s}{p} \right)^2 \delta - s^{3/4} \right) \cdot \mathbf{1} \left( \sum_{j \notin S_f} \hat{\eta}_j \leq \frac{s}{(p/s)^2} \right) \right\} \\
&\leq \frac{\frac{s}{(p/s)^2}}{\frac{s}{(p/s)^2} + s - 2s(s/p)^2 \delta - s^{3/4}} \\
&\overset{(i)}{\leq} \frac{\frac{s}{(p/s)^2}}{\frac{s}{(p/s)^2} + \frac{s}{3}} \leq \frac{2\delta}{3},
\end{aligned}
\tag{35}
$$

where inequality (i) follows from $p/s \geq \sqrt{12}$ and $s \geq 16$, and the last inequality follows from $p/s \geq \sqrt{9/(2\delta)}$. Besides, by a similar technique, we have

$$\mathbf{E}\left\{\frac{\sum_{j \notin S_f} \hat{\eta}_j}{1 \vee \sum_{j \in [p]} \hat{\eta}_j} \cdot \mathbf{1}\left(\sum_{j \notin S_f} \hat{\eta}_j > \frac{s}{(p/s)^2}\right)\right\} \leq \mathbf{P}\left\{\sum_{j \notin S_f} \hat{\eta}_j > \frac{s}{(p/s)^2}\right\} \leq \left(\frac{s}{p}\right)^2 \delta, \quad (36)$$

where the last inequality follows from (34), and the first inequality holds because $\frac{\sum_{j \notin S} \hat{\eta}_j}{1 \vee \sum_{j \in [p]} \hat{\eta}_j} \leq 1$. We also get

$$\mathbf{E}\left\{\frac{\sum_{j \notin S_f} \hat{\eta}_j}{1 \vee \sum_{j \in [p]} \hat{\eta}_j} \cdot \mathbf{1}\left(\sum_{j \in S_f} \hat{\eta}_j \leq s - 2s\left(\frac{s}{p}\right)^2 \delta - s^{3/4}\right) \cdot \mathbf{1}\left(\sum_{j \notin S_f} \hat{\eta}_j \leq \frac{s}{(p/s)^2}\right)\right\}$$

$$\leq \frac{s}{(p/s)^2} \mathbf{P}\left\{\sum_{j \in S_f} \hat{\eta}_j \leq s - 2s\left(\frac{s}{p}\right)^2 \delta - s^{3/4}\right\} \qquad (37)$$

$$\overset{(i)}{\leq} \frac{s e^{-2\sqrt{s}}}{(p/s)^2} \leq \left(\frac{s}{p}\right)^2 \delta,$$

where inequality (i) follows from (33), and the last inequality is based on that the function $g(x) = x e^{-2\sqrt{x}}$ is monotonically decreasing and tends to $0$ on $(1, \infty)$, and hence for every $\delta \in (0, 1)$, there exists a corresponding $C_\delta$ such that $s > C_\delta$ yields $s e^{-2\sqrt{s}} \leq \delta$.

### D.3 Conclusion

Combining (32), (35) (36), and (37), we get

$$\mathbf{E}\left\{\frac{\sum_{j \notin S_f} \hat{\eta}_j}{1 \vee \sum_{j \in [p]} \hat{\eta}_j} + \frac{\sum_{j \in S_f}(1 - \hat{\eta}_j)}{s}\right\}$$

$$= \mathbf{E}\left\{\frac{\sum_{j \notin S_f} \hat{\eta}_j}{1 \vee \sum_{j \in [p]} \hat{\eta}_j} \cdot \mathbf{1}\left(\sum_{j \in S_f} \hat{\eta}_j > s - 2s\left(\frac{s}{p}\right)^2 \delta - s^{3/4}\right) \cdot \mathbf{1}\left(\sum_{j \notin S_f} \hat{\eta}_j \leq \frac{s}{(p/s)^2}\right)\right\}$$

$$+ \mathbf{E}\left\{\frac{\sum_{j \notin S_f} \hat{\eta}_j}{1 \vee \sum_{j \in [p]} \hat{\eta}_j} \cdot \mathbf{1}\left(\sum_{j \in S_f} \hat{\eta}_j \leq s - 2s\left(\frac{s}{p}\right)^2 \delta - s^{3/4}\right) \cdot \mathbf{1}\left(\sum_{j \notin S_f} \hat{\eta}_j \leq \frac{s}{(p/s)^2}\right)\right\}$$

$$+ \mathbf{E}\left\{\frac{\sum_{j \notin S_f} \hat{\eta}_j}{1 \vee \sum_{j \in [p]} \hat{\eta}_j} \cdot \mathbf{1}\left(\sum_{j \notin S_f} \hat{\eta}_j > \frac{s}{(p/s)^2}\right)\right\}$$

$$+ \mathbf{E}\left\{\frac{\sum_{j \in S_f}(1 - \hat{\eta}_j)}{s}\right\}$$

$$\leq \frac{2\delta}{3} + \left(\frac{s}{p}\right)^2 \delta + \left(\frac{s}{p}\right)^2 \delta + 2\left(\frac{s}{p}\right)^2 \delta \leq \delta,$$

where the last inequality follows from $p/s \geq \sqrt{12}$. Therefore we get an upper bound of the combined risk in sparse multiple testing with a rate-optimal signal condition, which completes the proof of Theorem 1.

## E Proof of Theorem 2

The proof of Theorem 2 uses a similar technique to Section C.1 in the proof of Theorem 3. Recall the decoder $\eta_j = \eta_j(f_j) = \mathbf{1}(f_j \neq 0)$ and the corresponding vector $\eta = \eta(f) := (\eta_1(f_1), \cdots, \eta_p(f_p))^\top \in \{0, 1\}^p$. This proof focuses on the SPAM space $\tilde{\mathcal{F}}_s^{(k)}(r^2)$, which is

defined in (19). For every $f \in \tilde{\mathcal{F}}_s^{(k)}(r^2)$, its specific form is only determined by decoder $\eta \in \{0,1\}^p$, therefore a prior of $\eta$ can also be realized as a prior of the function space $\tilde{\mathcal{F}}_s^{(k)}(r^2)$, and in next we may use the notation $f = f(\eta)$.

We assume $\log \frac{c_2(\lfloor p/s \rfloor - 1)}{-\log(1 - c_{\max}(\delta))} \geq \max\left(\frac{16}{c_1^2}, \frac{32}{c_1} \log \frac{1}{1 - c_{\max}(\delta)}, 1\right)$, $s \geq \frac{-\log(1 - \sqrt{1-\delta})}{c\kappa^2(\delta)c_{\max}^2(\delta)}$, and $n \geq \frac{C_\delta \log(p/s)}{\mu_1}$, where $c_{\max}(\delta)$ and $\kappa(\delta)$ are two functions solely determined by $\delta \in (0,1)$ and will be clarified later.

## E.1 Preliminary

For any prior $\pi$ on $\{0,1\}^p$, we denote by $P_\pi$ the prior distribution of $\eta$. Then, similar to the proof of Theorem 3 we get

$$
\begin{aligned}
&\inf_{\hat{\eta}} \sup_{f \in \mathcal{F}_{=s}(r^2)} R(f, \hat{\eta}) \\
&\geq \inf_{\hat{\eta}: \mathbb{R}^{k \times p} \to \{0,1\}^p} \sup_{f \in \tilde{\mathcal{F}}_s^{(k)}(r^2)} R(f, \hat{\eta}) \\
&\overset{(i)}{\geq} \inf_{\hat{\eta}: \text{ norm selector}} \sup_{f \in \tilde{\mathcal{F}}_s^{(k)}(r^2)} R(f, \hat{\eta}) \\
&\overset{(ii)}{\geq} \frac{M(1 - \kappa)}{s + M(1 - \kappa)} \left(1 - e^{-c\kappa^2 M}\right) - p\mathbf{P}_\pi \left(\sum_{j \in [p]} \eta_j > s\right) - 2\mathbf{P}_\pi\left(f(\eta) \notin \tilde{\mathcal{F}}_s^{(k)}(r^2)\right),
\end{aligned}
$$

where inequality (i) follows from Lemma 1, and the last inequality (ii) follows from Lemma 3, with

$$
M = \sum_{j \in [p]} \mathbf{P}_\pi \left(\eta_j = 1, \mathbf{P}_{\pi|Z}(\eta_j = 0 \mid Z) > \frac{1}{2}\right),
$$

where $Z = (\|X_{1:k,1}\|_2, \cdots, \|X_{1:k,p}\|_2) \in \mathbb{R}^p$. This formulation is justified by Lemma 1, which indicates that we can focus solely on the norms of each column in $X_{1:K,\cdot}$, with $X_{1:K,\cdot}$ denoting the first $K$ rows of $X$. In particular, inequality (ii) holds with every $\kappa \in (0,1)$ and every prior $\pi$ on $\{0,1\}^p$. Next, we will construct a block prior distribution $\pi$ to conduct the lower bound of the minimax separation rate.

## E.2 The block prior

We consider a block prior $\pi$ which has often been used [Butucea et al., 2023a, Abraham et al., 2024]. Take prior $\pi$ as a product prior over $s+1$ blocks of consecutive coordinates $B_1 = \{1, 2, \cdots, q\}$, $B_2 = \{q+1, \cdots, 2q\}, \cdots, B_s = \{(s-1)q + 1, \cdots, p'\}$, where $q = \lfloor p/s \rfloor$ and $p' = qs$. We write $B_{s+1}$ for the (possibly empty) set $\{p' + 1, \cdots, p\}$. In each block $B_b, b \in [s]$, we uniformly choose an index $i \in B_b$ and set $\eta_i = 1$ and $\eta_j = 0$ for all $j \in B_b$ and $j \neq i$. For every $i \in B_{s+1}$, we just set $\eta_i = 0$. With this prior, we have

$$
\mathbf{P}_\pi \left(\sum_{j \in [p]} \eta_j > s\right) = \mathbf{P}_\pi\left(f(\eta) \notin \tilde{\mathcal{F}}_s^{(k)}(r^2)\right) = 0.
$$

Then we have

$$M = \sum_{j \in [p]} \mathbf{P}_\pi \Big( \eta_j = 1, \mathbf{P}_{\pi|Z} \left( \eta_j = 0 | Z \right) > 1/2 \Big)$$

$$= \sum_{b \in [s]} \sum_{j \in B_b} \mathbf{P}_\pi^{B_b} \left( \eta_j = 1, \frac{\sum_{u \in B_b \setminus \{j\}} \mathbf{P}_\pi \left( \eta_u = 1, Z \right)}{\sum_{u \in B_b} \mathbf{P}_\pi \left( \eta_u = 1, Z \right)} > 1/2 \right)$$

$$\stackrel{(i)}{=} \sum_{b \in [s]} \sum_{j \in B_b} \mathbf{P}_\pi^{B_b} \left( \eta_j = 1, \sum_{u \in B_b \setminus \{j\}} \frac{f_1}{f_0} (n\|X_{1:k,u}\|_2^2) > \frac{f_1}{f_0} (n\|X_{1:k,j}\|_2^2) \right)$$

$$= \sum_{b \in [s]} \sum_{j \in B_b} \mathbf{P}_{X|e_j}^{B_b} \left( \sum_{u \in B_b \setminus \{j\}} \frac{f_1}{f_0} (n\|X_{1:k,u}\|_2^2) > \frac{f_1}{f_0} (n\|X_{1:k,j}\|_2^2) \,\Big|\, \eta_j = 1 \right) \times \mathbf{P}_\pi^{B_b} \left( \eta_j = 1 \right)$$

$$\geq \sum_{b \in [s]} \sum_{j \in B_b} \frac{1}{q} \mathbf{P}_{X|e_j}^{B_b} \left( n\|X_{1:k,j}\|_2^2 \leq \max_{u \in B_b \setminus \{j\}} n\|X_{1:k,u}\|_2^2 \right),$$

where equality (i) follows the same notation in (20), and in the last inequality we focus on the probability on the block $B_b$, where "$X|e_j$" means we assume $\eta_u = 1$ and $\eta_i = 0$ for all $u \in B_b \setminus \{j\}$. Therefore, we transform the problem into the one we dealt with in Section C.1.2.

Recall we denote by $c_{\max}(\delta)$ and $\kappa(\delta)$ two functions of $\delta \in (0,1)$ independent of $p$, $s$, $k$ and n, which will be determined later. By taking $x = \log \frac{c_2(q-1)}{-\log(1-c_{\max}(\delta))}$, we follow a proof strategy similar to (22) and get

$$\mathbf{P}_{X|e_j}^{B_b} \left( \max_{u \in B_b \setminus \{j\}} n\|X_{1:k,u}\|_2^2 \geq k + c_1 x + c_1 \sqrt{kx} \right) \geq c_{\max}(\delta).$$

Additionally, when $nr^2 \leq \frac{c_1}{4} \left( x + \sqrt{kx} \right)$, we define $t := \log \frac{1}{1-c_{\max}(\delta)} > 1$ and have

$$nr^2 + 2\sqrt{(k + 2nr^2)t} + 2t \leq nr^2 + 2\sqrt{2nr^2 t} + 2\sqrt{kt} + 2t$$

$$\leq \frac{c_1}{4} \left( x + \sqrt{kx} \right) + 2\sqrt{\frac{c_1 t}{2} \left( x + \sqrt{kx} \right)} + 2 \left( \sqrt{k} + 1 \right)$$

$$\stackrel{(i)}{\leq} \frac{c_1}{2} \left( x + \sqrt{kx} \right) + \frac{c_2}{2} \sqrt{x} \left( \sqrt{k} + 1 \right)$$

$$\leq c_1 \left( x + \sqrt{kx} \right),$$

where inequality (i) follows the assumption $x = \log \frac{c_2(q-1)}{-\log(1-c_{\max}(\delta))} \geq \max \left( \frac{16}{c_1^2}, \frac{32}{c_1} \log \frac{1}{1-c_{\max}(\delta)}, 1 \right)$. Therefore, we follow a proof strategy similar to (24) and get

$$\mathbf{P} \left( \chi_k^2(nr^2) \leq k + c_1 x + c_1 \sqrt{kx} \right) \geq c_{\max}(\delta),$$

which leads that

$$\inf_{\hat\eta} \sup_{f \in \mathcal{F}_{=s}(r^2)} R(f, \hat\eta) \geq \frac{c_{\max}^2(\delta)(1 - \kappa(\delta))}{1 + c_{\max}^2(\delta)(1 - \kappa(\delta))} \left( 1 - e^{-cs\kappa^2(\delta)c_{\max}^2(\delta)} \right).$$

Note that we can always choose suitable $c_{\max}(\delta)$ and $\kappa(\delta)$ to let $\frac{c_{\max}^2(\delta)(1-\kappa(\delta))}{1+c_{\max}^2(\delta)(1-\kappa(\delta))} = \sqrt{1-\delta}$, and then by assuming $s \geq \frac{-\log(1-\sqrt{1-\delta})}{c\kappa^2(\delta)c_{\max}^2(\delta)}$, we conclude $\inf_{\hat\eta} \sup_{f \in \mathcal{F}_{=s}(r^2)} R(f, \hat\eta) \geq 1 - \delta$.

The optimal truncation $k$ follows a similar analysis as Section C.1.3, which proves that we cannot control the FDR plus FNR well when

$$r^2 \leq c_\delta \left\{ \frac{\log(p/s)}{n} + \max_{k \in \mathbb{N}^+} \left( \frac{\sqrt{k \log(p/s)}}{n} \wedge \mu_k \right) \right\},$$

which completes the proof of Theorem 2.

# F Proof of Theorem 4

The proof of Theorem 4 is similar to Section C.2. For simlpicity, we assume $\mu_k = k^{-2\alpha}$. Recall $\mathcal{K}_{rec} := \left\{ 2, 4, \cdots, 2^{\left\lceil \log_2\left(\frac{n^2}{\log p}\right)\right\rceil} \right\}$, we conclude $|\mathcal{K}_{rec}| \leq \log_2\left(\frac{n^2}{\log p}\right) + 1 \leq 4\log n$.

## F.1 Support

For every $\alpha \geq \alpha_{\min}$, we define $K^* := 2^{\left\lceil \frac{1}{1+4\alpha} \log_2\left(\frac{n^2}{\log\left(\frac{8p\log n}{\delta}\right)}\right)\right\rceil}$. By $0 < \frac{1}{1+4\alpha} \log_2\left(\frac{n^2}{\log\left(\frac{8p\log n}{\delta}\right)}\right) \leq \log_2\left(\frac{n^2}{\log p}\right)$, we conclude $K^* \in \mathcal{K}_{rec}$. Now with the signal condition

$$\|f_j\|_2^2 \geq \left(12\sqrt{2}+1\right)\left(\frac{\log(8p/\delta \cdot \log n)}{n^2}\right)^{\frac{2\alpha}{1+4\alpha}} + \frac{18\log(8p/\delta \cdot \log n)}{n}$$

holding for all $j \in S_f$, we have

$$\|\theta_{1:K^*,j}\|_2^2 = \sum_{i=1}^{K^*} \theta_{ij}^2$$

$$\geq \left(12\sqrt{2}+1\right)\left(\frac{\log(\frac{8p\log n}{\delta})}{n^2}\right)^{\frac{2\alpha}{1+4\alpha}} + \frac{18\log(\frac{8p\log n}{\delta})}{n} - (K^*)^{-2\alpha} \qquad (38)$$

$$\overset{(i)}{\geq} \frac{12\sqrt{K^*\log(\frac{8p\log n}{\delta})}}{n} + \frac{18\log(\frac{8p\log n}{\delta})}{n},$$

where inequality (i) follows from $\left(\frac{n^2}{\log(\frac{8p\log n}{\delta})}\right)^{\frac{1}{1+4\alpha}} \leq K^* \leq 2\left(\frac{n^2}{\log(\frac{8p\log n}{\delta})}\right)^{\frac{1}{1+4\alpha}}$. Then for every $j \in S$, we conclude that

$$\mathbf{E}_{f_j}\left\{\mathbf{1}\left(\hat{\eta}_j^{ad}(X) \neq 1\right)\right\}$$

$$\leq \mathbf{P}_{f_j}\left(\hat{\eta}_j^{K^*} = 0\right)$$

$$= \mathbf{P}_{f_j}\left(n\sum_{i=1}^{K^*} X_{ij}^2 < K^* + 2\sqrt{K^*\log\left(\frac{8p\log n}{\delta}\right)} + 2\log\left(\frac{8p\log n}{\delta}\right)\right) \qquad (39)$$

$$\overset{(i)}{\leq} \mathbf{P}\left(\chi_{K^*}^2(0) < K^* - 2\sqrt{K^*\log(2p/\delta)}\right) + \mathbf{P}\left(N(0,1) \leq -\sqrt{2\log(2p/\delta)}\right)$$

$$\leq \frac{\delta}{p},$$

where inequality (i) follows from (38), and uses a similar technique in (30) from the proof of Theorem 3.

## F.2 Non-support

For every $j \notin S_f$, by the subadditivity of the probability measure, we have

$$\mathbf{E}_{f_j}\left\{\mathbf{1}\left(\hat{\eta}_j^{ad}(X) \neq 0\right)\right\}$$

$$= \mathbf{P}_{f_j}\left(\bigcup_{k \in \mathcal{K}_{rec}}\left\{n\sum_{i=1}^{k} X_{ij}^2 \geq k + 2\sqrt{k\log\left(\frac{8p\log n}{\delta}\right)} + 2\log\left(\frac{8p\log n}{\delta}\right)\right\}\right)$$

$$\leq \sum_{k \in \mathcal{K}_{rec}} \mathbf{P}\left(\chi_k^2(0) \geq k + 2\sqrt{k\log\left(\frac{8p\log n}{\delta}\right)} + 2\log\left(\frac{8p\log n}{\delta}\right)\right) \qquad (40)$$

$$\overset{(i)}{\leq} |\mathcal{K}_{rec}|\frac{\delta}{8p\log n} \leq \frac{\delta}{2p}.$$

where inequality (i) follows from (61) in Lemma 2, and the last inequality follows from $|\mathcal{K}_{rec}| \leq 4\log n$. Combining (39) and (40), we conclude that

$$\mathbf{E}_f\left\{H\left(\hat{\eta}^{ad}(X), \eta(f)\right)\right\} = \sum_{j \in S_f}\mathbf{E}_{f_j}\left\{\mathbf{1}\left(\hat{\eta}_j^{ad}(X) \neq 1\right)\right\} + \sum_{j \notin S_f}\mathbf{E}_{f_j}\left\{\mathbf{1}\left(\hat{\eta}_j^{ad}(X) \neq 0\right)\right\}$$

$$\leq \frac{|S|\delta}{p} + \frac{(p-|S|)\delta}{2p} \leq \delta,$$

which completes the proof of Theorem 4.

# G   Proof of Theorem 5

## G.1   Lower bound with truncation

**Preliminary**   Firstly, we construct a prior uniform distribution of $\alpha$ as

$$\mathbf{P}\left(\alpha = \frac{1}{4}\left(-1 + \frac{\log\frac{n^2}{c\log(p\log n)}}{\log k}\right)\right) = \frac{1}{|\mathcal{K}_{rec}|}, \quad \text{for each } k \in \mathcal{K}_{rec},$$

where the constant $c > 0$ will be determined later. The prior of $\alpha$ corresponds to a uniformly distributed truncation $K$ in $\mathcal{K}_{rec}$ as

$$\mathbf{P}\left(K = k\right) = \frac{1}{|\mathcal{K}_{rec}|}, \quad \text{for each } k \in \mathcal{K}_{rec}.$$

We next construct a prior distribution of function $f \in \mathcal{F}_s(r^2, \alpha)$ for the given $\alpha$ (i.e., for the given $k \in \mathcal{K}_{rec}$). Specifically, we assume that only 0 or 1 univariate function can be the support, that is, $f = 0$, or $f = f_j$ for a $j \in [p]$. For the support $f_j$, if $i \leq k$, assume that its $i$-th entry is drawn from a uniform distribution as

$$\theta_{i,j} \sim \begin{pmatrix} \lambda(k) & -\lambda(k) \\ 1/2 & 1/2 \end{pmatrix},$$

where $\lambda(k) := n^{-1/2}\left(\frac{c\log(p\log n)}{k}\right)^{1/4}$ Otherwise just take $\theta_{ij} = 0$. Conversely, $f = 0$ directly indicates $\theta_{ij} = 0$ for each $i \in \mathbb{N}^+$ and $j \in [p]$. After we get an $f = \{\theta_{ij}\}_{i \in \mathbb{N}^+, j \in [p]}$, assume $X_{ij} \sim N(\theta_{ij}, 1/n)$, indepedently.

Finally, for a given $\alpha$ derived from $k$, we get $\sum_{i \in \mathbb{N}^+}\frac{\theta_{ij}^2}{\mu_i} \leq k^{1+2\alpha(k)} \cdot \lambda(k)^2 = 1$, indicating the setting of $\lambda(k)$ is valid. Therefore, we name the distribution with respect to the truncation $k$ and support $f_j$ as $\mathbf{P}_{j,k}$, and

$$\mathbf{P}_{j,k}(X) = \prod_{i \in [k]}\frac{\phi_{\lambda(k), 1/n}(X_{i,j}) + \phi_{-\lambda(k), 1/n}(X_{i,j})}{2} \times \prod_{i > k}\phi_{0, 1/n}(X_{i,j}) \times \prod_{j' \neq j, i \in \mathbb{N}^+}\phi_{0, 1/n}(X_{i,j'}).$$

For $j \in [p]$, we define $\mathbf{P}_j = \frac{1}{|\mathcal{K}_{rec}|}\sum_{k \in \mathcal{K}_{rec}}\mathbf{P}_{j,k}$, and we also denote by $\mathbf{P}_0$ the distribution with $j = 0$, i.e., $X_{ij} \sim N(0, 1/n)$ for each $(i,j) \in \mathbb{N}^+ \times [p]$.

Based on these settings, we transform the minimax lower bound into:

$$\inf_{\hat{\eta}}\sup_{\alpha > 0}\sup_{f \in \mathcal{F}_s(r^2, \alpha)}\mathbf{P}_f\left(\hat{\eta}(X) \neq \eta\right) \geq \inf_{\hat{\eta}}\sup_{k \in \mathcal{K}_{rec}}\sup_{j \in \{0\} \cup [p]}\mathbf{P}_{j,k}\left(\hat{\eta}(X) \neq \eta\right)$$

$$\geq \inf_{\hat{\eta}}\sup_{j \in \{0\} \cup [p]}\mathbf{P}_j\left(\hat{\eta}(X) \neq \eta\right).$$

$\chi^2$ **divergences calculation**   For $j \in [p]$, consider

$$\frac{d\mathbf{P}_j}{d\mathbf{P}_0}(X) = \frac{1}{|\mathcal{K}_{rec}|}\sum_{k \in \mathcal{K}_{rec}}\exp\left(-\frac{k \cdot \lambda(k)^2}{2/n}\right)\prod_{i \in [k]}\cosh\left(nX_{i,j} \cdot \lambda(k)\right),$$

which leads that

$$\mathbf{E}_{\mathbf{P}_0}\left(\frac{d\mathbf{P}_j}{d\mathbf{P}_0}(X)\right)^2$$

$$=\frac{1}{|\mathcal{K}_{rec}|^2}\sum_{k,k'\in\mathcal{K}_{rec}}\exp\left(-\frac{k\cdot\lambda(k)^2+k'\cdot\lambda(k')^2}{2/n}\right)\times\left\{\mathbf{E}\cosh\left(nX_{i,j}\cdot\lambda(k)\right)\cdot\cosh\left(nX_{i,j}\cdot\lambda(k')\right)\right\}^{k\wedge k'}$$

$$\times\left\{\mathbf{E}\cosh\left(nX_{i,j}\cdot[\lambda(k)\wedge\lambda(k')]\right)\right\}^{k\vee k'-k\wedge k'}$$

$$\overset{(i)}{=}\frac{1}{|\mathcal{K}_{rec}|^2}\sum_{k,k'\in\mathcal{K}_{rec}}\left\{\cosh\left[n\lambda(k)\lambda(k')\right]\right\}^{k\wedge k'}$$

$$\overset{(ii)}{\leq}\frac{1}{|\mathcal{K}_{rec}|^2}\sum_{k,k'\in\mathcal{K}_{rec}}\exp\left\{\frac{n^2\cdot\sqrt{k}\lambda^2(k)\cdot\sqrt{k'}\lambda^2(k')\cdot(k\wedge k')}{2\sqrt{k\cdot k'}}\right\}$$

$$=\frac{1}{|\mathcal{K}_{rec}|^2}\sum_{k,k'\in\mathcal{K}_{rec}}\exp\left\{\frac{c}{2}\cdot\log(p\log n)\cdot\frac{k\wedge k'}{\sqrt{k\cdot k'}}\right\},$$

where equality (i) follows from Lemma 6, inequality (ii) follows from $\cosh(x)\leq\exp(x^2/2)$ for every $x\in\mathbb{R}$, and the last equality follows from the definition of $\lambda(k)$.

We now define $q:=|\mathcal{K}_{rec}|=\left\lceil\log_2\left(\frac{n^2}{\log p}\right)\right\rceil\asymp\log n$, and then conclude

$$\chi^2(\mathbf{P}_j||\mathbf{P}_0)\leq\frac{1}{q^2}\sum_{u,v\in[q]}\exp\left\{\frac{c\log(p\log n)}{2}\cdot2^{-\frac{|u-v|}{2}}\right\}-1.$$

**Control the wrong recovery probability**  Inspired by Gao et al. [2020], we divide the set $[q]\times[q]$ into two subset as $T_1:=\{(u,v)\in[q]\times[q]:|u-v|\leq2\log_2 q\}$ and $T_2:=\{(u,v)\in[q]\times[q]:|u-v|>2\log_2 q\}$, therefore $|T_1|\leq5q\log_2 q$ and $|T_2|\leq q^2$. We then have

$$1+\chi^2(\mathbf{P}_j||\mathbf{P}_0)\leq\frac{1}{q^2}\sum_{(u,v)\in T_1}\exp\left\{\frac{c\log(p\log n)}{2}\right\}+\frac{1}{q^2}\sum_{(u,v)\in T_2}\exp\left\{\frac{c\log(p\log n)}{2q}\right\}$$

$$\leq\frac{5\log_2 q}{q}\exp\left\{\frac{c\log(p\log n)}{2}\right\}+\exp\left\{\frac{c\log(p\log n)}{2q}\right\}$$

$$\leq\frac{\delta^2 p}{8}+\frac{\delta^2 p}{8},$$

where the last inequality follows from $\frac{c}{2}\log(p\log n)\leq\log\left(\frac{\delta^2 p\log_2 n}{40\log_2\log_2 n}\right)$ and $c\log(p\log n)\leq2\log(\delta^2 p/8)\log_2 n$, both of which hold under sufficiently large $n,p$ and sufficiently small constant $c$. Therefore, by Lemma 7, we prove that with signal strength

$$0<r^2=k\cdot\lambda(k)^2\leq\left(\frac{c\log(p\log n)}{n^2}\right)^{\frac{2\alpha}{1+4\alpha}},\tag{41}$$

the wrong recovery probability is out of control, i.e.,

$$\inf_{\hat{\eta}}\sup_{\alpha>0}\sup_{f\in\mathcal{F}_s(r^2,\alpha)}\mathbf{P}_f\left(\hat{\eta}(X)\neq\eta\right)\geq\inf_{\hat{\eta}}\sup_{j\in\{0\}\cup[p]}\mathbf{P}_j\left(\hat{\eta}(X)\neq\eta\right)>1-\delta.\tag{42}$$

## G.2   Lower bound with sparse structure

Since

$$\left(\frac{\log(p\log n)}{n^2}\right)^{\frac{2\alpha}{1+4\alpha}}+\frac{\log(p\log n)}{n}\asymp\max\left\{\left(\frac{\log(p\log n)}{n^2}\right)^{\frac{2\alpha}{1+4\alpha}},\frac{\log(p\log n)}{n}\right\}$$

$$=\begin{cases}\left(\frac{\log(p\log n)}{n^2}\right)^{\frac{2\alpha}{1+4\alpha}} & \text{if }n^{\frac{1}{1+2\alpha}}\geq\log(p\log n),\\ \frac{\log(p\log n)}{n} & \text{if }n^{\frac{1}{1+2\alpha}}<\log(p\log n),\end{cases}$$

we only need to verify the necessity of the separation rate $\frac{\log(p\log n)}{n}$ under the condition $n^{\frac{1}{1+2\alpha}} < \log(p\log n)$. This condition indicates that $\log\log p \gtrsim \log n$, which leads to $c_\alpha \log(p\log n) \leq \log p$ for some constant $c_\alpha$ determined only by $\alpha$.

In Theorem 3, we prove that the wrong recovery probability can be lower bounded by $1-\delta$ if the signal strength $r^2 \leq \frac{c_\delta \log p}{n}$. Hence, for any valid smoothness parameter $\alpha^*$, if $n^{\frac{1}{1+2\alpha^*}} < \log(p\log n)$, we have

$$
\inf_{\hat{\eta}} \sup_{\alpha>0} \sup_{f\in\mathcal{F}_s\left(\frac{c_\delta c_\alpha \log(p\log n)}{n},\alpha\right)} \mathbf{P}_f\left(\hat{\eta}(X)\neq\eta\right) \geq \inf_{\hat{\eta}} \sup_{f\in\mathcal{F}_s\left(\frac{c_\delta \log p}{n},\alpha^*\right)} \mathbf{P}_f\left(\hat{\eta}(X)\neq\eta\right)
$$
$$
\geq 1-\delta. \tag{43}
$$

Therefore, combining (41), (42) and (43), we complete the proof of Theorem 5.

# H Proof of Theorem 6

This section proves the minimax optimality of the estimator (15). We begin by defining

$$
\lambda^2(k) := \frac{1}{n}\left(\sqrt{Ck\log p} + C\log p\right), \tag{44}
$$

therefore $\hat{\theta}_{ij} = X_{ij}\cdot\mathbf{1}(i\leq K_e)\cdot\mathbf{1}\left(\sum_{i'=1}^{K_e}X_{ij}^2 \geq K_e/n + \lambda^2(K_e)\right)$, where recall $K_e := \min\{k\in\mathbb{N}^+ : \mu_k \leq k/n\}$.

**Support** For each $f\in\mathcal{F}_s$, we denote by $S_f$ the index set of the support covariates in $f$. For every $j\in S_f$, we have

$$
\begin{aligned}
\mathbf{E}\left\|\hat{f}_j - f_j\right\|_2^2 &= \mathbf{E}\sum_{i=1}^{K_e}\left(\hat{\theta}_{ij} - \theta_{ij}\right)^2 + \sum_{i>K_e}\theta_{ij}^2 \\
&\overset{(i)}{\leq} \mathbf{E}\sum_{i=1}^{K_e}\left\{X_{ij} - \theta_{ij} - X_{ij}\cdot\mathbf{1}\left(\sum_{i'=1}^{K_e}X_{ij}^2 < K_e/n + \lambda^2(K_e)\right)\right\}^2 + \mu_{K_e} \\
&\leq 2\mathbf{E}\sum_{i=1}^{K_e}\left(X_{ij}-\theta_{ij}\right)^2 + 2\mathbf{E}\sum_{i=1}^{K_e}X_{ij}^2\cdot\mathbf{1}\left(\sum_{i'=1}^{K_e}X_{ij}^2 < K_e/n + \lambda^2(K_e)\right) + \mu_{K_e} \\
&\leq \frac{4K_e}{n} + 2\lambda^2(K_e) + \mu_{K_e}.
\end{aligned} \tag{45}
$$

where inequality (i) follows from $\sum_{i>K_e}\theta_{ij}^2 \leq \mu_{K_e}\sum_{i>K_e}\theta_{ij}^2/\mu_i \leq \mu_{K_e}$.

**Non-support** For every $j\notin S_f$, we have

$$
\begin{aligned}
\mathbf{E}\left\|\hat{f}_j - f_j\right\|_2^2 &= \mathbf{E}\left\{\left(-\frac{K_e}{n} + \sum_{i=1}^{K_e}X_{ij}^2\right)\cdot\mathbf{1}\left(\sum_{i'=1}^{K_e}X_{i',j}^2 \geq \frac{K_e}{n} + \lambda^2(K_e)\right)\right\} \\
&\quad + \mathbf{E}\left\{\frac{K_e}{n}\cdot\mathbf{1}\left(\sum_{i'=1}^{K_e}X_{i',j}^2 \geq \frac{K_e}{n} + \lambda^2(K_e)\right)\right\} \\
&\overset{(i)}{\leq} \sqrt{\mathbf{D}\left(\sum_{i=1}^{K_e}X_{ij}^2\right)}\cdot\sqrt{\mathbf{P}\left(\sum_{i'=1}^{K_e}X_{i',j}^2 \geq \frac{K_e}{n} + \lambda^2(K_e)\right)} \\
&\quad + \frac{K_e}{n}\cdot\mathbf{P}\left(\sum_{i'=1}^{K_e}X_{i',j}^2 \geq \frac{K_e}{n} + \lambda^2(K_e)\right) \\
&\overset{(ii)}{\leq} \frac{\sqrt{2K_e}}{n}\cdot\exp\left(-\frac{C}{4}\log p\right) + \frac{K_e}{n}\cdot\exp\left(-\frac{C}{2}\log p\right) \leq \frac{3K_e}{np},
\end{aligned} \tag{46}
$$

where in inequality (i) follows from Cauchy-Schwarz inequality, inequality (ii) follows from (61) in Lemma 2 with taking the constant $C \geq 4$.

Combining the definition of $\lambda^2(k)$, (45), and (46), we conclude

$$
\begin{aligned}
\mathbf{E}_f \left\| \hat{f} - f \right\|_2^2 &= \sum_{j \in S_f} \mathbf{E} \left\| \hat{f}_j - f_j \right\|_2^2 + \sum_{j \notin S_f} \mathbf{E} \left\| \hat{f}_j - f_j \right\|_2^2 \\
&\leq s \left( \frac{4K_e}{n} + 2\lambda^2(K_e) + \mu_{K_e} \right) + p \frac{3K_e}{np} \\
&\asymp \frac{sK_e}{n} + \frac{s\sqrt{K_e \log p}}{n} + \frac{s \log p}{n} + s\mu_{K_e}^2 \\
&\asymp \frac{s \log p}{n} + s \times \max_{k \in \mathbb{N}+} \left( \frac{k}{n} \wedge \mu_k \right),
\end{aligned}
\tag{47}
$$

where the last equality follows from $K_e/n + \mu_{K_e} \asymp \max_{k \in \mathbb{N}+} \left( \frac{k}{n} \wedge \mu_k \right)$. Therefore, by (47) we complete the proof of Theorem 6.

# I  Proof of Theorem 7

This section establishes a necessary signal condition for univariate function selection, under the case that we have to only leverage the first $K_e$ entries in $X_{.j}$ for each $j \in [p]$, that is, only use observations $X_{[K_e] \times [p]} := \{X_{ij}\}_{1 \leq i \leq K_e, 1 \leq j \leq p}$, where $K_e := \min\{k \in \mathbb{N}^+ : \mu_k \leq k/n\}$ is the optimal truncation for minimax function estimation, and $n = \sigma^{-2}$.

## I.1  Lower bound with truncation

**Preliminary**  We first construct a prior distribution of function $f \in \mathcal{F}_s(r^2)$. Specifically, in this prior, only 0 or 1 univariate function can be the support, that is, either $f = 0$, or $f = f_j$ for $j \in [p]$.

In the case $f = 0$, we take $\theta_{ij} = 0$ for each $(i, j) \in \mathbb{N}^+ \times [p]$, and thus assume $X_{ij} \sim N(0, 1/n)$, indepedently. We name the distribution as

$$
\mathbf{P}_0 := \prod_{i \in \mathbb{N}^+, j \in [p]} \phi_{0,1/n}(X_{i,j}).
$$

In the case $f = f_j$, we take $\theta_{i,j'} = 0$ for each $i \in \mathbb{N}^+, j' \in [p] \setminus \{j\}$. For the support $f_j$, we introduce two additional parameters $\lambda > 0, 1 \leq k \leq K_e$, which will be determined later. If $i \in [k]$, assume that its $i$-th entry is drawn as $\mathbf{P}(\theta_{i,j} = \lambda) = \mathbf{P}(\theta_{i,j} = -\lambda) = 1/2$. If $i = K_e + 1$, we take $\theta_{K_e+1,j} = \sqrt{\mu_{K_e+1}/2}$. For other $i$ we just take $\theta_{i,j} = 0$. We name the distribution with respect to the support $f_j$ as $\mathbf{P}_j$, and

$\mathbf{P}_j(X)$

$$
:= \underbrace{\prod_{i \in [k]} \frac{\phi_{\lambda,1/n}(X_{i,j}) + \phi_{-\lambda,1/n}(X_{i,j})}{2}}_{\text{Distribution of first } k \text{ entries in } f_j} \times \underbrace{\phi_{\sqrt{\mu_{K_e+1}/2},1/n}(X_{K_e+1,j})}_{\text{Distribution of the } (K_e+1)\text{-th entry in } f_j} \times \underbrace{\prod_{i \in \mathbb{N}^+ \setminus ([k] \cup \{K_e+1\})} \phi_{0,1/n}(X_{i,j})}_{\text{Distribution of the residual entries in } f_j}
$$

$$
\times \underbrace{\prod_{j' \neq j, i \in \mathbb{N}^+} \phi_{0,1/n}(X_{i,j'})}_{\text{Distribution of other } f_{j'}}
$$

for every $j \in [p]$. Based on these settings, we transform the minimax lower bound into:

$$
\begin{aligned}
&\inf_{\hat{\eta}(X_{[K_e] \times [p]}) \in \{0,1\}^p} \sup_{f \in \mathcal{F}_s(r^2)} \mathbf{P}_f \left( \hat{\eta}(X_{[K_e] \times [p]}) \neq \eta \right) \\
&\geq \inf_{\hat{\eta}(X_{[K_e] \times [p]}) \in \{0,1\}^p} \sup_{j \in \{0\} \cup [p]} \mathbf{P}_j \left( \hat{\eta}(X_{[K_e] \times [p]}) \neq \eta \right) \\
&= \inf_{\hat{\eta}(X_{[K_e] \times [p]}) \in \{0,1\}^p} \sup_{j \in \{0\} \cup [p]} \mathbf{P}_{j,[K_e] \times [p]} \left( \hat{\eta}(X_{[K_e] \times [p]}) \neq \eta \right),
\end{aligned}
$$

where $\mathbf{P}_{j,[K_e] \times [p]}$ is the marginal distribution of $X_{[K_e] \times [p]}$.

**Control the wrong recovery probability via marginal $\chi^2$ divergences** For $j \in [p]$, consider

$$\frac{\mathrm{d}\mathbf{P}_{j,[K_e]\times[p]}}{\mathrm{d}\mathbf{P}_{0,[K_e]\times[p]}}(X_{[K_e]\times[p]}) = \exp\left(-\frac{k\lambda^2}{2/n}\right)\prod_{i\in[k]}\cosh\left(n\lambda X_{i,j}\right),$$

which leads that

$$\chi^2\left(\mathbf{P}_{j,[K_e]\times[p]}\big\|\mathbf{P}_{0,[K_e]\times[p]}\right) = \mathbf{E}_{\mathbf{P}_{0,[K_e]\times[p]}}\left(\frac{\mathrm{d}\mathbf{P}_{j,[K_e]\times[p]}}{\mathrm{d}\mathbf{P}_{0,[K_e]\times[p]}}(X_{[K_e]\times[p]})\right)^2 - 1$$

$$= \cosh^k\left(n\lambda^2\right) - 1 \;\leq\; \exp\left(\frac{n^2\lambda^4 k}{2}\right) - 1.$$

Recall that we only focus on those restricted selectors based on observations $X_{[K_e]\times[p]}$, and hence only focus on the marginal distribution of $X_{[K_e]\times[p]}$. Then our aim is to find proper $(\lambda, k) \in \mathbb{R}^+ \times [K_e]$ such that:

$$\max : \lambda^2 k, \tag{48}$$

$$\text{subject to: } \exp\left(\frac{n^2\lambda^4 k}{2}\right) \leq c_1 \cdot p, \tag{49}$$

$$\lambda^2 k \leq \frac{\mu_{K_e}}{2}, \tag{50}$$

$$1 \leq k \leq K_e. \tag{51}$$

where (49) controls the average $\chi^2$ divergence and $c_1$ is a sufficiently small positive constant, (50) ensures our construction is in the ellipsoid $\mathcal{E}$, and (51) ensures that the truncation $k$ is valid. Therefore, by Lemma 7, we prove that with signal strength

$$0 < r^2 = k \cdot \lambda^2 + \frac{\mu_{K_e}}{2} \leq c_2\left\{\max_{k\in[K_e]}\left(\frac{\sqrt{k\log p}}{n}\wedge\mu_k\right) + \mu_{K_e+1}\right\}, \tag{52}$$

the wrong recovery probability is lower bounded by $1/2$:

$$\inf_{\hat\eta(X_{[K_e]\times[p]})\in\{0,1\}^p}\sup_{f\in\mathcal{F}_s(r^2)}\mathbf{P}_f\left(\hat\eta(X_{[K_e]\times[p]})\neq\eta\right)$$

$$\geq \inf_{\hat\eta(X_{[K_e]\times[p]})\in\{0,1\}^p}\sup_{j\in\{0\}\cup[p]}\mathbf{P}_{j,[K_e]\times[p]}\left(\hat\eta(X_{[K_e]\times[p]})\neq\eta\right) \geq \frac{1}{2}. \tag{53}$$

### I.2 Lower bound with sparse structure

We now quantify the influence of the sparse structure on the necessary signal condition. Similarly to Section I.1, we first construct a prior distribution of function $f \in \mathcal{F}_s(r^2)$, where only 0 or 1 univariate function can be the support.

In the case $f = 0$, we still take

$$\mathbf{P}_0 := \prod_{i\in\mathbb{N}^+,j\in[p]}\phi_{0,1/n}(X_{i,j}).$$

For every $j \in [p]$, in the case $f = f_j$, we only take the first entry $\theta_{1,j} = \lambda$, and take other $\theta_{i,j'} = 0$. The distribution is described as

$$\mathbf{P}_j(X) := \underbrace{\phi_{\lambda,1/n}(X_{1,j})}_{\text{Distribution of the first entry in } f_j} \times \underbrace{\prod_{i\geq 2}\phi_{0,1/n}(X_{i,j})}_{\text{Distribution of the residual entries in } f_j} \times \underbrace{\prod_{j'\neq j,i\in\mathbb{N}^+}\phi_{0,1/n}(X_{i,j'})}_{\text{Distribution of other } f_{j'}}.$$

Based on these settings, we transform the minimax lower bound into:

$$\inf_{\hat\eta(X_{[K_e]\times[p]})\in\{0,1\}^p}\sup_{f\in\mathcal{F}_s(r^2)}\mathbf{P}_f\left(\hat\eta(X_{[K_e]\times[p]})\neq\eta\right)$$

$$\geq \inf_{\hat\eta(X_{[K_e]\times[p]})\in\{0,1\}^p}\sup_{j\in\{0\}\cup[p]}\mathbf{P}_{j,[K_e]\times[p]}\left(\hat\eta(X_{[K_e]\times[p]})\neq\eta\right).$$

It is straightforward to check that
$$\chi^2 \left( \mathbf{P}_{j,[K_e]\times[p]} \big\| \mathbf{P}_{0,[K_e]\times[p]} \right) = \exp\left( n\lambda^2 \right) - 1.$$
Therefore, our aim is to find proper $\lambda \in \mathbb{R}^+$ such that:
$$\max : \lambda^2, \tag{54}$$
$$\text{subject to: } \exp\left( n\lambda^2 \right) \leq c_4 \cdot p, \tag{55}$$
$$\lambda^2 \leq \mu_1, \tag{56}$$
where (55) controls the average $\chi^2$ divergence and $c_4$ is a sufficiently small positive constant, (56) ensures our construction is in the ellipsoid $\mathcal{E}$. Under assumption $n \geq \frac{c_5 \log p}{\mu_1}$, by Lemma 7, we prove that with signal strength
$$0 < r^2 = \lambda^2 \leq c_6 \frac{\log p}{n}, \tag{57}$$
the wrong recovery probability is lower bounded by $1/2$:
$$\inf_{\hat{\eta}(X_{[K_e]\times[p]})\in\{0,1\}^p} \sup_{f\in\mathcal{F}_s(r^2)} \mathbf{P}_f \left( \hat{\eta}(X_{[K_e]\times[p]}) \neq \eta \right)$$
$$\geq \inf_{\hat{\eta}(X_{[K_e]\times[p]})\in\{0,1\}^p} \sup_{j\in\{0\}\cup[p]} \mathbf{P}_{j,[K_e]\times[p]} \left( \hat{\eta}(X_{[K_e]\times[p]}) \neq \eta \right) \geq \frac{1}{2}. \tag{58}$$
Therefore, combining (52), (53), (57) and (58), we complete the proof of Theorem 7.

# J  Some extended conclusions

This appendix discusses theoretical results obtained under assumptions more general than those in the main text.

## J.1  Violating the separate rate

In a more realistic setting where the true signal strength $\|f_j\|_2^2$ may fall below the minimax threshold, our selectors still have some useful properties:

1. The selector from Theorem 1 selects at most $2s$ variables with probability at least $1 - \delta$ (proved by following equation (34) in Appendix D.2).

2. The selector from equation (11) ensures $\hat{S} \subseteq S$ with probability at least $1 - \delta$, i.e., it guarantees zero false positives (proved by following Appendix C.2.3).

These guarantees hold without knowing the signal strength of each $f_j$ in advance, showing that our selectors remain both sparse and interpretable under a practical condition.

We also provide a specific example in which our procedures are appropriate. Consider a system with $p$ channels, some of which carry a true signal while the rest are pure white noise, and we aim to identify those channels with a signal. Then the selector (11) achieves that, with high probability, no noise-only channel is selected, and any channel whose signal strength exceeds the minimax separation rate (10) will be selected. In this way, our selector provides a false-positive-free method for this problem.

## J.2  Heterogeneous univariate functions

We now extend our framework to heterogeneous settings where $f_j \in \mathcal{H}_j$ and $\mathcal{H}_j$ might be different across $j \in [p]$. This setting was considered in Raskutti et al. [2012] for function estimation. Define the parameter space:
$$\mathcal{F}_s(r_1^2, \cdots, r_p^2) := \left\{ f = \sum_{j\in[p]} f_j : \quad \sum_{j\in[p]} \mathbf{1}(f_j \neq 0) \leq s, \quad f_j \in \mathcal{H}_j(r_j^2) \cup \{\mathbf{0}\} \text{ for all } j \in [p] \right\},$$
where
$$\mathcal{H}_j(r_j^2) := \left\{ f_j = \sum_{i=1}^{\infty} \theta_{ij}\psi_i : \quad \sum_{i=1}^{\infty} \frac{\theta_{ij}^2}{\mu_{ij}} \leq 1, \quad \sum_{i=1}^{\infty} \theta_{ij}^2 \geq r_j^2 \right\}.$$
Heterogeneity across $\mathcal{H}_j(r_j^2)$ is captured via distinct sequence $\{\mu_{ij}\}_{i\in\mathbb{N}^+}$ for every $j \in [p]$.

**Upper bound**   We apply a component-wise selector

$$\hat{\eta}_j^{heter} := \mathbf{1}\left( n\sum_{i=1}^{K_j} X_{ij}^2 \geq K_j + 2\sqrt{K_j \log(2p/\delta)} + 2\log(2p/\delta) \right),$$

where $K_j := \min\left\{ k \in \mathbb{N}^+ : \mu_{kj} \leq \frac{\sqrt{k\log p}}{n} \right\}$ and recall $n = \sigma^{-2}$. If the condition

$$r_j^2 \geq C_\delta \left\{ \frac{\log p}{n} + \max_{k\in\mathbb{N}^+} \left( \frac{\sqrt{k\log p}}{n} \wedge \mu_{kj} \right) \right\}$$

holds for every $j \in [p]$, we have the exact support recovery guarantee

$$\inf_{f\in\mathcal{F}_s(r_1^2,\cdots,r_p^2)} \mathbf{P}_f\left( \hat{S}(X) = S(f) \right) \geq 1 - \delta.$$

The proof proceeds analogously to that given in Appendix C.2.

**Lower bound**   We next prove that, if there exists some $j \in [p]$ with $r_j^2 \leq c\left\{ \frac{\log p}{n} + \max_{k\in\mathbb{N}^+}\left( \frac{\sqrt{k\log p}}{n} \wedge \mu_{kj} \right) \right\}$ for a sufficiently small constant $c$, then no selector can achieve consistent support recovery.

Define

$$\mathcal{F}_{least,j} := \left\{ f = f_j = \sum_{i=1}^{\infty} \theta_{ij}\psi_i : \quad \theta_{ij} \in \{\lambda_j, -\lambda_j\} \text{ if } i \leq k_j, \ \theta_{ij} = 0 \text{ if } i > k_j \right\},$$

where $\lambda_j, k_j$ will be clarified later. We then design the least favorable set

$$\mathcal{F}_{least} := \left( \bigcup_{j\in[p]} \mathcal{F}_{least,j} \right) \bigcup \{f = 0\}.$$

The set $\mathcal{F}_{least}$ assumes that at most one univariate function could be the support, and the very support has a weak signal strength $r_j^2 = \lambda_j^2 k_j$. Then we get

$$\inf_{\hat{S}} \sup_{f\in\mathcal{F}_{least}} \mathbf{P}_f\left( \hat{S}(X) \neq S(f) \right) \geq \inf_{\hat{S}} \max_{j=0,\cdots,p} \mathbf{P}_j\left( \hat{S}(X) \neq \{j\} \right),$$

where $\mathbf{P}_j$ follows a similar definition in Appendix I.1. Then, if

$$\frac{1}{p}\sum_{j\in[p]} \left( e^{\frac{n^2\lambda_j^4 k_j}{2}} - 1 \right) \leq \frac{\delta^2 p}{2(2-\delta)} - 1,$$

we obtain

$$\inf_{\hat{S}} \sup_{f\in\mathcal{F}_{least}} \mathbf{P}_f\left( \hat{S}(X) \neq S(f) \right) \geq 1 - \delta,$$

for arbitrary constant $\delta \in (0,1)$. Therefore, it suffices to consider the optimization problem

$$\max : \lambda_j^2 k_j,$$

$$\text{subject to: } \exp\left( \frac{n^2\lambda_j^4 k_j}{2} \right) \leq c_1 \cdot p,$$

$$\lambda_j^2 k_j \leq \mu_{K_j},$$

for every $j \in [p]$. The result shows that $r_j^2 = \lambda_j^2 k_j \leq c\max_{k\in\mathbb{N}^+}\left( \frac{\sqrt{k\log p}}{n} \wedge \mu_{kj} \right)$ leads unreliable selection.

On the other hand, by assuming $\sigma^{-2} = n \gtrsim \frac{\log p}{\min_{j\in[p]}\mu_{1j}}$ and following the proof technique in Appendix I.2, we can prove that $r_j^2 \leq c\frac{\log p}{n}$ also leads unreliable selection, therefore we prove the matching lower bound.

# K   Technical Lemma

The following Lemma shows that, if $f \in \mathcal{F}_s^{(K)}(r^2)$, it suffices to only consider those selectors depending on $\|X_{1:K,j}\|_2 = \sqrt{\sum_{i=1}^K X_{ij}^2}$.

**Lemma 1 (Norm selector)** *Assume $K$ is a positive integer satisfying $\mu_K \geq r^2$. For every measurable function $\omega(\cdot, \cdot)$ and for every selector $\hat{\eta}(X_{K \times p}) = (\hat{\eta}_1(X_{K \times p}), \cdots, \hat{\eta}_p(X_{K \times p})) \in \{0, 1\}^p$, there exists a randomized selector $\bar{\eta}(\|X_{\cdot,1}\|_2, \cdots, \|X_{\cdot,p}\|_2)$ such that*

$$\sup_{f \in \mathcal{F}_s^{(K)}(r^2)} \mathbf{E}_f\{\omega(\hat{\eta}, \eta(f))\} \geq \sup_{f \in F_s^{(K)}(r^2)} \mathbf{E}_f\{\omega(\bar{\eta}, \eta(f))\}.$$

*Taking $\omega(x, y) = \mathbf{1}(x \neq y)$, we get*

$$\sup_{f \in \mathcal{F}_s^{(K)}(r^2)} \mathbf{P}_f(\hat{\eta} \neq \eta(f)) \geq \sup_{f \in \mathcal{F}_s^{(K)}(r^2)} \mathbf{P}_f(\bar{\eta} \neq \eta(f)).$$

The proof of Lemma 1 is as similar as the proof of Lemma 1 in Butucea et al. [2023a], with additional checking that $r^2 \leq \|O_j X_{1:K,j}\|_2^2 = \|X_{1:K,j}\|_2^2 \leq \mu_K$ holds for every orthogonal matrix $O_j \in \mathbb{R}^{K \times K}$, every $X_{\cdot,j} \in \mathcal{F}_{1:K}(r^2)$ and every index $j \in \{j : f_j \neq 0\}$.

**Lemma 2 (Chi-squared inequalities)** *Let $\chi_k^2(B)$ denote a $\chi^2$ random variable with $k$ degrees of freedom and non-centrality parameter $B \geq 0$. Then for every $x > 0$, there exist absolute constants $c_1 > 0$ and $c_2 \in (0, 1)$ such that*

$$\mathbf{P}\left(\chi_k^2(0) \geq k + c_1\sqrt{kx} + c_1 x\right) \geq c_2 e^{-x}, \tag{59}$$

$$\mathbf{P}\left(\chi_k^2(0) \leq k - x\right) \leq \exp\left(-\frac{x^2}{4k}\right), \tag{60}$$

$$\mathbf{P}\left(\chi_k^2(B) \geq k + B + 2\sqrt{(k+2B)x} + 2x\right) \leq e^{-x}. \tag{61}$$

*Inequalities (59), (60) and (61) are proved in Corollary 3 in Zhang and Zhou [2020], Theorem 2 in Ghosh [2021] and Lemma 8.1 in Birgé [2001] respectively.*

**Lemma 3 (Minimax lower bound based on combined risk, Abraham et al. [2024])** *Assume we observe $X \sim \mathbf{P}_\theta, \theta \in \mathbb{R}^p$. For any prior $\pi$ (of $\theta$) on $\mathbb{R}^p$, we denote by $\mathbf{P}_\pi$ the distribution of $(X, \theta)$ in the Bayesian model. Then, for all $1 \leq s \leq p$, all $\kappa \in (0, 1)$, and all measurable $\Theta \subset \mathbb{R}^p$, we have:*

$$\inf_\varphi \sup_{\theta \in \Theta} R(\theta, \varphi) \geq \frac{\lambda}{1+\lambda}\left(1 - e^{-c\kappa^2 M}\right) - nP_\pi(\|\theta\|_0 > s) - 2\mathbf{P}_\pi(\theta \notin \Theta),$$

*for some universal constant $c > 0$, where*

$$\lambda = \frac{1-\kappa}{s}\sum_{j=1}^p \mathbf{P}_\pi\left\{\theta_j \neq 0, \ \mathbf{P}_\pi(\theta_j = 0|X) > \frac{1}{2}\right\}.$$

Lemma 3 is derived from Theorem S-3 in the supplementary of [Abraham et al., 2024], with taking $\rho = 1$ and the combined risk

$$R(\theta, \varphi) = \mathbf{E}_\theta\left(\frac{\sum_{j:\theta_j=0}\varphi_j}{1 \vee \sum_{j\in[p]}\varphi_j} + \frac{\sum_{j:\theta_j\neq 0}(1-\varphi_j)}{s}\right).$$

**Lemma 4 (Truncation for classification)** *Assume $C > 0$ is a postive constant and $K^{(C)} := \min\left\{k \in \mathbb{N}^+ : \mu_k \leq C\frac{\sqrt{k \log(p-s)}}{n}\right\}$, then under assumption $n > \frac{C\log(p-s)}{\mu_1}$, we have*

$$\frac{\sqrt{K^{(C)}\log(p-s)}}{n} \asymp \max_{k \in \mathbb{N}^+}\left(\mu_k \wedge \frac{\sqrt{k\log(p-s)}}{n}\right).$$

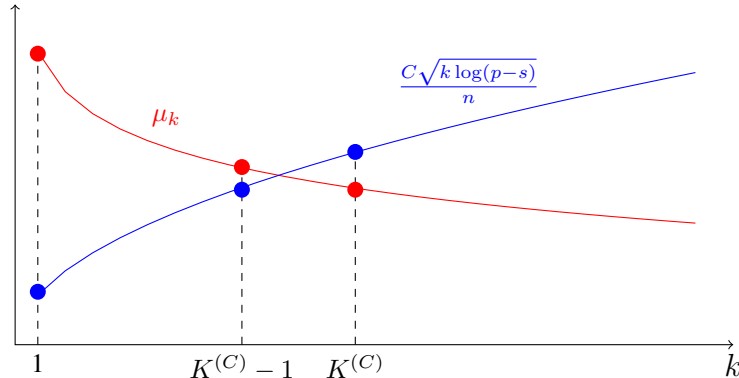

Figure 5: A guide to show $\mu_1 > C\frac{\sqrt{\log(p-s)}}{n}$ and $\frac{C\sqrt{K^{(C)}\log(p-s)}}{n} \geq \max\left\{\mu_{K^{(C)}}, \frac{C\sqrt{(K^{(C)}-1)\log(p-s)}}{n}\right\}$.

**Proof 1 (Proof of Lemma 4)** *The proof is inspired by Lemma 2.1 in Kotekal and Gao [2024]. By taking $K = 1$ and $n > \frac{C\log(p-s)}{\mu_1}$, we have $\mu_1 > \frac{C\log(p-s)}{n} \geq C\frac{\sqrt{\log(p-s)}}{n}$, which shows $K^{(C)} \geq 2$.*

*For every $k \geq K^{(C)}$, by definition of $K^{(C)}$, we conclude $\mu_k \leq \mu_{K^{(C)}} \leq \frac{C\sqrt{K^{(C)}\log(p-s)}}{n} \leq \frac{C\sqrt{k\log(p-s)}}{n}$, which leads that $\mu_k \wedge \frac{C\sqrt{k\log(p-s)}}{n} = \mu_k \leq \mu_{K^{(C)}}$. For every $1 \leq k \leq K^{(C)} - 1$, we conclude $\mu_k \geq \mu_{K^{(C)}-1} > \frac{C\sqrt{(K^{(C)}-1)\log(p-s)}}{n} \geq \frac{C\sqrt{k\log(p-s)}}{n}$, which leads that $\mu_k \wedge \frac{C\sqrt{k\log(p-s)}}{n} = \frac{C\sqrt{k\log(p-s)}}{n} \leq \frac{C\sqrt{(K^{(C)}-1)\log(p-s)}}{n}$. Therefore, we conclude that*

$$\max_{k\in\mathbb{N}^+}\left(\mu_k \wedge \frac{C\sqrt{k\log(p-s)}}{n}\right) = \max\left(\mu_{K^{(C)}}, \frac{C\sqrt{(K^{(C)}-1)\log(p-s)}}{n}\right),$$

*see Figure 5 for a clear demonstration. And it is straightforward that*

$$\frac{C\sqrt{K^{(C)}\log(p-s)}}{n} \geq \max\left(\mu_{K^{(C)}}, \frac{C\sqrt{(K^{(C)}-1)\log(p-s)}}{n}\right). \tag{62}$$

*Besides, from $K^{(C)} \geq 2$, we get $K^{(C)} \leq 2(K^{(C)} - 1)$, therefore*

$$\begin{aligned}
\frac{C\sqrt{K^{(C)}\log(p-s)}}{n} &\leq \frac{C\sqrt{2(K^{(C)}-1)\log(p-s)}}{n} \\
&\leq \sqrt{2}\max\left(\mu_{K^{(C)}}, \frac{C\sqrt{(K^{(C)}-1)\log(p-s)}}{n}\right) \\
&= \sqrt{2}\max_{k\in\mathbb{N}^+}\left(\mu_k \wedge \frac{C\sqrt{k\log(p-s)}}{n}\right)
\end{aligned} \tag{63}$$

*From* (62) *and* (63) *we conclude*

$$\frac{\sqrt{K^{(C)}\log(p-s)}}{n} \asymp \max_{k\in\mathbb{N}^+}\left(\mu_k \wedge \frac{C\sqrt{k\log(p-s)}}{n}\right) \asymp \max_{k\in\mathbb{N}^+}\left(\mu_k \wedge \frac{\sqrt{k\log(p-s)}}{n}\right),$$

*which completes the proof of Lemma 4.*

Following the same proof technique, we can get the following results.

**Lemma 5 (Truncation for joint estimation and multiple testing)** *Assume $C > 0$ is a positive constant and $K^{(C)} := \min\left\{k \in \mathbb{N}^+ : \mu_k \leq C\frac{\sqrt{k\log(p/s)}}{n}\right\}$, then under assumption $n > \frac{C\log(p/s)}{\mu_1}$, we have*

$$\max_{k \in \mathbb{N}^+}\left(\mu_k \wedge \frac{C\sqrt{k\log(p/s)}}{n}\right) \leq \frac{C\sqrt{K^{(C)}\log(p/s)}}{n} \leq \sqrt{2}\max_{k \in \mathbb{N}^+}\left(\mu_k \wedge \frac{C\sqrt{k\log(p/s)}}{n}\right),$$

*which means*

$$\frac{\sqrt{K^{(C)}\log(p/s)}}{n} \asymp \max_{k \in \mathbb{N}^+}\left(\mu_k \wedge \frac{\sqrt{k\log(p/s)}}{n}\right).$$

*Besides, define $K' := \min\left\{k \in \mathbb{N}^+ : \mu_k \leq \frac{\sqrt{k}}{n}\right\}$, then under assumption $n > \frac{1}{\mu_1}$, we have*

$$\max_{k \in \mathbb{N}^+}\left(\mu_k \wedge \frac{\sqrt{k}}{n}\right) \leq \frac{\sqrt{K'}}{n} \leq \sqrt{2}\max_{k \in \mathbb{N}^+}\left(\mu_k \wedge \frac{\sqrt{k}}{n}\right).$$

**Lemma 6 (Expectation with hyperbolic cosine)** *Assume $X \sim N(0, 1/n)$, $k_1, k_2$ are two positive integers, then we have*

$$\left\{\mathbf{E}\cosh\left(nX\lambda(k_1)\right)\cdot\cosh\left(nX\lambda(k_2)\right)\right\}^{k_1 \wedge k_2} \times \left\{\mathbf{E}\cosh\left[nX\cdot(\lambda(k_1)\wedge\lambda(k_2))\right]\right\}^{k_1 \vee k_2 - k_1 \wedge k_2}$$
$$= \cosh^{k_1 \wedge k_2}(n\lambda(k_1)\lambda(k_2)) \times \exp\left(\frac{n}{2}\left\{(k_1 \vee k_2)\cdot\lambda(k_1 \vee k_2)^2 + (k_1 \wedge k_2)\cdot\lambda(k_1 \wedge k_2)^2\right\}\right),$$

*where $\lambda(\cdot)$ can be arbitrary non-increasing function on $\mathbb{N}^+$.*

**Proof 2 (Proof of Lemma 6)** *By $\cosh(x_1)\cosh(x_2) = \frac{1}{2}\left(\cosh(x_1 + x_2) + \cosh(x_1 - x_2)\right)$ and $\mathbf{E}_{x \sim N(0,1)}\cosh(\lambda x) = e^{\lambda^2/2}$, we derive that*

$$\mathbf{E}\cosh\left(nX\lambda(k_1)\right)\cdot\cosh\left(nX\lambda(k_2)\right) = \frac{1}{2}\left\{e^{\frac{n}{2}(\lambda(k_1)+\lambda(k_2))^2} + e^{\frac{n}{2}(\lambda(k_1)-\lambda(k_2))^2}\right\}$$
$$= e^{\frac{n}{2}(\lambda(k_1)^2 + \lambda(k_2)^2)}\cosh(n\lambda(k_1)\lambda(k_2)),$$

*and*

$$\mathbf{E}\cosh\left[nX\cdot(\lambda(k_1)\wedge\lambda(k_2))\right] = e^{\frac{n}{2}(\lambda(k_1)\wedge\lambda(k_2))^2}.$$

*Therefore*

$$\left\{\mathbf{E}\cosh\left(nX\lambda(k_1)\right)\cdot\cosh\left(nX\lambda(k_2)\right)\right\}^{k_1 \wedge k_2} \times \left\{\mathbf{E}\cosh\left[nX\cdot(\lambda(k_1)\wedge\lambda(k_2))\right]\right\}^{k_1 \vee k_2 - k_1 \wedge k_2}$$
$$= \cosh^{k_1 \wedge k_2}(n\lambda(k_1)\lambda(k_2))$$
$$\times \exp\left(\frac{n}{2}\left\{(\lambda(k_1)^2 + \lambda(k_2)^2)\cdot(k_1 \wedge k_2) + \lambda(k_1 \vee k_2)^2\cdot(k_1 \vee k_2 - k_1 \wedge k_2)\right\}\right)$$
$$= \cosh^{k_1 \wedge k_2}(n\lambda(k_1)\lambda(k_2)) \times \exp\left(\frac{n}{2}\left\{(k_1 \vee k_2)\cdot\lambda(k_1 \vee k_2)^2 + (k_1 \wedge k_2)\cdot\lambda(k_1 \wedge k_2)^2\right\}\right),$$

*which completes the proof of Lemma 6.*

**Lemma 7 (Minimax lower bound in $\chi^2$ divergence)** *For a given constant $\delta \in (0, 1)$, assume that $\mathbf{P}_0, \mathbf{P}_1, \cdots, \mathbf{P}_p$ be $p$ probability measures satisfying $p \geq 4/\delta^2$ and*

$$\frac{1}{p}\sum_{j=1}^{p}\chi^2\left(\mathbf{P}_j\|\mathbf{P}_0\right) \leq \frac{\delta^2 p}{2(2-\delta)} - 1.$$

*Then*

$$\inf_{\psi}\sup_{0 \leq j \leq p}\mathbf{P}_j(\psi \neq j) \geq (1 - \delta/2)^2 > 1 - \delta,$$

*where $\inf_{\psi}$ represents the infimum over all tests of the form $\psi : X \to \{0, 1, \cdots, p\}$ with $X \sim \mathbf{P}_j$.*

The proof of Lemma 7 follows from Proposition 2.4 of Tsybakov [2009] with taking $M = p$, $\alpha_* = \frac{\delta^2 p}{2(2-\delta)} - 1$ and $\tau = \frac{2-\delta}{\delta p}$.

