# OpenReview forum: "Minimax-Optimal Univariate Function Selection in Sparse Additive Models: Rates, Adaptation, and the Estimation-Selection Gap"
_NeurIPS.cc/2025/Conference — NeurIPS 2025 spotlight_

### Official Review · Reviewer_oppP · 2025-06-16

**Clarity:** 4
**Significance:** 3
**Originality:** 3
**Rating:** 5
**Confidence:** 4

**Summary:**

The authors study variable selection in sparse additive models and establish minimax separation rates under both Hamming loss and a multiple testing loss (FDR + FNR). They obtain sharp results and show the problem is fundamentally different from the problem of estimating the model. Specializing to the case of $\alpha$-smooth component functions, the authors also address adaptation to $\alpha$ and demonstrate an unavoidable cost in the rate.

**Questions:**

- Can the authors address adaptation to sparsity? Currently, the variable selection procedure under the FDR + FNR loss requires choosing the truncation $K$ depending on $s$. The sparsity level $s$ may not typically be known. Can adaptation be done here? Is there some unavoidable cost that must also be paid?

- Can the authors address adaptation more generally besides just Sobolev space? A major finding of the work by Kotekal and Gao cited in the paper was that, in the signal detection problem, there are different adaptation costs depending on the underlying function space. Some spaces impose an unavoidable cost, whereas others do not. Is there a similar phenomenon in the variable selection problem?

- The selectors the authors study are essentially component-wise. Can the authors handle the more general setting where the component functions might live in different spaces? For example, $f_j \in \mathcal{H}_j$ where the $\mathcal{H}_j$ might be different across $j$? This is the setting considered by Raskutti, Wainwright, and Yu.

- Line 769: "taht" should be "that"

**Ethical Concerns:**

["NO or VERY MINOR ethics concerns only"]

**Final Justification:**

My questions have been addressed and I retain my score.

**Limitations:**

Yes

**Paper Formatting Concerns:**

I noticed no issues.

**Quality:**

4

**Strengths And Weaknesses:**

__Strengths__
Sparse additive models are a popular and important topic in machine learning, so these results are quite welcome. The variable selection problem is quite important, and the authors obtain sharp theoretical results of a very fundamental nature. Furthermore, the setting is fairly general, allowing the component functions to live in a generic RKHS (not just e.g. Sobolev space).

__Weaknesses__
There are no major weaknesses. One might nitpick and say the white noise model is too abstract, but the paper's focus is theoretical and there is a long tradition in the statistics literature of relying on asymptotic equivalence to focus on the essence.

---

> ### Author Rebuttal · Authors · 2025-07-31
>
> We sincerely appreciate your recognition of our results on minimax optimal variable selection and the general function space setting. Below, we will provide a response that centers around weaknesses and questions related to our paper.
>
> # Weaknesses:
>
> ## 1. One might nitpick and say the white noise model is too abstract, but the paper's focus is theoretical and there is a long tradition in the statistics literature of relying on asymptotic equivalence to focus on the essence.
>
> **Reply:** Thank you for recognizing the theoretical focus of our work. We agree that the Gaussian white noise model provides an abstract but powerful framework to study the fundamental statistical limits of our problem. This approach aligns with our goal of delivering rigorous theoretical insights.
>
> # Questions:
>
> ## 1. Can the authors address adaptation to sparsity? Currently, the variable selection procedure under the FDR + FNR loss requires choosing the truncation $K$ depending on $s$. The sparsity level $s$ may not typically be known. Can adaptation be done here? Is there some unavoidable cost that must also be paid?
>
> **Reply:** We demonstrate that Lepski's method can be used to adapt to the unknown sparsity level $s$: Suppose we know an upper bound $M\ge s$ with $p/M\ge C$ for some constant $C>1$. Define the dyadic grid $\mathcal S := \left\\{1,2,4, \dots,2^{\lfloor\log_2M\rfloor}\right\\}$. For each $s\_m \in \mathcal S$, define
> $$
> \tilde \eta^{s\_m}\_j := \mathbf1 \left(n \sum\_{i=1}^{K(s_m)} X\_{ij}^2 \ge K(s_m) + 2\sqrt{K(s_m)\cdot \log \left( \frac{8p}{s_m\delta} \right)} + 2\log\left( \frac{8p}{s_m\delta} \right) \right),
> $$
> where $K(s_m)=\min\left\\{k\in\mathbb N^+: \mu_k^2\le\tfrac{k\log(p/s_m)}{n^2}\right\\}$. Then we consider the adaptive selector $\tilde\eta^{\hat s} = \left(\tilde\eta^{\hat s}\_1, \cdots, \tilde\eta^{\hat s}\_p \right) \in \{0,1\}^p$, with
> $$
> \hat s = \min\left\\{ s_m \in \mathcal S: \sum\_{j\in [p]} |\tilde \eta_j^{s_m} - \tilde \eta_j^{s_\ell}| \le \delta \cdot s_\ell, \text{    for all } s_\ell\in \mathcal S \text{ and }s_\ell \ge s_m \right\\}.
> $$
> By following the proof procedure of Theorem 1 in our manuscript and Theorem 5.2 in Butucea et al.[1], as long as $c \le (\log s)/(\log p)  \le C$ for two constants $0<c<C<1$, this adaptive selector $\tilde\eta^{\hat s}$ achieves the same minimax separation rate under FDR+FNR loss, with no additional cost for not knowing $s$ in advance.
>
> We can also understand this conclusion from another perspective: when $0<c \le (\log s)/(\log p) \le C<1$, we have $\log(p/s) \asymp \log p$. Hence, one may simply replace $\log(p/s)$ with $\log p$ in the selector $\hat \eta_j^{test}$ (in equation (6)) to adapt to the unknown $s$, while preserving the minimax rate-optimal signal strength requirement.
>
>
> ## 2. Can the authors address adaptation more generally besides just Sobolev space? A major finding of the work by Kotekal and Gao cited in the paper was that, in the signal detection problem, there are different adaptation costs depending on the underlying function space. Some spaces impose an unavoidable cost, whereas others do not. Is there a similar phenomenon in the variable selection problem?
>
> **Reply:** Thank you for this question. There are two types of parameters that require adaptation, and we discuss each of them separately:
>
> **Adaptation to sparsity $s$.** As shown in our previous reply, whenever  $0<c \le (\log s)/(\log p) \le C<1$, by applying Lepski's method, we could provide a procedure adaptive to unknown $s$ without any additional rate cost, regardless of the underlying function space.
>
> **Adaptation to $\\{\mu\_k\\}\_{k \in \mathbb N^+}$.**  In Section 4 we handle Sobolev ellipsoids ($\mu_k=k^{-2\alpha}$) with unknown smoothness parameter $\alpha>0$, showing our adaptive truncation remains minimax rate-optimal and there is an unavoidable $\log\log n$ term as the cost for unknown $\alpha$. Similarly, for the finite dimension or exponential decay ellipsoids, the adaptation cost (for the unknown smoothness parameter) should be worked out on a case-by-case basis. And we believe our adaptive selector from equation (12) provides a valuable blueprint for these specific settings.
>
> Finally, we provide a remark: the work by Kotekal and Gao[2] examined: **(i)** the adaptation cost for unknown sparsity $s$ in different underlying function spaces (their Section III), and **(ii)** the adaptation cost for unknown Sobolev smoothness $\alpha$ (their Section IV). They did not address the general problem of adapting to an arbitrary sequence $\\{\mu\_k\\}\_{k \in \mathbb N^+}$, underscoring that the adaptation to smoothness should be analyzed on a case-by-case basis. Moreover, because their focus is on testing $f\equiv0$ while we address support set recovery of nonzero components $f_j$, the incurred adaptation costs are inherently different.
>
>
> ## 3. The selectors the authors study are essentially component-wise. Can the authors handle the more general setting where the component functions might live in different spaces? For example, $f_j \in \mathcal H_j$  where the $\mathcal H_j$ might be different across $j$? This is the setting considered by Raskutti, Wainwright, and Yu.
>
> **Reply:** Thank you for this insightful question. We can indeed extend our framework to heterogeneous settings where $f_j \in \mathcal H_j$ and $\mathcal H_j$ might be different across $j \in [p]$. Define the parameter space:
> $$
> \mathcal F\_s (r_1^2,\cdots ,r_p^2) :=  \left\\{ f = \sum\_{j \in [p]} f\_j :\quad  \sum\_{j\in [p]}\mathbf 1(f\_j \ne 0) \le s, \quad f_j \in \mathcal H\_j(r\_j^2)\cup\\{ \mathbf 0\\} \text{ for all } j\in [p] \right\\},
> $$
> where
> $$
> \mathcal H_j(r_j^2) := \left\\{ f_j=\sum\_{i=1}^\infty \theta_{ij} \psi_i:\quad  \sum\_{i =1}^\infty \frac{\theta_{ij}^2}{\mu_{ij}}\le 1 ,\quad   \sum\_{i =1}^\infty \theta_{ij}^2 \ge r_j^2 \right\\}.
> $$
> Heterogeneity across $\mathcal H_j(r_j^2)$ is captured via distinct sequence $\\{\mu\_{ij} \\}\_{i\in\mathbb N^+}$for every $j \in [p]$.
>
> **Upper bound.** We apply a component‑wise selector
> $$
> \hat \eta_{j}^{heter} := \mathbf1 \left(n \sum_{i=1}^{K_j} X_{ij}^2 \ge K_j + 2\sqrt{K_j \log (2p/\delta )} + 2\log (2p/\delta ) \right),
> $$
> where $K_j := \min \left\\{k \in \mathbb N^+: \mu_{kj} \le \frac{\sqrt{k \log p}}{n} \right\\}$.
> If the condition $r_j^2 \ge C_\delta \left[ \frac{\log p }{n} + \max\_{k \in \mathbb N^+} \left(\frac{\sqrt{k\log p}}{n} \wedge \mu_{kj}  \right) \right]$ holds for every $j \in [p]$, we have the exact support recovery guarantee
> $$
> \inf\_{f\in \mathcal F_{s}(r_1^2,\cdots ,r_p^2)} \mathbf P_f \left(\hat S(X) = S(f) \right) \ge 1-\delta.
> $$
> The proof proceeds analogously to that given in Appendix C.2.
>
> **Lower bound.** If there exists some $j\in[p]$ with $r_j^2 \le c \left[ \frac{\log p }{n} + \max\_{k \in \mathbb N^+} \left(\frac{\sqrt{k\log p}}{n} \wedge \mu_{kj}  \right) \right]$ for a sufficiently small constant $c$, then no selector can achieve consistent support recovery. This lower bound proof follows Theorem 5's framework, but requires a finer control of chi-square divergences due to component-dependent truncation levels and signal strengths settings.
>
>
> ## 4. Line 769: "taht" should be "that".
>
> **Reply:** Thank you for identifying the typo, and we have corrected it accordingly.
>
>
> **Reference**
>
> [1] Cristina Butucea, Mohamed Ndaoud, Natalia A. Stepanova, and Alexandre B. Tsybakov. Variable selection with Hamming loss. The Annals of Statistics, 46(5):1837-1875, 2018.
>
> [2] Subhodh Kotekal and Chao Gao. Minimax signal detection in sparse additive models. IEEE Transactions on Information Theory, 70(12):8892–8928, 2024.

---

### Official Review · Reviewer_FqYP · 2025-06-20

**Clarity:** 4
**Significance:** 3
**Originality:** 3
**Rating:** 5
**Confidence:** 4

**Summary:**

The paper addresses the problem of variable selection in the nonparametric Gaussian white noise model, where the unknown signal is assumed to be a smooth multivariate function. This function is modeled as a sum of a small number of univariate components, each belonging to an ellipsoid defined via a quadratic constraint on its Fourier coefficients. The proposed selection procedure relies on (group) thresholding the Fourier coefficients of the noisy observed signal. The authors establish the minimax rates of separation for detecting the relevant variables. Additionally, they explore the case of unknown smoothness and propose an adaptive detection strategy.

**Questions:**

Can the proposed detection methods be easily adapted to deal with the case of unknown $n$ ?

**Ethical Concerns:**

["NO or VERY MINOR ethics concerns only"]

**Final Justification:**

I find the paper interesting and appreciate the authors’ responses to my remarks and questions.

**Limitations:**

yes

**Quality:**

3

**Strengths And Weaknesses:**

**Strengths**

1. The problem of variable selection is one of the central challenges in nonparametric statistics and learning theory.
2. The approach based on establishing minimax rates of separation is a highly relevant and principled way of quantifying the difficulty of the task.
3. The paper establishes matching upper and lower bounds, which significantly strengthens the theoretical contribution.
4. The writing is clear and the paper is pleasant to read.

**Weaknesses**

1. While this paper would be a perfect fit for a statistics journal, I am concerned that only a small portion of the NeurIPS audience will find these results directly relevant.
2. The noise level is assumed to be known and fixed at $1/\sqrt{n}$. A more general and practically relevant setting would allow the noise level to be an unknown parameter $\sigma$.

**Minor Remarks**

1. Some important references are missing:
   - [Comminges and Dalalyan, AoS 2012] and [Min Xu, Minhua Chen, John Lafferty, AoS 2016] studied variable selection in the same problem under different conditions.
   - [Dalalyan, Ingster, Tsybakov, PTRF 2014] and [Tyagi et al., AISTATS 2016] considered a closely related model where the components of the additive model are not necessarily univariate. These works derived minimax rates for estimation.

2. In the main theorems, the phrasing ''for every $\delta$'' should be revised. As currently written, it may suggest that the assumptions on $n$ and other parameters must hold uniformly for all $\delta$. A clearer formulation would be to begin the theorem with: "Let $\delta$ be an arbitrary number in..." and then state the result. This makes it explicit that $\delta$ is fixed within the statement.

---

> ### Author Rebuttal · Authors · 2025-07-31
>
> We sincerely appreciate your recognition of the minimax optimal variable selection and your positive remarks on the clarity of our presentation.
> Below, we will provide a response that centers around weaknesses, remarks, and questions related to our paper.
>
> # Weaknesses:
>
> ## 1. While this paper would be a perfect fit for a statistics journal, I am concerned that only a small portion of the NeurIPS audience will find these results directly relevant.
>
> **Reply:** Thank you for this feedback. We think our work offers several insights of direct interest to the NeurIPS audience:
>
> **Information-theoretic limits.** We derive minimax lower bounds for support set recovery in SpAM, showing that no algorithm---including modern deep-learning methods---can reliably select variables once the true signal strengths are below the minimax separation rate (see Theorems 2,3, and 5). This key finding can guide ML researchers in understanding the necessity of the signal strength requirement.
>
> **Adaptation.** In Section 4, we propose a selector that is adaptive to the unknown smoothness parameter $\alpha>0$ in Sobolev classes $\mu_k = k^{-2\alpha}$, and quantify the extra $\log\log n$ cost for adaptation. Our adaptive method and analytical framework can inspire principled hyperparameter tuning in some more complex models.
>
> **Trade-off between estimation and selection.** In Theorem 7, we prove that some estimators that are optimal for function estimation can be suboptimal for variable selection. This offers a negative but objective suggestion to ML researchers studying generalization, model selection, and interpretability: good prediction cannot always guarantee good variable selection.
>
> Additionally, in Appendix A, we discuss the variable selection problem in SpAM under local differential privacy constraints, demonstrating the extensibility of our work. We hope these points highlight the relevance and potential impact of this paper for the broader NeurIPS audience.
>
>
> ## 2. The noise level is assumed to be known and fixed at $\sqrt{1/n}$. A more general and practically relevant setting would allow the noise level to be an unknown parameter $\sigma$. **And also from Question Part:** Can the proposed detection methods be easily adapted to deal with the case of unknown $n$ ?
>
> **Reply:** We can replace $\sqrt{1/n}$ with a general noise level $\sigma$ to obtain more flexible statements and results. For instance, the selector (equation (10)) in Remark 3 then becomes
> $$
> \hat \eta_j^{rec} (X) = \mathbf1 \left( \frac1{\sigma^2} \sum_{i=1}^{K'} X_{ij}^2 \ge K'  + 2\sqrt{K'\log(2p/\delta)} + 2 \log(2p/\delta) \right)  , \text{ for every } j \in [p],
> $$
> where $K' := \min \left\\{k \in \mathbb N^+: \mu_k \le \sigma^2 \sqrt{k \log p} \right\\}$. Under this formulation, the minimax separation rate for support recovery (Theorem 3) becomes $ \sigma^2 \log p + \max\_{k \in \mathbb N^+} \left\\{ \left(\sigma^2\sqrt{k\log p}\right)\wedge \mu_k  \right\\}$. In particular, for a Sobolev ellipsoid $\mu_k\asymp k^{-2\alpha}$, it simplifies to $\epsilon\_{rec}^2 \asymp \sigma^2 \log p + \sigma^{\frac{8\alpha}{1+4\alpha}} (\log p)^{\frac{2\alpha}{1+4\alpha}}$.
>
> **Adaptation to unknown $\sigma$.** We now outline a simple approach to adapt to the unknown $\sigma$. Denote $X_{1\cdot}:=(X_{11},\cdots, X_{1p}) \in \mathbb R^p$, and $ \theta_{1\cdot} := \mathbf E (X_{1\cdot}) = (\theta_{11}, \cdots, \theta_{1p}) \in \mathbb R^p$. Under the SpAM setting, the vector $\theta_{1\cdot}$ is $s$-sparse. We then fit a Square-Root Lasso estimator (Belloni et al.[1])
> $$
> \hat\theta\_{1\cdot} = \arg\min\_{\gamma \in \mathbb R^p} \left\\{ \\|X\_{1\cdot}- \gamma \\|\_2 + C\sqrt{\frac{\log p}{p}} \\| \gamma\\|\_1 \right\\} .
> $$
> With a probability greater than $1-O(e^{-c s\log p})$, one has $\\|\hat\theta\_{1\cdot} - \theta\_{1\cdot} \\|\_2^2 \lesssim \sigma^2 s\log p$ and $\\|\hat\theta\_{1\cdot} - \theta\_{1\cdot} \\|\_1 \lesssim \sigma s \sqrt{\log p}$ (Derumigny[2]). We then construct an estimator of $\sigma$ as $\hat\sigma = \frac{\\|X\_{1\cdot}- \hat\theta\_{1\cdot}\\|\_2}{\sqrt{p}}$. Following the proof of Proposition 2 in Carpentier et al.[3], with a probability greater than $1-O(e^{-c s\log p})$, one has $\bigl|\hat\sigma^2/\sigma^2 - 1\bigr|\lesssim\sqrt{(s\log p)/p} $. Hence, as long as $\sigma^2\log p\lesssim\mu_1$ and $K'\lesssim \max\left( \frac{p}{s},  \sqrt{\frac{p\log p}{s}} \right)$, we can replace $\sigma$ by $\hat\sigma$ in our selector and retain the theoretical guarantees.
>
> We will include the above analysis in the appendix. And we hope this addresses your concern about the noise level assumption.
>
>
> # Minor Remarks:
>
> ## 1. Some important references are missing: [Comminges and Dalalyan, AoS 2012] and [Min Xu, Minhua Chen, John Lafferty, AoS 2016] studied variable selection in the same problem under different conditions. [Dalalyan, Ingster, Tsybakov, PTRF 2014] and [Tyagi et al., AISTATS 2016] considered a closely related model where the components of the additive model are not necessarily univariate. These works derived minimax rates for estimation.
>
> **Reply:** Thank you for highlighting these important works. We have now cited all four papers in Sections 1 and 2 of the revised manuscript. Dalalyan et al.[4] and Tyagi et al.[5] studied function estimation and variable selection in sparse additive models that permit low-dimensional interactions. Comminges and Dalalyan[6] and Xu et al.[7] analyzed support recovery in a $d$-dimensional nonparametric regression with an intrinsic $s$-variate underlying function. Additionally, the framework in Comminges and Dalalyan[6] can also be interpreted as an additive model allowing $s$-dimensional interaction (but there is only ONE $s$-variate support function).
>
> ## 2. In the main theorems, the phrasing ''for every $\delta$'' should be revised. As currently written, it may suggest that the assumptions on $n$ and other parameters must hold uniformly for all $\delta$. A clearer formulation would be to begin the theorem with: "Let $\delta$ be an arbitrary number in..." and then state the result. This makes it explicit that $\delta$ is fixed within the statement.
>
> **Reply:** We appreciate your rigorous review. As you pointed out, we first fix a $\delta \in (0,1)$ before deriving the corresponding theoretical results. We have revised the manuscript to clarify this point.
>
>
> **Reference**
>
> [1] A. Belloni, V. Chernozhukov, L. Wang. Square-root lasso: pivotal recovery of sparse signals via conic programming. Biometrika, 98(4):791–806, 2011.
>
> [2] Alexis Derumigny. Improved bounds for Square-Root Lasso and Square-Root Slope. Electronic Journal of Statistics, 12(1): 741-766, 2018.
>
> [3] Alexandra Carpentier, Olivier Collier, Laetitia Comminges, Alexandre B. Tsybakov, Yuhao Wang. Estimation of the $\ell_2$-norm and testing in sparse linear regression with unknown variance. Bernoulli, 28(4):2744-2787, 2022.
>
> [4] Arnak Dalalyan, Yuri Ingster, and Alexandre B Tsybakov. Statistical inference in compound functional models. Probability Theory and Related Fields, 158(3):513–532, 2014.
>
> [5] Hemant Tyagi, Anastasios Kyrillidis, Bernd Gärtner, and Andreas Krause. Learning sparse additive models with interactions in high dimensions. volume 51 of Proceedings of Machine Learning Research, 111–120, 2016.
>
> [6] Laëtitia Comminges and Arnak S. Dalalyan. Tight conditions for consistency of variable selection in the context of high dimensionality. The Annals of Statistics, 40(5):2667 – 2696, 2012.
>
> [7] Min Xu, Minhua Chen, and John Lafferty. Faithful variable screening for high-dimensional convex regression. The Annals of Statistics, 44(6):2624 – 2660, 2016.

---

> > ### Comment · Reviewer_FqYP · 2025-08-04
> > **Acknowledgment of receipt and review of the response**
> >
> > Thank you for your detailed response. It is clear that nonparametric statistical techniques exist that allow the construction of procedures adaptive to the noise level $\sigma$. I found your response to this question somewhat surprising, as it suggests that such adaptivity comes at no additional cost (at least from a statistical perspective) and incurs only a computational overhead. That said, this point does not affect my evaluation of the manuscript, and I maintain my score, as I consider the paper to be an interesting and valuable contribution to nonparametric statistics.
> >
> > I would not recommend that the authors incorporate adaptivity into the current version of the paper, as doing so would introduce new results that would not undergo peer review. However, introducing a multiplicative factor $\sigma$ in the noise and assuming it is known seems worthwhile, as it likely does not require any new mathematical arguments.

---

> > > ### Author Response · Authors · 2025-08-04
> > > **Thank you for your review！**
> > >
> > > Thank you for your insightful feedback and for maintaining your score. We fully agree that introducing a known multiplicative factor $\sigma$ in our model is not only straightforward but also enhances the manuscript’s readability. We will include it in the revised manuscript.

---

### Official Review · Reviewer_8MUY · 2025-06-24

**Clarity:** 4
**Significance:** 3
**Originality:** 3
**Rating:** 5
**Confidence:** 3

**Summary:**

This work establishes minimax separation rates for sparse additive models with Gaussian white noise, from the perspectives of sparse multiple testing and support recovery.

**Questions:**

1. Since variable selection in additive models is well-studied, could you elaborate on the proposed truncation method's strengths over existing approaches, such as its ease of tuning?

2. In Corollary 1 for the finite-dimensional case, the setting resembles the detection boundary in signal testing. Could you derive a more specific bound on the relationship between $\epsilon_{\text{test}}$ and the number of non-zero components $m$, together with $\mu_1>0$?

3. For the selector's truncation set  $\mathcal{K}_{rec}$  that adapts to unknown smoothness parameters, can bases other than 2 be chosen (e.g., base 3)? What is the essential impact of using base 2?

**Ethical Concerns:**

["NO or VERY MINOR ethics concerns only"]

**Final Justification:**

I am satisfied with this paper. The authors have thoroughly addressed all the further questions I raised regarding the work, providing clear and insightful responses that effectively resolved my concerns. The rebuttal is also of high quality and I have decided to maintain my original score.

**Limitations:**

Yes

**Quality:**

3

**Strengths And Weaknesses:**

Strengths: 1. Establishes minimax rates that integrate smoothness conditions, providing theoretical bounds for sparse additive models under Gaussian noise.  2. Proposes adaptive schemes to accommodate varying smoothness levels, enhancing practical applicability.   3. Clarifies key distinctions between optimal estimation and variable selection to avoid conceptual confusion.

Weaknesses:  While variable selection in additive models is a well-studied, the study lacks experimental comparisons between the proposed truncation method and existing selection approaches.

---

> ### Author Rebuttal · Authors · 2025-07-31
>
> We sincerely appreciate your recognition of the minimax separation rates, adaptation, and the key distinctions between estimation and selection.
> Below, we will provide a response that centers around weaknesses and questions related to our paper.
>
> # Weaknesses:
>
> ## 1. While variable selection in additive models is a well-studied, the study lacks experimental comparisons between the proposed truncation method and existing selection approaches.
>
> **Reply:** While we did not directly benchmark against other existing selection approaches, Appendix B.2 provides a comprehensive comparison of four methods based on truncation:
>
> **(i)** the rate-optimal selector proposed in Section 3 (Optimal method);
>
> **(ii)** the adaptive selector proposed in Section 4 (Adaptation method);
>
> **(iii)** the selector based on the optimal truncation in the single univariate function detection problem (Baraud[1], Univariate method);
>
> **(iv)** the selector based on the optimal truncation in the function estimation problem (Birgé and Massart[2], Suboptimal method).
>
> Under both FDR+FNR loss and Hamming loss, the “Optimal” and “Adaptation” methods maintain the lowest selection errors across most regimes. These simulations also confirm the finding from Section 5 that a minimax-optimal function estimation may fail to guarantee optimal variable selection.
>
> Moreover, Appendix B.1 examines the impact of dimension $p$ and signal strength $r^2$, and Appendix B.3 examines the impact of unknown smoothness parameters. In each case, the experimental results align closely with our theoretical results. We hope these comparisons could address your concerns.
>
> # Questions:
>
> ## 1. Since variable selection in additive models is well-studied, could you elaborate on the proposed truncation method's strengths over existing approaches, such as its ease of tuning?
>
> **Reply:** Our truncation-based selector offers three key advantages:
>
> **Minimax optimality.** In Section 3, we establish that our method achieves consistent variable selection precisely at the minimax separation rate, which is the necessary and sufficient signal strength for reliable selection. In contrast, most existing methods require signal levels substantially above the minimax separation rate (Dai et al.[3]).
> Additionally, in Section 5, we demonstrate that some function-estimation-based selectors---even achieving minimax optimal estimation---may fail to attain optimal variable selection performance, thereby underscoring the superiority of our direct-selection-based procedure.
>
> **Adaptation to smoothness.** In Section 4, we consider the Sobolev classes $\mu_k=k^{-2\alpha}$ with smoothness parameter $\alpha>0$, and derive the minimax separation rate when $\alpha$ is unknown. Our adaptive method remains rate-optimal without prior knowledge of $\alpha$, thereby demonstrating adaptive robustness unavailable in most existing methods.
>
> **Ease of tuning and extensibility.** Our algorithm involves only two tuning parameters: the truncation location and the universal threshold, making it more straightforward to calibrate than those penalty-based methods. Besides, our method is built on the fundamental concept of hard-thresholding operator, which we believe endows our method with broad extensibility (see Appendix A).
>
>
> ## 2. In Corollary 1 for the finite-dimensional case, the setting resembles the detection boundary in signal testing. Could you derive a more specific bound on the relationship between $\epsilon_{test}$ and the number of non-zero components $m$, together with $\mu_1>0$?
>
> **Reply:** For clarity, we set $\mu_1 =\cdots = \mu_m>0$, and $\frac{\log(p/s)}n \le \mu_1$. Then we discuss the minimax separation rate of the finite dimension case in two regimes:
>
> **(a)** If $\mu_1 \lesssim \frac{\sqrt{m \log(p/s)}}{n}$, then we derive that $\mu_1\vee \frac{ \log(p/s)}{n} \lesssim \frac{\sqrt{m \log(p/s)}}{n}$, leading $m \gtrsim n\mu_1$. In this case $\epsilon_{test}^2 \asymp \frac{ \log (p/s)}{n} + \mu_1 \asymp \mu_1$.
>
> **(b)** If $\mu_1 \succ \frac{\sqrt{m \log(p/s)}}{n}$, then we get $\epsilon_{test}^2 \asymp \frac{ \log (p/s)}{n} + \frac{\sqrt{m \log (p/s)}}{n}$, which aligns with the minimax separation rate in the group sparsity setting (Butucea et al.[4]).
>
> Combining these cases gives a more precise separation rate $\epsilon_{test}^2 \asymp \min\left(\frac{ \log (p/s)+\sqrt{m \log (p/s)}}{n} ,~ \mu_1\right)$. Here the minimum reflects our ellipsoid space constraint: Since $\\|f_j \\|_2^2 = \sum\_{i \in \mathbb N^+} \theta\_{ij}^2 \le \mu_1 \sum\_{i \in \mathbb N^+} \frac{\theta\_{ij}^2}{\mu_i} \le \mu_1$, each $f_j$ has its squared $L_2$ norm upper bounded by $\mu_1$.
>
> In addition, there are two remarks:
>
> **Remark I.** In case (a) we have $\epsilon_{test}^2\asymp \mu_1$. In other words, to control FDR+FNR, we would need every active univariate function $f_j$ to satisfy $\\|f\_j\\|\_2\ge C\mu_1$ for some large constant $C>0$. However, by definition of $\mathcal E$ and $\mathcal F_s(r^2)$ (in Section 2.1), every $f_j$ obeys $\\|f\_j\\|\_2\le \mu_1$. Hence, there are basically no $f_j$ that can attain a detectable norm, and the support recovery problem is essentially trivial in this case.
>
> **Remark II.** To ensure our results (e.g., equations (8) and (9)) hold for any choice of smoothness sequence $\\{\mu_k\\}\_{k \in \mathbb N^+}$, the constants in our separation rates are not optimized for sharpness. In contrast, several variable selection studies derived asymptotically exact constants (Butucea et al.[5]) and showed the phase diagrams of signal strength for support recovery (Jin and Zheng[6]). We conjecture that, by employing more refined asymptotic techniques, our results can similarly attain constant-sharp minimax optimality.
>
>
> ## 3. For the selector's truncation set $\mathcal K_{rec}$ that adapts to unknown smoothness parameters, can bases other than 2 be chosen (e.g., base 3)? What is the essential impact of using base 2?
>
> **Reply:** Thank you for this question. Using base 2 is purely for notational convenience---actually any constant base $a>1$ (e.g., $3,4,\cdots$) can be used as long as $\log_a n \asymp \log_e n$. Essentially, the process of our adaptive method is to: **(i)** build a representative and geometric grid $\mathcal K_{rec}$ of candidate truncation levels $K$, **(ii)** compute the selectors corresponding to each $K$, and **(iii)** take the maximum over them for each coordinate (see equation (12)). This procedure remains minimax rate-optimality provided the common ratio of the geometric grid is any absolute constant greater than 1.
>
>
> **Reference**
>
> [1] Yannick Baraud. Non-asymptotic minimax rates of testing in signal detection. Bernoulli, 8(5): 577–606, 2002.
>
> [2] Birgé, Lucien, and Pascal Massart. Gaussian model selection. Journal of the European Mathematical Society 3(3): 203-268, 2001.
>
> [3] Xiaowu Dai, Xiang Lyu, and Lexin Li. Kernel knockoffs selection for nonparametric additive models. Journal of the American Statistical Association, 118(543):2158–2170, 2023.
>
> [4] Cristina Butucea, Enno Mammen, Mohamed Ndaoud, and Alexandre B. Tsybakov. Variable selection, monotone likelihood ratio and group sparsity. The Annals of Statistics, 51(1):312–333, 2023.
>
> [5] Cristina Butucea, Mohamed Ndaoud, Natalia A. Stepanova, and Alexandre B. Tsybakov. Variable selection with Hamming loss. The Annals of Statistics, 46(5):1837–1875, 2018.
>
> [6] Jin, Jiashun, and Zheng Tracy Ke. Rare and weak effects in large-scale inference: methods and phase diagrams. Statistica Sinica, 26(1):1-34, 2016.

---

> > ### Comment · Reviewer_8MUY · 2025-08-05
> > **Thank you.**
> >
> > Thanks for responding. Your rebuttal help me clarify your paper.

---

> > > ### Author Response · Authors · 2025-08-05
> > > **Thank you for your review！**
> > >
> > > Thank you for your time and for reading our response. We are glad to hear it helped clarify this paper. We appreciate your feedback and the opportunity to improve our work.

---

### Official Review · Reviewer_BfNF · 2025-07-05

**Clarity:** 4
**Significance:** 2
**Originality:** 2
**Rating:** 4
**Confidence:** 4

**Summary:**

This paper studies the model selection problem in sparse additive model (SpAM), under a gaussian white noise (GWN) observation model. More specifically, it assumes that the dependence of the response on each of the relevant coordinates is modeled by a function belonging to an ellipsoid, and studies the problem of determining which coordinates of the input vector affect the output.
Under a separation assumption between relevant and irrelevant coordinates, it determines minimax rates for this model selection problem.

**Questions:**

General question:
- Can you describe some concrete examples in which this exact recovery criterion (and the associated minimum strength assumption) are appropriate, and the work presented here provides some insights?

A couple of minor points:
-Above the (speculation) formula in Section 2.2. Here you did not yet define formally the problem, so it is unclear what does this speculation refer to.
-Above Theorem 1: Please avoid "sophisticated analysis", this is a judgement that you are expressing about your own work.

**Ethical Concerns:**

["NO or VERY MINOR ethics concerns only"]

**Limitations:**

Yes.

**Quality:**

3

**Strengths And Weaknesses:**

Strengths. The problem is quite canonical and the gives a complete answer for what concern rates.
The paper is also clear and easy to read.

Weaknesses: Technically, the treatment is quite standard.
Also, such exact support  recovery questions require to assume a separation condition (either y is independent of coordinate x_i, or the dependence is above a minimum value). This separation is typically considered unrealistic, so other forms of recoveries are of greater relevance.

---

> ### Author Rebuttal · Authors · 2025-07-31
>
> We sincerely appreciate your recognition of the value, completeness, and readability of our work.
> Below, we will provide a response that centers around weaknesses and questions related to our paper.
>
> # Weaknesses:
>
> ## 1. Technically, the treatment is quite standard. Also, such exact support recovery questions require to assume a separation condition (either y is independent of coordinate $x_i$, or the dependence is above a minimum value). This separation is typically considered unrealistic, so other forms of recoveries are of greater relevance.
>
> **Reply:** We acknowledge that some elements in our paper, such as the loss function settings and the definition of the minimax separation rates, are standard.
>  However, the results we obtain within this framework are novel and offer fresh insights:
>
>  **(i)** We propose general and non-asymptotic minimax separation rates of the variable selection problem in the SpAM, which is, to our knowledge, the first optimal finite-sample guarantee.
>
>  **(ii)** We provide a selection procedure adaptive to the unknown smoothness parameter of the Sobolev spaces, and prove that an additional $\log(\log n)$ term is necessary for this adaptation.
>
>  **(iii)** We demonstrate that the optimal function estimators can be suboptimal in univariate function selection, highlighting the distinction between function estimation and variable selection in SpAM.
>
> Additionally, in a more realistic setting where **the true signal may fall below the minimax threshold**, our selectors still have some useful properties:
>
> **(i)** The selector from Theorem 1 selects at most $2s$ variables with probability at least $1-\delta$ (proved by following equation (33) in Appendix D.2).
>
> **(ii)** The selector from equation (10) ensures $\hat S \subseteq S$ with probability at least $1-\delta$, i.e., it guarantees zero false positives (proved by following Appendix C.2.3).
>
> These guarantees hold without knowing the signal strength of each $f_j$ in advance, showing that our selectors remain both sparse and interpretable under a realistic condition. We hope these results could address your concern about the unrealistic assumptions.
>
> # Questions:
>
> ## 1. Can you describe some concrete examples in which this exact recovery criterion (and the associated minimum strength assumption) are appropriate, and the work presented here provides some insights?
>
> **Reply:** Thank you for the question. We will illustrate the value of our selectors through two examples:
>
> **Multi-channel signal selection.** Consider a system with $p$ channels, some of which carry a true signal while the rest are pure white noise, and we aim to identify those channels with a signal. Then our selector (equation (10)) achieves that, with high probability: **(i)** no noise-only channel is selected; **(ii)** any channel whose signal strength exceeds the minimax separation rate (equation (9)) will be selected. In this way, our selector provides a false-positive-free method for this problem.
>
> **Norm estimation.** Consider estimating the $L_2$ norm of each univariate function, i.e., jointly estimating $N(f) = (\\| f\_1 \\|\_2, \cdots, \\| f\_p \\|\_2 ) \in \mathbb R^p$. These norms quantify each covariate’s strength and play a key role in confidence interval construction. By using our thresholding selector (equation (6)), we define the norm estimator for each $f_j$ as
>  $$
> \hat N_j = \sqrt{ \sum\_{i=1}^{K } X\_{ij}^2 - K/n } \cdot \mathbf1 \left( n\sum\_{i=1}^{K } X\_{ij}^2-K \ge 2\sqrt{K \log(p/s)} +2\log(p/s)\right), \text{ for all } j \in [p],
> $$
> where $K= \min \left\\{ k \in \mathbb{N}^+: \mu_k \le \frac{\sqrt{k \log(p/s)}}{n} \right\\}$. We can show that the minimax estimation rate
> $$
> \inf_{\hat N} \sup_{f \in \mathcal F_s} \mathbf E_f \left\\| \hat N - N(f) \right\\|_2^2
> \asymp \frac{s \log(p/s)}{n} + s \times \max\_{k \in \mathbb N^+}\left( \frac{ \sqrt{k \log(p/s)} }{n } \wedge \mu_k \right),
> $$
> and our norm estimator is minimax rate-optimal.
> However, a naive plug-in estimator based on the optimal function estimation (in Section 5) may fail to attain the minimax optimality.
>
> These two examples illustrate both the practical relevance and theoretical insights offered by our truncation-based selectors.
>
>
> ## 2. Above the (speculation) formula in Section 2.2. Here you did not yet define formally the problem, so it is unclear what does this speculation refer to.
>
> **Reply:** Thank you for the suggestion. We agree that the speculation formula is clearer when presented after the formal problem definition. We have moved the speculation formula to appear immediately after Definition 1 in the revised manuscript. This formula conjectures that the minimax separation rate (for sparse multiple testing) is the sum of: **(i)** the high‑dimensional selection error $\tfrac{\log(ep/s)}{n}$ (Abraham et al.[1]), and **(ii)** the univariate detection rate $\max_{k\in\mathbb N^+}\bigl(\tfrac{\sqrt{k}}{n}\wedge\mu_k\bigr)$ (Baraud[2]). Then Theorems 1 and 2 show that this conjecture is incorrect, and Remark 1 explains why.
>
> ## 3.  Above Theorem 1: Please avoid "sophisticated analysis", this is a judgement that you are expressing about your own work.
>
> **Reply:** Thank you for pointing this out. We have removed this subjective phrasing and reviewed the manuscript to ensure no similar expressions remain.
>
> **Reference**
>
> [1] Kweku Abraham, Ismaël Castillo, and Étienne Roquain. Sharp multiple testing boundary for sparse sequences. The Annals of Statistics, 52(4):1564-1591, 2024.
>
> [2] Yannick Baraud. Non-asymptotic minimax rates of testing in signal detection. Bernoulli, 8(5): 577-606, 2002.

---

### Official Review · Reviewer_hVsu · 2025-07-09

**Clarity:** 3
**Significance:** 3
**Originality:** 2
**Rating:** 4
**Confidence:** 4

**Summary:**

This paper considers variable selection in a sparse additive model. Assuming the existence of an orthonormal basis, the  problem is reformulated as follows: For each $1\leq j\leq p$, we observe an infinite sequence $X_{ij}\sim N(\theta_{ij}, 1/n)$, where $\theta_{ij}$ is the true coefficient of the $j$th function on the $i$th basis, satisfying that $\sum_i \frac{\theta^2_{ij}}{\mu_i}<1$. It is assumed that for most $j$, the infinite sequence {$\theta_{ij}$}$_{i\geq 1}$ is a zero sequence (i.e., the $j$th function $f_j$ is a zero function). The variable selection goal is to identify those $j$ such that $f_j$ is nonzero.

The authors consider two loss metrics, the sum of FDR and FNR, and the wrong recovery probability. For loss metric, the minimax order of signal strength for success has been derived. Compared to the previous work on estimation, the authors found that the minimax order has a factor of $\log(p/s)$ or $\log(p)$, which is considered as the ``interplay" between high-dimensional selection error and estimation error of each individual function $f_j$.

**Questions:**

When $\mu_k$'s are unknown, how to implement the selector?

**Ethical Concerns:**

["NO or VERY MINOR ethics concerns only"]

**Final Justification:**

Based on the author rebuttal, I am more convinced that the paper contains some technically significant result. Even though I still think this paper is more of a theoretical statistical journal paper than a conference paper, I'm willing to raise my score.

**Limitations:**

No numerical work and real applications.

**Quality:**

3

**Strengths And Weaknesses:**

1. There is a gap here transiting from the sparse additive model to the model of $X_{ij}=\theta_{ij}+N(0,1/n)$ for infinite $j$. For people who are familiar with this problem, it may be straightforward. But for general readers, it is better to explain it. For example, there is no sample size $n$ in the current setting. The readers may get confused why $n$ appears here (it is actually a re-parametrization, right?).

2. The "interplay" discovered here is a result of the loss function for variable selection. I don't really understand why this is a "subtle and critical" phenomenon. For example, in Gaussian sequence model, we need to set the threshold at $\log(p/s)$ or $\log(p)$ to achieve successful selection. What is special here?

3. In a remark, the authors mention that the non-trivial part is due to the truncated chi-square statistics, in which we need to choose the truncation parameter $K$. However, is there any particular technical difficulty here? The novelty of the theory compared to those for Gaussian sequence model needs to be clarified.

4. A major limitation of this work is that there is completely no numerical work and applications.

---

> ### Author Rebuttal · Authors · 2025-07-31
>
> We sincerely appreciate your thoughtful summary and evaluation of our work. Below, we will provide a response that centers around weaknesses and questions related to our paper.
>
> # Weaknesses:
>
> ## 1. There is a gap here transiting from the sparse additive model to the model of $X_{ij} = \theta_{ij} + N(0,1/n)$ for infinite $j$...
>
> **Reply:** We illustrate the connection between these two models through a simple example: consider $N$ i.i.d. observations $(X_1^k,\cdots, X_p^k, Y^k)\_{k\in[N]}$ follow the sparse additive model
> $$Y^k = \sum_{j\in[p]} f_j(X_j^k)+\epsilon^k = \sum_{j\in[p]} \sum_{i\in \mathbb N^+} \theta_{ij} \psi_i(X_j^k) + \epsilon^k,$$
> where $\\{\psi_i \\}\_{i\in \mathbb N^+}$ is an orthonormal basis on $[0,1]$, each $X_j^k\sim\mathrm{Unif}(0,1)$ independently, and $\epsilon^k\sim N(0,\sigma^2)$. Define the projection estimator
> $$
> \hat\theta_{ij} = \frac1N \sum_{k\in[N]} \psi_i(X_j^k)\cdot Y^k
> $$
> By orthonormality and independence, we have
> $\hat\theta\_{ij}\approx \theta\_{ij} + \frac1N \sum\_{k\in[N]} \psi_i(X_j^k)\cdot \epsilon^k$, and therefore $\hat\theta_{ij}$ approximately follows from the distribution $N(\theta_{ij}, \sigma^2/N)$.
>
> By a re-parametrization $n := N/\sigma^2$, we then establish the link between the sparse additive model and the Gaussian white noise model $X_{ij} = \theta_{ij} + N(0,1/n)$. For a rigorous account of this relationship, see, e.g., Tsybakov[1] and Reiß[2]. We will add the above analysis to the appendix. The Gaussian white noise model simplifies the analysis by avoiding technical complexities (e.g., variable correlations) while preserving focus on essential structures (e.g., sparsity and smoothness constraints). Therefore, we adopt this model for our study.
>
> ## 2. The "interplay" discovered here is a result of the loss function for variable selection. I don't really understand why this is a "subtle and critical" phenomenon. For example, in Gaussian sequence model, we need to set the threshold at $\log(p/s)$ or $\log p$ to achieve successful selection. What is special here?
>
> **Reply:** To illustrate why the interplay between sparsity and truncation is subtle and critical, we use the Sobolev setting ($\mu_k = k^{-2\alpha}$) as an example and answer from two aspects:
>
> **How sparse structure affects the truncation.**
> We revisit the function estimation problem in SpAM, where Raskutti et al.[3] established the minimax rate as
> $$
> \inf_{\hat{f}} \sup_{f \in \mathcal F_s} \mathbf E_f \\| \hat{f}(X) - f  \\|_2^2  \asymp s \times \frac{\log(ep/s)}{n}  + s  \times \max\_{k \in \mathbb N^+} \left( \frac{k}{n} \wedge \mu_k \right),
> $$
> hence the optimal truncation $K$ satisfies $K \asymp n^{\frac1{1+2\alpha}}$, independent of the sparsity structure $\log(p/s)$.
>
> Returning to the variable selection problem in SpAM, in equation (8) we show that the optimal truncation level satisfies $\frac{\sqrt{K \log(p/s)}}{n} \asymp K^{-2\alpha}$, i.e., $K \asymp \left(\frac{n^2}{\log(p/s)} \right)^{\frac{1}{1+ 4 \alpha}}$. By contrast, the term $\log(p/s)$ does not affect the truncation $K$ for optimal function estimation, while it directly affects the truncation $K$ for optimal variable selection: the higher the term $\log(p/s)$, the fewer basis terms are needed.
> This phenomenon appears under both the FDR+FNR loss (Section 3.1) and the Hamming loss (Section 3.2), highlighting a fundamental distinction between function estimation and variable selection. Exploiting this insight, we propose Theorem 7 and demonstrate that an optimal function estimation can be suboptimal for the variable selection in SpAM.
>
> **How sparse structure affects the threshold.** Consider the selector $\hat \eta_j^{test} (X_{\cdot j}) = \mathbf1 \left( \sum_{i=1}^{K } X_{ij}^2 - \frac{K}{n}  \ge  \lambda^2 \right)$ with $K \asymp \left(\frac{n^2}{\log(p/s)} \right)^{\frac{1}{1+ 4 \alpha}}$. By Theorems 1 and 2, we reveal a phase transition in the optimal threshold $\lambda^2$: If $n\prec \log^{1+2\alpha}(p/s)$, then $\lambda^2\asymp\tfrac{\log(p/s)}n$; if $n\gtrsim\log^{1+2\alpha}(p/s)$, then $\lambda^2 \asymp n^{-\frac{4\alpha}{4\alpha+1}} \log ^{\frac{2\alpha}{4\alpha+1}}(p/s)$. No such phase transition phenomenon appears in the Gaussian sequence model (Butucea et al.[4]; Abraham et al.[5]).
>
> In summary, we do not merely "found that the minimax order has a factor of $\log(p/s)$ or $\log p$"; we derive a general and nonasymptotic minimax separation rate for variable selection, which in turn reveals the fundamental distinction between function estimation and variable selection in SpAM.
>
>
> ## 3. In a remark, the authors mention that the non-trivial part is due to the truncated chi-square statistics, in which we need to choose the truncation parameter $K$. However, is there any particular technical difficulty here? The novelty of the theory compared to those for Gaussian sequence model needs to be clarified.
>
> **Reply:** We will discuss our technical difficulty and novelty from the perspectives of upper and lower bounds, respectively:
>
> **Upper bounds:**
>
> **(i)** We finely balance the truncation level $K$ against the signal strength (in equation (28) and Lemma 4), ensuring that the signal after truncation remains large enough for reliable selection.
>
> **(ii)** Controlling the missed selection probability in SpAM requires more complex probabilistic tools (in Appendix C.2.2) than in the Gaussian sequence model.
>
> **(iii)** To bound the false discovery rate, we have to partition the expectation into three cases and analyze each separately (in Appendix D.2), which increases the technical difficulty.
>
> **Lower bounds:**
>
> **(i)** Rather than relying on classical Le Cam's or Fano's method, we adopt a novel probabilistic technique (inspired by Butucea et al.[6]) that leverages the monotonicity of likelihood ratios to analyze the selection lower bound (in Appendix C.1). We believe this technique could also be effective for other complex variable selection problems.
>
> **(ii)** In handling adaptation to the Sobolev smoothness parameter $\alpha$ in Theorem 5, we construct joint priors on $\alpha$ and the element-wise signal strengths with inherent dependence. This coupling complicates the calculations and scaling arguments of the chi-square divergence (in Appendix G.1). Therefore, the lower bound analysis is more involved than in the Gaussian sequence model, as the latter does not involve smoothness parameters or truncation levels.
>
> Additionally, Theorem 7 reveals a novel "**estimation-selection gap**" in SpAM: some estimators that are optimal for function estimation cannot yield optimal variable selection performance. This divergence does not occur in the Gaussian sequence model, where a proper thresholding estimator simultaneously guarantees minimax optimality in both estimation and variable selection (Butucea et al.[4]; Ndaoud[7]).
>
>
> ## 4. A major limitation of this work is that there is completely no numerical work and applications.
>
> **Reply:** In Appendix B, we have presented three sets of numerical experiments, all evaluated via FDR+FNR loss and Hamming loss, that respectively examine:
>
> **(i)** The impact of dimension $p$ and signal strength $r^2$.
>
> **(ii)** The impact of $n$ and different truncation locations.
>
> **(iii)** The impact of unknown smoothness parameters.
>
> These experiments confirm the minimax optimality of our methods and illustrate that, in sparse additive models, an estimator that is minimax optimal for function estimation may not achieve minimax optimal variable selection (see results in Appendix B.2). We believe these numerical results could reinforce our theoretical contributions.
>
>
> # Questions:
>
> ## 1. When $\mu_k$'s are unknown, how to implement the selector?
>
> **Reply:** Thank you for this question. In Section 4, we handle the adaptation problem by working on a Sobolev space ($\mu_k\asymp k^{-2\alpha}$) with unknown smoothness parameter $\alpha>0$. We then introduce a fully data-driven selector (equation (12)) that adapts to the unknown smoothness $\alpha$. Theorems 4 and 5 establish that this adaptive procedure attains the minimax separation rate up to an extra $\log(\log n)$ term, which is also the necessary cost of adaptation. Since Sobolev spaces are a canonical model in nonparametric statistics (Tsybakov[1]) and deep learning theory (Li and Lin[8]), our approach provides a practical implementation when $\mu_k$'s are unknown.
>
>
> **Reference**
>
> [1] Alexandre B. Tsybakov. Introduction to nonparametric estimation. Springer Ser. Stat. New York, NY: Springer, 2009.
>
> [2] Markus Reiß. (2008). Asymptotic equivalence for nonparametric regression with multivariate and random design. The Annals of Statistics, 36(4), 1957-1982.
>
> [3] Garvesh Raskutti, Martin J Wainwright, and Bin Yu. Minimax-optimal rates for sparse additive models over kernel classes via convex programming. Journal of machine learning research, 13(2), 2012.
>
> [4] Cristina Butucea, Mohamed Ndaoud, Natalia A. Stepanova, and Alexandre B. Tsybakov. Variable selection with Hamming loss. The Annals of Statistics, 46(5):1837-1875, 2018.
>
> [5] Kweku Abraham, Ismaël Castillo, and Étienne Roquain. Sharp multiple testing boundary for sparse sequences. The Annals of Statistics, 52(4):1564-1591, 2024.
>
> [6] Cristina Butucea, Enno Mammen, Mohamed Ndaoud, and Alexandre B. Tsybakov. Variable selection, monotone likelihood ratio and group sparsity. The Annals of Statistics, 51(1):312-333, 2023.
>
> [7] Mohamed Ndaoud. Interplay of minimax estimation and minimax support recovery under sparsity. volume 98 of Proceedings of Machine Learning Research, 647-668, 2019.
>
> [8] Li, Yicheng, and Qian Lin. On the asymptotic learning curves of kernel ridge regression under power-law decay. Advances in Neural Information Processing Systems 36 (2023): 49341-49364.

---

> > ### Comment · Reviewer_hVsu · 2025-08-02
> > **Reply to Author Response**
> >
> > I thank the authors for a detailed response. I will raise my score to 4.

---

> > > ### Author Response · Authors · 2025-08-03
> > > **Thank you for your review！**
> > >
> > > Thank you for your kind feedback and for increasing your score. We greatly appreciate your suggestions and questions.

---

### Comment · Area_Chair_gbTc · 2025-08-04
**Author-Reviewer Discussion Period Ending Soon**

Dear Reviewers,

The Author–Reviewer discussion period has begun and will end on August 6. Please read the rebuttal and let us know whether it satisfactorily addresses your concerns. If not, could you specify what remains inadequate? Your response will help us evaluate the paper and assist the authors in improving their work.

Please avoid responding at the last minute, as the authors may not have sufficient time to clarify your concerns.

Thank you!

Best,
AC

---

### Author Response · Authors · 2025-08-08
**A general response**

We thank all reviewers for their thoughtful and constructive feedback and for recognizing the main contributions of our work. In particular, we appreciate the reviewers for highlighting the following contributions:

**(i)** the minimax rate-optimality of univariate function selection in sparse additive models (SpAM) **(Reviewers BfNF, 8MUY, FqYP, oppP)**;

**(ii)** the extension to adaptive selectors with unknown smoothness **(Reviewers hVsu, 8MUY, oppP)**;

**(iii)** the distinction between variable selection and function estimation in SpAM **(Reviewers 8MUY)**.

We also thank all reviewers for acknowledging the clarity of the manuscript’s presentation.

# Revisions

We carefully addressed every comment from each reviewer and believe we have successfully responded to most of the concerns. In the revised manuscript, we:

 **(i)** introduced a multiplicative noise factor $\sigma$ into the white-noise term in equation (1) **(Reviewer FqYP)**;

**(ii)** added a discussion of the finite-dimensional case after Corollary 1 **(Reviewer 8MUY)**.

Furthermore, we expanded the appendix with:

**(i)** an introduction from the empirical regression model to the Gaussian white noise model **(Reviewer hVsu)**;

**(ii)** some examples illustrate both the practical relevance and theoretical insights of our selectors **(Reviewer BfNF)**;

**(iii)** discussions of heterogeneity across component function spaces and adaptation to sparsity $s$ **(Reviewer oppP)**.

We also added some relevant references and corrected several unclear or incorrect phrasings throughout the text **(Reviewers BfNF, FqYP, oppP)**.

We believe our clarifications and revisions have improved the manuscript. We thank the reviewers again for their valuable feedback.

---

### Note · Authors · 2025-08-12

We thank the reviewers and the area chair for their constructive feedback and thoughtful evaluation. As noted in our general response, we addressed every comment and believe we have resolved most of the concerns. Guided by the reviewers’ suggestions, we have made a few targeted revisions to enhance the readability, accuracy, and extensibility of our manuscript.

---

### Decision · Program_Chairs · 2025-09-17

**Decision:**

Accept (spotlight)

**Comment:**

Consider the problem of univariate function selection in a continuous-time sparse additive model with white Gaussian noise. This paper characterizes the minimax risk for both multiple testing and support recovery. The case of support recovery under unknown smoothness is also studied. Finally, the paper presents an example where optimal estimation does not lead to optimal support recovery, demonstrating that the two tasks are fundamentally different.

The paper is well written. The problem studied is fundamental, and the minimax risk results are clean. The reviewers, several of whom are experts in related research topics, are all positive toward this paper, and the authors satisfactorily addressed the questions raised. Reviewer BfNF believes the technical novelty of this paper is limited; nevertheless, I think such a fundamental and novel result already deserves publication.

Given the above, I suggest accepting this paper as a spotlight.